

# Effective field theory for hydrodynamics without boosts

**Jay Armas**[1,2]★ **and Akash Jain**[3]†

**1** Institute for Theoretical Physics, University of Amsterdam,
1090 GL Amsterdam, The Netherlands
**2** Dutch Institute for Emergent Phenomena (DIEP), 1090 GL Amsterdam, The Netherlands
**3** Department of Physics & Astronomy, University of Victoria,
1700 STN CSC, Victoria, BC, V8W 2Y2, Canada

★ j.armas@uva.nl, † ajain@uvic.ca

## Abstract

We formulate the Schwinger-Keldysh effective field theory of hydrodynamics without boost symmetry. This includes a spacetime covariant formulation of classical hydrodynamics without boosts with an additional conserved particle/charge current coupled to Aristotelian background sources. We find that, up to first order in derivatives, the theory is characterised by the thermodynamic equation of state and a total of 29 independent transport coefficients, in particular, 3 hydrostatic, 9 non-hydrostatic non-dissipative, and 17 dissipative. Furthermore, we study the spectrum of linearised fluctuations around anisotropic equilibrium states with non-vanishing fluid velocity. This analysis reveals a pair of sound modes that propagate at different speeds along and opposite to the fluid flow, one charge diffusion mode, and two distinct shear modes along and perpendicular to the fluid velocity. We present these results in a new hydrodynamic frame that is linearly stable irrespective of the boost symmetry in place. This provides a unified covariant stable approach for simultaneously treating Lorentzian, Galilean, and Lifshitz fluids within an effective field theory framework and sets the stage for future studies of non-relativistic intertwined patterns of symmetry breaking.



# 1 Introduction

Hydrodynamics describes the long-wavelength collective behaviour of low-energy excitations in a broad range of physical systems. In this regime, the dynamics is insensitive to most microscopic details and is universally captured by a set of conservation laws. The range of applicability of hydrodynamics spans widely separated scales, in particular those of quantum gravity [1, 2], viscous electron flows [3, 4], biological fluids [5], and the dynamics of black hole accretion disks [6], to mention only a few. Traditionally, hydrodynamics has been a phenomenological field of study. One specifies the symmetry-breaking pattern; postulates a set of currents with associated conservation laws; invokes the second law of thermodynamics (through the positivity of the divergence of an entropy current) together with Onsager's relations, and determines the constitutive relations in a gradient expansion [7]. While this classical approach has been extremely successful, one expects that symmetry alone should be sufficient to characterise the hydrodynamic regime.

Treating hydrodynamics as a *bona fide* thermal field theory, a more fundamental approach

has been developed in the context of relativistic fluids in the past few years [8–10], and has recently been adapted to Galilean-invariant fluids as well [11]. This formulation is based on the Schwinger-Keldysh effective field theory (EFT) framework for non-equilibrium thermal systems; see [12] for a review. The starting point of this EFT framework is a generating functional from which correlation functions of hydrodynamic operators and hydrodynamic equations of motion can be derived. In addition, the framework systematically accounts for the effects of stochastic/thermal noise on the hydrodynamic evolution via stochastic interactions. In order to describe out-of-equilibrium thermal systems, the EFT generating functional must satisfy certain requirements, such as KMS symmetry, which lead to an emergent second law of thermodynamics and implementation of the Onsager's relations at the classical level, in addition to fluctuation-dissipation constraints on the correlation functions. The main goal of this work is to develop a Schwinger-Keldysh EFT for the hydrodynamic description of physical systems that lack any boost symmetry, Lorentzian or Galilean, to begin with. In particular, this covers systems which have their boost symmetry explicitly broken due to the presence of a background medium[1] or systems that do respect a boost symmetry but is not explicitly manifest at the macroscopic level. Systems without a boost symmetry are ubiquitous in non-Fermi liquid phases of matter such as metallic quantum critical systems [15]. Fermi-liquids can also exhibit phases characterised by the absence of a boost symmetry, the prime example being liquid helium-3 at sufficiently low temperatures [14].[2] In the realm of classical physics, various many-body systems in soft matter physics and biophysics [17] do not respect Galilean boost symmetry. Common examples include models with self-propelled agents such as flocks of birds and colonies of bacteria swimming in a medium.

In general, however, physical systems that break boost symmetry also exhibit other patterns of symmetry breaking. In the context of quantum matter, spatial translations are usually also spontaneously broken, due to the presence of the ionic lattice, or explicitly broken due to the presence of impurities [18]. Charge density wave phases are one such example; see also [19, 20]. In the setting of classical fluids, self-propelled agents break spacetime translations explicitly due to the presence of driving forces [21], while active liquid crystal phases can break translations and rotations spontaneously [22]. Such situations compromise the gradient expansion of hydrodynamics and thus it is important to move away from traditional treatments and understand what are *the rules of the game* for building hydrodynamics models with intertwined patterns of symmetry breaking.

Schwinger-Keldysh EFT provides a controlled framework for developing such hydrodynamic theories and studying stochastic corrections to classical hydrodynamics. In particular, it was shown recently in the context of isotropic relativistic fluids that stochastic corrections break the hydrodynamic derivative expansion at third derivative order, leading to non-classical contributions to hydrodynamic correlation functions [23]. However, such effects may possibly appear earlier in the derivative expansion in systems with specific kinds of broken symmetries. The work presented here considers only the case of broken boost symmetry and has a three-fold purpose: (1) to accurately classify the transport properties of hydrodynamics without boosts in the presence of a conserved U(1) particle-number/charge current; (2) to provide a unified field theoretic framework that can simultaneously describe Lorentzian as well as Galilean and Lifshitz fluids,[3] which can be obtained by restoring different types of boost symmetries or

---

[1]In [13, 14] the case of spontaneous breaking of Lorentz boost symmetry was considered. This is distinct from the setup considered in this paper, where the boost symmetry is explicitly broken and the respective Ward identity is absent to begin with. This can be thought of as being accomplished by "integrating out" the medium through which the fluid is moving.

[2]In the context of quantum matter, it has also been argued that electron flows in graphene may break Lorentz invariance due to the presence of long-range Coulomb interactions [16].

[3]This framework could also potentially describe Carrollian-boost invariant fluids [24–26], but we have not explored this possibility here.

taking different scaling limits; and (3) to provide the necessary foundations for future explorations addressing more complicated patterns of symmetry breaking and associated stochastic contributions.

Several recent works have motivated this study. Earlier literature established a classical covariant approach to ideal fluids without boost symmetry, by coupling the fluid to Aristotelian geometry [26]. A full treatment of one-derivative corrections was carried out in [27], but the U(1) current responsible for particle-number/charge conservation was not introduced. A linearised analysis of fluctuating isotropic and homogeneous configurations in charged hydrodynamics without boosts was done in [28]. A classification of first order transport in flat spacetime with the additional U(1) current was undertaken in [29], but a complete analysis of the second law constraints was not carried out. In this work, we develop further all of these lines of research by providing a complete covariant treatment and classification of transport in hydrodynamics without boosts within a field theoretic framework, including the presence of a U(1) current, and consider the most general fluctuation analysis around equilibrium states that are inherently anisotropic.

In contrast with all the previous literature, we present our results in a new hydrodynamic frame, which we call *density frame*, that is linearly stable (in the sense of [30–32]) irrespective of the boost symmetry in place (Galilean or Lorentzian), or absence thereof, and is thus better suited for potential numerical simulations. This frame choice aligns the fluid velocity with the flow of momentum, rather than the flow of internal energy (as in the Landau frame) or charge/particle-number (as in the Eckart frame). Note that momentum is a reference-frame dependent quantity. Therefore, when employed in Galilean or relativistic hydrodynamics, the density frame will lead to a manifestly non-covariant representation of the respective constitutive relations. However, the equations of motion are still manifestly covariant and boost-invariant (up to second derivative corrections). We emphasise that hydrodynamic models, irrespective of the hydrodynamic frame utilised to represent the constitutive relations, are not suitable to make reliable universal predictions about gapped "high-energy" modes. In this sense, the aforementioned linear stability (i.e. the absence of unstable gapped modes) in the density frame is *not* a physical prediction of the model. It is rather a technical characteristic of the model that makes it "more suitable" for setting up initial-value problems aimed at exploring the low-energy long-wavelength physics of fluids.

This paper is organised as follows. In section 2, we introduce classical aspects of hydrodynamics without boost symmetry, conservation laws as well as entropy production, and the basics of Aristotelian geometry to which these fluids couple to. In section 3, we use these considerations in order to formulate the Schwinger-Keldysh effective field theory for these systems. In section 4, we write down the specific Lagrangian that includes all the dissipative and non-dissipative transport coefficients that characterise the effective theory up to first order in a gradient expansion. In section 5, we examine special limits where one recovers Lorentzian, Galilean, and Lifshitz fluids. In section 6, we study fluctuations around generic anisotropic equilibrium configurations and obtain explicit expressions for sound, shear, and charge diffusion modes in a linearly stable hydrodynamic frame. Finally, we conclude with some discussion in section 7. Appendix A is dedicated to expressing our results in the Landau frame in order to compare with the previous literature. Appendix B provides the interaction Lagrangian for the linearised effective field theory of hydrodynamics without boosts, which can be used for studying stochastic contributions to hydrodynamic correlation functions.

# 2 Classical boost-agnostic hydrodynamics

In this section, we review various aspects of classical boost-agnostic hydrodynamics. We start with the energy, momentum, and charge/particle-number conservation equations and use the second law of thermodynamics to derive the constitutive relations of an ideal fluid without boost symmetry. We discuss how to introduce curved background sources into these equations coupled to various hydrodynamic observables, which will be crucial for our subsequent discussion of the EFT framework. Following this, we outline a generic procedure to implement the second law constraints at arbitrarily high orders in the derivative expansion using the adiabaticity equation. A more concrete construction of the allowed one-derivative corrections is presented later in section 4.

## 2.1 Ideal hydrodynamics on flat background

### 2.1.1 Symmetries and conservation laws

Hydrodynamics is a theory of locally conserved quantities. One starts by outlining the complete set of Noether currents associated with any global symmetries that the system might enjoy, and expresses various "fluxes" in terms of the conserved "densities", arranged in a perturbative expansion in derivatives. For a given set of such "constitutive relations", the time evolution of the conserved densities is determined by their respective conservation equations. In typical hydrodynamic systems, these conserved densities are the energy $\epsilon$, momentum $\pi^i$, and particle number density $n$ of the fluid, associated with time and space translational invariance and an abstract internal U(1) phase shift invariance of the theory. The associated fluxes are the energy flux $\epsilon^i$, stress tensor $\tau^{ij}$, and mass/particle number flux $j^i$, with conservation equations

$$
\begin{aligned}
\text{Energy conservation:} \quad & \partial_t \epsilon + \partial_i \epsilon^i = 0 \,, \\
\text{Momentum conservation:} \quad & \partial_t \pi^j + \partial_i \tau^{ij} = 0 \,, \\
\text{Continuity equation:} \quad & \partial_t n + \partial_i j^i = 0 \,.
\end{aligned}
\tag{1}
$$

In addition, hydrodynamic systems usually feature rotational invariance and some kind of boost invariance. Provided that the fluid does not carry an intrinsic spin density, rotational invariance requires the orbital angular momentum density to be conserved

$$
\text{Angular-momentum conservation:} \quad \partial_t \left( \pi^i x^j - \pi^j x^i \right) + \partial_k \left( \tau^{ki} x^j - \tau^{kj} x^i \right) = \tau^{ji} - \tau^{ij} \,,
\tag{2}
$$

ensuring the stress tensor to be symmetric. If the theory is required to be invariant under Galilean boosts, the center of inertia will need to be conserved

$$
\text{Center-of-mass conservation:} \quad \partial_t \left( m\, n\, x^i - \pi^i t \right) + \partial_k \left( m\, j^k x^i - \tau^{ki} t \right) = m\, j^i - \pi^i \,,
\tag{3a}
$$

where $m$ is the constant mass per particle. This leads to the momentum density being aligned with the mass flux $\pi^i = m\, j^i$. Similarly, we have a conserved center-of-energy in the relativistic case

$$
\text{Center-of-energy conservation:} \quad \partial_t \left( \frac{1}{c^2} \epsilon\, x^i - \pi^i t \right) + \partial_k \left( \frac{1}{c^2} \epsilon^k x^i - \tau^{ki} t \right) = \frac{1}{c^2} \epsilon^i - \pi^i \,,
\tag{3b}
$$

where $c$ is the speed of light, equating the momentum density to the energy-flux $\pi^i = \epsilon^i/c^2$ instead.[4] The paradigm of the present work is to study systems which might not necessarily respect a boost symmetry. In this sense, we do not tie $\pi^i$ to either $j^i$ or $\epsilon^i$. We will still focus on systems respecting rotational invariance (on hydrodynamic length scales), so $\tau^{ij}$ is assumed to be symmetric.

---

[4]Note that in the relativistic theory, we have a conserved energy momentum tensor $T^{\mu\nu}$. Momentum density equalling the energy-flux is merely the statement that $T^{ti} = T^{it}$.

### 2.1.2 Constitutive relations and second law

The starting point of hydrodynamics is the assumption that the low-energy dynamics of the system near thermal equilibrium is entirely governed by its conserved operators: density $n$, energy density $\epsilon$, and momentum density $\pi^i$. Hydrodynamics is then characterised by the most generic expressions for the fluxes $j^i$, $\epsilon^i$, and $\tau^{ij}$, written in terms of the chosen variables and their spatial derivatives, i.e.

$$j^i[n, \varepsilon, \pi^i, \partial_i] \,, \qquad \epsilon^i[n, \varepsilon, \pi^i, \partial_i] \,, \qquad \tau^{ij}[n, \varepsilon, \pi^i, \partial_i] \,. \tag{4}$$

These are known as the *hydrodynamic constitutive relations*. Note that the temporal derivatives of various quantities are determined by eq. (1) and hence are not independent. Our assumption of *near-equilibrium* allows us to arrange the constitutive relations in a *derivative expansion*, truncated at a given order in derivatives according to the phenomenological sensitivity required. At any given order in the derivative expansion, the constitutive relations contain all the possible tensor structures made out of derivatives of $n$, $\epsilon$, and $\pi^i$, consistent with symmetries, appended with arbitrary *transport coefficients* as a functions of $n$, $\epsilon$, $\vec{\pi}^2$.

The hydrodynamic constitutive relations are required to respect certain phenomenological constraints. Most important of these is the "local second law of thermodynamics" that requires that there must exist an entropy density $s^t$ and an associated flux $s^i$ such that

$$\partial_t s^t + \partial_i s^i \geq 0 \,. \tag{5}$$

At the leading order in the derivative expansion, entropy density is merely given by an arbitrary function of $n$, $\epsilon$, and $\vec{\pi}^2$, i.e. $s^t = s(\epsilon, n, \vec{\pi}^2)$. Let us define intensive parameters: temperature $T(\epsilon, n, \vec{\pi}^2)$, chemical potential $\mu(\epsilon, n, \vec{\pi}^2)$, velocity $u^i(\epsilon, n, \vec{\pi}^2)$, and pressure $p(\epsilon, n, \vec{\pi}^2)$ via the thermodynamic relations: local first law of thermodynamics and Euler relation respectively

$$T\mathrm{d}s = \mathrm{d}\epsilon - \mu\,\mathrm{d}n - u^i\mathrm{d}\pi_i \,, \qquad p = Ts + \mu n + u^i\pi_i - \epsilon \,. \tag{6}$$

Due to rotational invariance, the velocity must be aligned with momentum, i.e. $u^i = \pi^i/\rho$, where $\rho$ is the momentum susceptibility. It is easy to check that

$$\partial_t s^t + \partial_i s^i = -\frac{1}{T^2}\left(\epsilon^i - (\epsilon+p)u^i\right)\partial_i T - \left(j^i - n\,u^i\right)\partial_i\frac{\mu}{T} - \left(\tau^{ij} - \rho\,u^iu^j - p\,\delta^{ij}\right)\partial_i\frac{u_j}{T} \,, \tag{7}$$

where we have identified the entropy flux as

$$T\,s^i = p\,u^i + \epsilon^i - \mu\,j^i - \tau^{ij}u_j + \mathcal{O}(\partial) \,. \tag{8}$$

We need to require that the RHS of eq. (7) is positive semi-definite for arbitrary fluid configurations. At the leading order in derivatives, this leads to the ideal fluid constitutive relations

$$\epsilon^i = (\epsilon+p)u^i + \mathcal{O}(\partial) \,, \qquad \tau^{ij} = \rho\,u^iu^j + p\,\delta^{ij} + \mathcal{O}(\partial) \,, \qquad j^i = n\,u^i + \mathcal{O}(\partial) \,,$$
$$s^i = s\,u^i + \mathcal{O}(\partial) \,. \tag{9}$$

The respective dynamics is given by substituting these into the conservation equations (1). Note that entropy is conserved at ideal order. We can, in principle, extend this analysis to higher orders in the derivative expansion (see [27]). We shall return to this when equipped with more tools.

The relation $s = s(\epsilon, n, \vec{\pi}^2)$, or equivalently $\epsilon = \epsilon(s, n, \vec{\pi}^2)$, can be understood as the microcanonical equation of state of the fluid and completely characterises its constitutive relations at ideal order through the thermodynamic relations (6). We note, however, that hydrodynamics as a physical system is better defined in the grand canonical ensemble, because a fluid element

is allowed to freely exchange particles, energy, and momentum with its surroundings. Keeping this in mind, we can take the fundamental dynamical fields to be $T$, $\mu$, $u^i$ instead of $\epsilon$, $n$, and $\pi_i$. In this case, the equation of state is given in terms of $p(T, \mu, \vec{u}^2)$ instead of $\epsilon(s, n, \vec{\pi}^2)$ with the thermodynamic relations: Gibbs-Duhem relation and Euler relation respectively

$$\mathrm{d}p = s\,\mathrm{d}T + n\,\mathrm{d}\mu + \pi_i \mathrm{d}u^i, \qquad \epsilon = Ts + \mu n + u^i \pi_i - p. \tag{10}$$

Recall that $\pi_i = \rho\, u_i$. These relations define $\epsilon$, $n$, and $\pi_i$ in terms of $T$, $\mu$, and $u^i$.

## 2.2 Coupling to background sources

### 2.2.1 Aristotelian background sources

We would like to introduce a set of Aristotelian background sources to which fluids without boost symmetry couple to, as discussed in [27]. These are similar to the Newton-Cartan sources prevalent in Galilean hydrodynamics, but with no Milne boost symmetry. The absence of this symmetry, in fact, makes these sources easier to implement in an effective theory. These are given by

$$\text{Clock-form:} \quad n_\mu, \qquad \text{Spatial metric:} \quad h_{\mu\nu}, \qquad \text{Gauge field:} \quad A_\mu, \tag{11}$$

where $h_{\mu\nu}$ is a symmetric matrix of signature $(0, 1, 1, 1, \ldots)$. $n_\mu$ and $h_{\mu\nu}$ can be vaguely understood as the time and space components of the relativistic spacetime metric $g_{\mu\nu}$, now being treated independently due to the lack of any boost symmetry. Since $h_{\mu\nu}$ is a degenerate matrix, it admits a zero-eigenvector $v^\mu$, normalised as $v^\mu n_\mu = 1$, such that $v^\mu h_{\mu\nu} = 0$. Its spatial components $v^i$ can be identified as the velocity of a lab frame observer. This can be used to define an "inverse spatial metric" $h^{\mu\nu}$ via the relations $h^{\mu\nu} n_\nu = 0$ and $h^{\mu\nu} h_{\nu\lambda} + v^\mu n_\lambda = \delta^\mu_\lambda$. Note that $h^{\mu\nu}$ is *not* the inverse of $h_{\mu\nu}$. Together,

$$\text{Frame velocity:} \quad v^\mu, \qquad \text{Inverse spatial metric:} \quad h^{\mu\nu}, \tag{12}$$

should be understood along the same lines as the inverse metric $g^{\mu\nu}$ in relativistic field theories. They are entirely fixed by the sources (11) via the conditions

$$v^\mu n_\mu = 1, \qquad v^\mu h_{\mu\nu} = 0, \qquad h^{\mu\nu} n_\nu = 0, \qquad h^{\mu\nu} h_{\nu\lambda} + v^\mu n_\lambda = \delta^\mu_\lambda. \tag{13}$$

The flat background limit is given as $n_\mu = \delta^t_\mu$, $h_{\mu\nu} = \delta^i_\mu \delta_{i\nu}$, $A_\mu = 0$, $v^\mu = \delta^\mu_t$, and $h^{\mu\nu} = \delta^{i\mu} \delta^\nu_i$.

Let $d$ be the number of spatial dimensions. The clock-form $n_\mu$ couples to the energy density and flux $\epsilon^\mu$, the $d$ independent components of the frame velocity $v^\mu$ couple to the momentum density $\pi_\mu$ (normalised as $v^\mu \pi_\mu = 0$), while the remaining $d(d+1)/2$ independent components of the spatial metric $h_{\mu\nu}$ couple to the stress tensor $\tau^{\mu\nu}$ (satisfying $\tau^{\mu\nu} n_\nu = 0$ and $\tau^{\mu\nu} = \tau^{\nu\mu}$), and finally the gauge field $A_\mu$ couples to the particle number current $j^\mu$. In terms of the non-covariant densities and fluxes we have

$$\epsilon^\mu = \begin{pmatrix} \epsilon \\ \epsilon^i \end{pmatrix}, \quad \pi_\mu = \begin{pmatrix} -v^k \pi_k / v^t \\ \pi_i \end{pmatrix}, \quad \tau^{\mu\nu} = \begin{pmatrix} n_k n_l \tau^{kl} / n_t^2 & -n_k \tau^{kj} / n_t \\ -n_k \tau^{ki} / n_t & \tau^{ij} \end{pmatrix}, \quad j^\mu = \begin{pmatrix} n \\ j^i \end{pmatrix}. \tag{14}$$

We will often use $\pi^\mu = h^{\mu\nu} \pi_\nu$ satisfying $\pi^\mu n_\mu = 0$. The coupling can be denoted in terms of the variation of an (equilibrium) effective action $S$ describing the theory as

$$\delta S = \int \mathrm{d}t \mathrm{d}^d x \sqrt{\gamma} \left( j^\mu \delta A_\mu - \epsilon^\mu \delta n_\mu - \pi_\mu \delta v^\mu + \frac{1}{2} \tau^{\mu\nu} \delta h_{\mu\nu} \right)$$

$$= \int \mathrm{d}t \mathrm{d}^d x \sqrt{\gamma} \left( j^\mu \delta A_\mu - \epsilon^\mu \delta n_\mu + \left( v^\mu \pi^\nu + \frac{1}{2} \tau^{\mu\nu} \right) \delta h_{\mu\nu} \right), \tag{15}$$

where $\gamma = \det(h_{\mu\nu} + n_\mu n_\nu)$. The second line is to highlight that $v^\mu$ is not an independent source.

We require the theory to be invariant under local diffeomorphisms parametrised by arbitrary invertible maps $x'^\mu(x)$ and U(1) gauge transformation parametrised by $\Lambda(x)$. We have collectively denoted the spacetime coordinates $x^\mu = (t, x^i)$. Their action on the background fields is defined as usual

$$n_\mu(x) \to n'_\mu(x') = \frac{\partial x^\nu}{\partial x'^\mu} n_\nu(x), \qquad h_{\mu\nu}(x) \to h'_{\mu\nu}(x') = \frac{\partial x^\rho}{\partial x'^\mu} \frac{\partial x^\sigma}{\partial x'^\nu} h_{\rho\sigma}(x),$$

$$A_\mu(x) \to A'_\mu(x') = \frac{\partial x^\nu}{\partial x'^\mu} (A_\nu(x) + \partial_\nu \Lambda(x)), \tag{16}$$

and similarly for $v^\mu$ and $h^{\mu\nu}$. Under infinitesimal version of these transformations, with $x'^\mu(x) = x^\mu + \xi^\mu(x)$, the diffeomorphisms merely act as Lie derivatives

$$n_\mu \to n_\mu + \$_\xi n_\mu, \quad h_{\mu\nu} \to h_{\mu\nu} + \$_\xi h_{\mu\nu}, \quad A_\mu \to A_\mu + \$_\xi A_\mu + \partial_\mu \Lambda. \tag{17}$$

Implementing this on the action variation in eq. (15), we can work out the covariant conservation equations

$$\text{Energy conservation:} \qquad \left(\nabla_\mu + F^n_{\mu\lambda} v^\lambda\right) \epsilon^\mu = -v^\nu \left(F_{\nu\mu} j^\mu - F^n_{\nu\mu} \epsilon^\mu\right) - \tau^{\mu\lambda} h_{\lambda\nu} \nabla_\mu v^\nu,$$

$$\text{Momentum conservation:} \qquad \left(\nabla_\mu + F^n_{\mu\lambda} v^\lambda\right)(v^\mu \pi^\nu + \tau^{\mu\nu}) = h^{\nu\lambda}\left(F_{\lambda\mu} j^\mu - F^n_{\lambda\mu} \epsilon^\mu\right) - \pi^\mu \nabla_\mu v^\nu,$$

$$\text{Continuity equation:} \qquad \left(\nabla_\mu + F^n_{\mu\lambda} v^\lambda\right) j^\mu = 0. \tag{18}$$

Here $F_{\mu\nu} = 2\partial_{[\mu} A_{\nu]}$ and $F^n_{\mu\nu} = 2\partial_{[\mu} n_{\nu]}$ are the field strengths associated with $A_\mu$ and $n_\mu$, and $\nabla_\mu$ is the covariant derivative operator associated with the connection[5]

$$\Gamma^\lambda_{\mu\nu} = v^\lambda \partial_\mu n_\nu + \frac{1}{2} h^{\lambda\rho} \left(\partial_\mu h_{\nu\rho} + \partial_\nu h_{\mu\rho} - \partial_\rho h_{\mu\nu}\right). \tag{19}$$

This connection satisfies

$$\nabla_\mu n_\nu = \nabla_\mu h^{\nu\lambda} = 0, \qquad \nabla_\lambda h_{\mu\nu} = -n_{(\mu} \$_\nu h_{\nu)\lambda}, \qquad h_{\nu\lambda} \nabla_\mu v^\lambda = \frac{1}{2} \$_\nu h_{\mu\nu},$$

$$\Gamma^\mu_{\mu\nu} + F^n_{\nu\mu} v^\mu = \frac{1}{\sqrt{\gamma}} \partial_\nu \sqrt{\gamma}, \qquad 2\Gamma^\lambda_{[\mu\nu]} = v^\lambda F^n_{\mu\nu}. \tag{20}$$

Note that this connection is torsional. In addition to torsional contributions on the left, the conservation of energy and momentum in eq. (18) is sourced by Lorentz force-like terms coupled to the field strengths $F_{\mu\nu}$ and $F^n_{\mu\nu}$, and pseudo-force terms coupled to the covariant derivative of the frame velocity $\nabla_\mu v^\nu$.

### 2.2.2 Hydrodynamics on curved background

The ideal order hydrodynamic constitutive relations (9) can be coupled to background sources naturally as

$$\epsilon^\mu = \epsilon u^\mu + p \vec{u}^\mu + \mathcal{O}(\partial), \qquad \pi^\mu = \rho \vec{u}^\mu + \mathcal{O}(\partial), \qquad \tau^{\mu\nu} = \rho \vec{u}^\mu \vec{u}^\nu + p h^{\mu\nu} + \mathcal{O}(\partial),$$

$$j^\mu = n u^\mu + \mathcal{O}(\partial), \qquad s^\mu = s u^\mu + \mathcal{O}(\partial), \tag{21}$$

---

[5]In Galilean theories, it is often convenient to work with a different connection, namely

$$\tilde\Gamma^\lambda_{\mu\nu} = v^\lambda \partial_\mu n_\nu + \frac{1}{2} h^{\lambda\rho} \left(\partial_\mu h_{\nu\rho} + \partial_\nu h_{\mu\rho} - \partial_\rho h_{\mu\nu}\right) + n_{(\mu} F_{\nu)\rho} h^{\rho\lambda},$$

which is Milne boost-invariant on backgrounds with $F^n_{\mu\nu} = 0$. Since we do not have any boost invariance, we choose to work with the simpler connection. The choice of connection has no bearing on the physical results.

where the covariant version of various hydrodynamic observables are defined in eq. (14). We have also taken $u^t = (1 - u^i n_i)/n_t$, which is just equal to 1 on a flat background, so that the covariant fluid velocity $u^\mu$ satisfies the normalisation condition $u^\mu n_\mu = 1$. The velocity of the fluid with respect to the Galilean frame is defined as $\vec{u}^\mu = u^\mu - v^\mu$, $\vec{u}_\mu = h_{\mu\nu} u^\nu = h_{\mu\nu} \vec{u}^\nu$, satisfying $\vec{u}^\mu n_\mu = 0$, $\vec{u}_\mu v^\mu = 0$. These constitutive relations should be understood as written in the grand canonical ensemble characterised by a function $p(T, \mu, \vec{u}^2)$, where $\vec{u}^2 = u^\mu u^\nu h_{\mu\nu} = \vec{u}^\mu \vec{u}^\nu h_{\mu\nu}$, and the thermodynamic relations (10). The structure of the constitutive relations is fixed by the second law of thermodynamics; it can be checked that eq. (21) represents the most generic leading derivative order constitutive relations satisfying

$$\left(\nabla_\mu + F^n_{\mu\lambda} v^\lambda\right) s^\mu = \frac{1}{\sqrt{\gamma}} \partial_\mu \left(\sqrt{\gamma} s^\mu\right) \geq 0 , \qquad s^\mu = s u^\mu + \mathcal{O}(\partial) . \tag{22}$$

The equations of motion of hydrodynamics are obtained by substituting the constitutive relations (21) into the conservation equations (18). We obtain

$$\frac{\vec{u}^\nu / T^2}{\sqrt{\gamma}} \delta_{\mathscr{B}}(\sqrt{\gamma} T^2 \rho) - (\epsilon + p) h^{\mu\nu} \delta_{\mathscr{B}} n_\mu + n h^{\mu\nu} \delta_{\mathscr{B}} A_\mu + \rho u^\sigma h^{\nu\rho} \delta_{\mathscr{B}} h_{\sigma\rho} = \mathcal{O}(\partial^2) ,$$

$$\frac{1}{\sqrt{\gamma}} \delta_{\mathscr{B}}(\sqrt{\gamma} T n) = \mathcal{O}(\partial^2) , \qquad \frac{1}{\sqrt{\gamma}} \delta_{\mathscr{B}}(\sqrt{\gamma} T s) = \mathcal{O}(\partial^2) . \tag{23}$$

Here $\delta_{\mathscr{B}}$ denotes a Lie derivative along $u^\mu / T$ combined with a gauge-shift along $(\mu - u^\mu A_\mu)/T$. Explicitly, we find

$$\delta_{\mathscr{B}} n_\mu = -\frac{1}{T^2} \partial_\mu T - \frac{1}{T} F^n_{\mu\nu} u^\nu, \qquad \delta_{\mathscr{B}} h_{\mu\nu} = 2 h_{\lambda(\mu} \nabla_{\nu)} \frac{u^\lambda}{T} + \frac{u^\lambda}{T} \nabla_\lambda h_{\mu\nu} ,$$

$$\delta_{\mathscr{B}} A_\mu = \partial_\mu \frac{\mu}{T} - \frac{1}{T} F_{\mu\nu} u^\nu . \tag{24}$$

Note the identity for arbitrary function $f(T, \mu, \vec{u}^2)$,

$$\frac{1}{\sqrt{\gamma}} \delta_{\mathscr{B}}(\sqrt{\gamma} f) = -\left[\left(T \frac{\partial f}{\partial T} + \mu \frac{\partial f}{\partial \mu} + 2 \vec{u}^2 \frac{\partial f}{\partial \vec{u}^2}\right) u^\mu - f v^\mu\right] \delta_{\mathscr{B}} n_\mu$$

$$+ \frac{\partial f}{\partial \mu} u^\mu \delta_{\mathscr{B}} A_\mu + \left(2 \frac{\partial f}{\partial \vec{u}^2} u^\mu u^\nu + f h^{\mu\nu}\right) \frac{1}{2} \delta_{\mathscr{B}} h_{\mu\nu} . \tag{25}$$

There are a few lessons to be learnt from the equations of motion (23). Firstly, note that the equations of motion can be written entirely in terms of $\delta_{\mathscr{B}} n_\mu$, $\delta_{\mathscr{B}} h_{\mu\nu}$, and $\delta_{\mathscr{B}} A_\mu$. Let us say that the background fields admit a timelike Killing vector $K^\mu$, i.e. $\$_K n_\mu = \$_K h_{\mu\nu} = \$_K A_\mu = 0$, where $\$_K$ denotes a Lie derivative along $K^\mu$. Coupled to such a background, the equations of motion admit a trivial "equilibrium solution" given by $u^\mu / T = K^\mu$ and $\mu / T = K^\mu A_\mu$. Secondly, we can always eliminate any $(d + 2)$ number of linear combinations of $\delta_{\mathscr{B}} n_\mu$, $\delta_{\mathscr{B}} h_{\mu\nu}$, and $\delta_{\mathscr{B}} A_\mu$ from the higher derivative corrections to the constitutive relations using equations of motion. This shall be useful later while writing down the set of independent one-derivative corrections to the hydrodynamic constitutive relations.

### 2.2.3 Hydrodynamic frame transformations

Recall that we had defined the hydrodynamic variables $u^i$, $T$, and $\mu$ using the thermodynamic relations (6). However, these definitions are only well posed in equilibrium. Out of equilibrium, there is no unique notion of fluid velocity, temperature, or chemical potential. This is important because we can always redefine these quantities with terms involving spacetime derivatives which will vanish in equilibrium, such as

$$T \to T + \delta T , \qquad \mu \to \mu + \delta\mu , \qquad u^i \to u^i + \delta u^i , \tag{26}$$

where $\delta T$, $\delta\mu$, $\delta u^i$ contain terms with at least one derivative. We can also define $\delta u^\mu$ as the change in the covariant fluid velocity, with $n_\mu \delta u^\mu = 0$. More details on the explicit action of these redefinitions on the hydrodynamic constitutive relations can be found in appendix A.

Often, it is convenient to work in a "hydrodynamic frame" where one imposes extra constraints on the derivative corrections that can enter the ideal order constitutive relations (21), so that this freedom is exactly fixed. A natural choice is to ensure that the conserved densities: energy, momentum, and particle number, do not obtain any corrections, by requiring

$$\epsilon^\mu n_\mu = \epsilon\,, \qquad \pi^\mu = \rho\,\vec{u}^\mu\,, \qquad j^\mu n_\mu = n\,, \tag{27}$$

which we call the *density frame*. This frame ties the fluid velocity with the flow of momentum. In the Galilean case, this is same as the "mass frame" with the fluid velocity aligned with the flow of mass. For relativistic hydrodynamics, there are other more popular hydrodynamic frames used in the literature, such as the Landau and Eckart frame, where the fluid velocity is aligned with internal energy and charge flow, respectively. However these frames are known to exhibit unphysical pathologies such as superluminal propagation and unstable modes in the linear spectrum in a finite velocity state [30, 33–36]. By contrast, the density frame defined above is always well-defined. We will return to these issues in section 6. More details about hydrodynamic frame transformations in boost-agnostic hydrodynamics can be found in appendix A. As it turns out, the most useful choice for us is to leave the hydrodynamic field redefinition freedom to be unfixed for now. We shall return to it in the next subsection.

## 2.3 Adiabaticity equation, thermodynamic frame, and discrete symmetries

We can write down a covariant version of the second law of thermodynamics given in eq. (5) as

$$\left(\nabla_\mu + F^n_{\mu\lambda} v^\lambda\right) s^\mu = \frac{1}{\sqrt{\gamma}} \partial_\mu \left(\sqrt{\gamma}\, s^\mu\right) = k_{\rm B} \Delta \geq 0\,. \tag{28}$$

Here $\Delta$ has to be a positive semi-definite quadratic form and $k_{\rm B}$ is the Boltzmann constant. The second law is imposed onshell, i.e. it is only required to be satisfied by configurations satisfying the conservation equations (18). Nonetheless, we can convert it into a offshell statement by adding arbitrary combinations of conservation equations. Introducing an arbitrary vector multiplier $\beta^\mu$ and a scalar one $\Lambda_\beta$, we can write [37, 38]

$$\begin{aligned}
\left(\nabla_\mu + F^n_{\mu\lambda} v^\lambda\right) s^\mu &- k_{\rm B}\beta^\rho n_\rho \left[\left(\nabla_\mu + F^n_{\mu\lambda} v^\lambda\right) \epsilon^\mu + \dots\right] \\
&+ k_{\rm B}\beta^\rho h_{\rho\nu} \left[\left(\nabla_\mu + F^n_{\mu\lambda} v^\lambda\right)(v^\mu \pi^\nu + \tau^{\mu\nu}) + \dots\right] \\
&+ k_{\rm B}\left(\Lambda_\beta + \beta^\rho A_\rho\right)\left(\nabla_\mu + F^n_{\mu\lambda} v^\lambda\right) j^\mu = k_{\rm B}\Delta \geq 0\,, \tag{29}
\end{aligned}$$

which will be satisfied offshell for some $\beta^\mu$ and $\Lambda_\beta$. It can be checked that the ideal fluid constitutive relations (21) satisfy this relation for $\Delta = \mathcal{O}(\partial^2)$, provided that we choose $k_{\rm B}\beta^\mu = u^\mu/T + \mathcal{O}(\partial)$ and $k_{\rm B}\Lambda_\beta = (\mu - u^\mu A_\mu)/T + \mathcal{O}(\partial)$.

Recall that we had an immense amount of redefinition freedom on our hands in the choice of hydrodynamic variables $u^\mu$, $T$, and $\mu$ that we left unfixed at the end of section 2.2.3. We can fix this freedom by requiring the multipliers $\beta^\mu$, $\Lambda_\beta$ to be exactly equal to their ideal order values with no derivative corrections

$$\beta^\mu = \frac{u^\mu}{k_{\rm B} T}\,, \qquad \Lambda_\beta = \frac{\mu - u^\mu A_\mu}{k_{\rm B} T}\,. \tag{30}$$

This is known as a *thermodynamic frame*. This, however, is not a complete fixing. We can imagine performing certain redefinitions of $u^\mu$, $T$, $\mu$, and by extension of $\beta^\mu$, $\Lambda_\beta$, that only

change the constitutive relations satisfying the adiabaticity equation (33) up to combinations of conservation equations. Such redefinitions still need to be accounted for as they leave the dynamics invariant. This can be unambiguously done following our discussion around eq. (23) and eliminating any $(d + 2)$ combinations among $\delta_{\mathscr{B}} A_\mu$, $\delta_{\mathscr{B}} n_\mu$, and $\delta_{\mathscr{B}} h_{\mu\nu}$ from the hydrodynamic data, leaving us with $d(d + 5)/2$ independent components. Note that $v^\mu v^\nu \delta_{\mathscr{B}} h_{\mu\nu}$ is trivially zero. Different choices lead to different thermodynamic frames. Of particular interest to us is the thermodynamic density frame, where we choose the independent data to be

$$h^{\mu\nu}\delta_{\mathscr{B}} A_\nu, \qquad h^{\mu\nu}\delta_{\mathscr{B}} n_\nu, \qquad h^{\mu\rho}h^{\nu\sigma}\delta_{\mathscr{B}} h_{\rho\sigma} . \tag{31}$$

This matches up with the density frame defined in eq. (27) in the non-hydrostatic sector (i.e. part of the constitutive relations that vanish in a hydrostatic/equilibrium configuration), but differ substantially in the hydrostatic sector. The discussion for thermodynamic Landau frame is presented in appendix A. In the core of this paper we will be working in the thermodynamic density frame. The main reason for this choice is that this frame, unlike the Landau or Eckart frames, does not exhibit unphysical instabilities in the linearised mode spectrum [30,33–36].[6] This fact will be clear when studying linearised fluctuations in section 6.

Eq. (29) can be transformed into a more useful form by defining the free energy current

$$N^\mu = \frac{1}{k_B}s^\mu - \beta^\nu n_\nu \epsilon^\mu + v^\mu \beta^\nu \pi_\nu + \beta^\lambda h_{\lambda\nu}\tau^{\mu\nu} + \left(\Lambda_\beta + \beta^\rho A_\rho\right)j^\mu , \tag{32}$$

which leads to the *adiabaticity equation*

$$\left(\nabla_\mu + F^n_{\mu\lambda}v^\lambda\right)N^\mu = -\epsilon^\mu \delta_{\mathscr{B}} n_\mu + \left(v^\mu \pi^\nu + \frac{1}{2}\tau^{\mu\nu}\right)\delta_{\mathscr{B}} h_{\mu\nu} + j^\mu \delta_{\mathscr{B}} A_\mu + \Delta , \qquad \Delta \geq 0 . \tag{33}$$

The operator $\delta_{\mathscr{B}}$ combines a Lie derivative $\pounds_\beta$ along $\beta^\mu$ and a gauge shift along $\Lambda_\beta$, i.e.

$$\delta_{\mathscr{B}} n_\mu = \pounds_\beta n_\mu , \qquad \delta_{\mathscr{B}} h_{\mu\nu} = \pounds_\beta h_{\mu\nu} , \qquad \delta_{\mathscr{B}} A_\mu = \pounds_\beta A_\mu + \partial_\mu \Lambda_\beta . \tag{34}$$

This form of the second law of thermodynamics is more useful to implement on a curved background. It can be checked that the constitutive relations (21) are the most general solution of the adiabaticity equation (33) at the leading derivative order with $\Delta = 0$. This also justifies their explicit form in the presence of background sources. Note that $\Delta$ being zero at this derivative order means that ideal fluids are non-dissipative, as we would physically expect.

Additional phenomenological requirements beyond Aristotelian symmetries, and the second law of thermodynamics, are also usually imposed on the hydrodynamic constitutive relations. Such is the case of discrete time-reversal (T), parity (P), and charge conjugation (C) symmetries. In particular, underlying microscopic theories are often taken to respect some kind of time-reversal symmetry, like T, PT, or CPT. Denoting the action of these symmetries by $\Theta$, in table 1 we provide the transformation properties of various quantities of interest under these symmetries. Such symmetries are responsible for imposing Onsager's conditions on the hydrodynamic correlation functions (see [7]) or for requiring the constitutive relations to be $\Theta$-invariant in equilibrium [39–41]. These discrete symmetries will be crucial when formulating the Schwinger-Keldysh EFT in the next section. This completes our brief review of Aristotelian hydrodynamics – the explicit one-derivative order corrections will be considered in section 4.

## 3 Effective field theory for boost-agnostic hydrodynamics

In this section, we discuss the Schwinger-Keldysh effective field theory for boost-agnostic hydrodynamics. Unlike the EFT for relativistic hydrodynamics developed over the last decade [8,

---

[6]Stability of various hydrodynamic frames in relativistic, Galilean, and Carrollian fluids was studied in [31].

Table 1: Action of parity, time-reversal, and charge conjugation on various quantities in classical hydrodynamics and effective field theory. In the next section we introduce Schwinger-Keldysh double copies of various quantities in the effective theory, with labels "1/2" or "$r/a$", which have the same transformation properties as their unlabelled counterparts.

|  | C | P | T | PT | CPT |
|---|---|---|---|---|---|
| $X^0, t, \tau$ | + | + | − | − | − |
| $X^i, x^i, \sigma^i$ | + | − | + | − | − |
| $\varphi$ | − | + | − | − | + |
| $u^i, \beta^i, \mathbb{\beta}^i$ | + | − | − | + | + |
| $T, \beta^t, \mathbb{\beta}^\tau$ | + | + | + | + | + |
| $\mu, \Lambda_\beta, \Lambda_{\mathbb{\beta}}$ | − | + | + | + | − |
| $\epsilon^t, n_t$ | + | + | + | + | + |
| $\epsilon^i, n_i$ | + | − | − | + | + |
| $\pi_i, v^i$ | + | − | − | + | + |
| $\tau^{ij}, h_{ij}$ | + | + | + | + | + |
| $j^t, b_t$ | − | + | + | + | − |
| $j^i, b_i$ | − | − | − | + | − |

10,42–50], the EFT description of boost-agnostic hydrodynamics needs to treat time and space directions on independent footing. A similar discussion for Galilean hydrodynamics appeared recently in [11], where the time and space directions were indeed treated independently, but nonetheless had to be tied down to respect the underlying Milne boost symmetry. As noted there, Milne boosts actually make things quite hard for an effective field theorist; to make this symmetry manifest, one needs to pass to a higher-dimensional "null-background" representation followed by a null reduction to obtain the final results. Since boost-agnostic hydrodynamics does not worry about boosts altogether, the ensuing EFT is formally simpler than its Galilean cousin. In fact, the following discussion is mostly a reproduction of section 5 of [11], but with the Milne boost symmetry revoked. The lack of a symmetry does mean that many more terms can now enter the effective action at a given derivative order that were previously not allowed, making the boost-agnostic case structurally more richer; we will see an example of this for one-derivative fluids in section 4.

### 3.1 Schwinger-Keldysh sigma model on the fluid worldvolume

In this section we introduce a worldvolume formulation of the EFT. The fluid worldvolume is a $(d+1)$-dimensional manifold endowed with coordinates $\sigma^\alpha$. These coordinates can be interpreted as labels associated with each fluid element in the physical spacetime. The dynamical fields living on the worldvolume are $X_s^\mu(\sigma)$ and $\varphi_s(\sigma)$ with $s = 1, 2$, and are Schwinger-Keldysh double copies of spacetime coordinates and U(1) phases of a given fluid element. We can decompose these fields in average combinations according to $X_{1,2}^\mu = X_r^\mu \pm \hbar/2 X_a^\mu$ and $\varphi_{1,2} = \varphi_r \pm \hbar/2 \varphi_a$. The combination $X_r^\mu(\sigma)$ denotes the physical spacetime coordinates and is akin to an embedding map, while $\varphi_r(\sigma)$ denotes the physical U(1) phase of the fluid elements. In turn, the average combinations $X_a^\mu(\sigma)$ and $\varphi_a(\sigma)$ encode the stochastic degrees of freedom. The worldvolume also contains two fixed reference fields, namely, a thermal vector field $\mathbb{\beta}^\alpha(\sigma)$ and a chemical shift field $\Lambda_{\mathbb{\beta}}(\sigma)$. These additional fields define the global rest frame and global chemical potential associated with states in global thermal equilibrium.

The EFT is required to be invariant under translations and rotations of the coordinates $X_s^\mu(\sigma)$ as well as under global U(1) shifts of the phases $\varphi_s(\sigma)$ acting independently on the two Schwinger-Keldysh spacetimes. In order to understand their action within the effective field theory, one must introduce double copies of Aristotelian sources as in section 2.2.1. In particular, associated with each Schwinger-Keldysh spacetime we have the clock forms $n_{s\mu}(X_s)$, degenerate spatial metrics $h_{s\mu\nu}(X_s)$ and gauge fields $A_{s\mu}(X_s)$. Thus, under local Schwinger-Keldysh spacetime diffeomorphisms and gauge transformations

$$X_s^\mu(\sigma) \to X_s'^\mu(X_s(\sigma)) \,, \qquad \varphi_s(\sigma) \to \varphi_s(\sigma) - \Lambda_s(X_s(\sigma)) \,, \tag{35}$$

the action on the background sources is given by eq. (16). It is useful to make the symmetries (35) manifest on the fluid worldvolume by defining pullbacks of the background sources (with an additional gauge transformation) such that

$$\mathbb{n}_{s\alpha}(\sigma) = n_{s\mu}(X_s(\sigma))\,\partial_\alpha X_s^\mu(\sigma) \,, \qquad \mathbb{h}_{s\alpha\beta}(\sigma) = h_{s\mu\nu}(X_s(\sigma))\,\partial_\alpha X_s^\mu(\sigma)\partial_\beta X_s^\nu(\sigma) \,,$$
$$\mathbb{A}_{s\alpha}(\sigma) = A_{s\mu}(X_s(\sigma))\,\partial_\alpha X_s^\mu(\sigma) + \partial_\alpha \varphi_s(\sigma) \,. \tag{36}$$

All the dependence on the dynamical and background fields in the effective theory must enter via these invariants to respect Schwinger-Keldysh spacetime symmetries. This fixes the structure of coupling between background and dynamical fields in the effective theory.

The EFT on the fluid worldvolume is required to be locally reparametrisation invariant and invariant under local shifts of the U(1) phases $\varphi_s(\sigma)$. In particular, under

$$\sigma^\alpha \to \sigma'^\alpha(\sigma) \,, \qquad \varphi_s(\sigma) \to \varphi_s(\sigma) + \lambda(\sigma) \,, \tag{37a}$$

in which the two phases shift simultaneously. The pullback of background sources,

$$\mathbb{n}_{s\alpha}(\sigma) \to \mathbb{n}'_{s\alpha}(\sigma') = \frac{\partial \sigma^\beta}{\partial \sigma'^\alpha}\mathbb{n}_{s\beta}(\sigma) \,, \qquad \mathbb{h}_{s\alpha\beta}(\sigma) \to \mathbb{h}'_{s\alpha\beta}(\sigma') = \frac{\partial \sigma^\gamma}{\partial \sigma'^\alpha}\frac{\partial \sigma^\delta}{\partial \sigma'^\beta}\mathbb{h}_{s\gamma\delta}(\sigma) \,,$$
$$\mathbb{A}_{s\alpha}(\sigma) \to \mathbb{A}'_{s\alpha}(\sigma') = \frac{\partial \sigma^\beta}{\partial \sigma'^\alpha}\left(\mathbb{A}_{s\beta}(\sigma) + \partial_\beta \lambda(\sigma)\right) \,, \tag{37b}$$

transform as tensors under such reparametrisations and phase shifts while the worldvolume fields $\beta^\alpha(\sigma)$ and $\Lambda_\beta(\sigma)$ transform in the expected manner, namely

$$\beta^\alpha(\sigma) \to \beta'^\alpha(\sigma') = \frac{\partial \sigma'^\alpha(\sigma)}{\partial \sigma^\beta}\beta^\beta(\sigma) \,, \qquad \Lambda_\beta(\sigma) \to \Lambda'_\beta(\sigma') = \Lambda_\beta(\sigma) - \beta^\alpha(\sigma)\partial_\alpha \lambda(\sigma) \,. \tag{37c}$$

Given the transformation properties (37), it is possible to build a gauge-invariant combination using the pullback of the gauge fields $\mathbb{A}_{a\alpha} = (\mathbb{A}_{1\alpha} - \mathbb{A}_{2\alpha})/\hbar$. On the other hand, the combination $\mathbb{A}_{r\alpha} = (\mathbb{A}_{1\alpha} + \mathbb{A}_{2\alpha})/2$ is not gauge-invariant. As such, when considering an effective action, $\mathbb{A}_{r\alpha}$ can only enter via the gauge-invariant combinations $\beta^\alpha \mathbb{A}_{r\alpha} + \Lambda_\beta$ and $2\partial_{[\alpha}\mathbb{A}_{r\beta]}$.

It is possible to partially fix the reparametrisation freedom by choosing a set of worldvolume coordinates $\sigma^\alpha = (\tau, \sigma^i)$ and setting $\beta^\alpha = \beta_0 \delta_\tau^\alpha$ as well as $\Lambda_\beta = \beta_0 \mu_0$. Here, $\beta_0 = (k_B T_0)^{-1}$ is the (constant) inverse temperature and $\mu_0$ the (constant) chemical potential of the global thermal state. Given these choices, we are left with residual spatial reparametrisation freedom $\tau \to \tau + f(\vec{\sigma})$ and $\sigma^i \to \sigma'^i(\vec{\sigma})$ as well as with U(1) phase shifts $\varphi_s \to \varphi_s + \lambda(\vec{\sigma})$ (see [11]).

The effective action $S$ for boost-agnostic hydrodynamics is the most generic functional made out of the background and dynamical fields, respecting the Schwinger-Keldysh spacetime symmetries (35) and the fluid worldvolume symmetries (37). Using the invariants $\Phi_s = (\mathbb{n}_{s\alpha}, \mathbb{h}_{s\alpha\beta}, \mathbb{A}_{s\alpha})$ and the reference thermal data $\mathbb{B} = (\beta^\alpha, \Lambda_\beta)$, the effective action can be written in terms of a Lagrangian density

$$S[\Phi_1, \Phi_2; \mathbb{B}] = \int d^{d+1}\sigma \sqrt{\mathbb{0}_r}\, \mathcal{L}[\Phi_1, \Phi_2; \mathbb{B}] \,, \tag{38}$$

where we have defined $\mho_r = \det(\mathbb{n}_{r\alpha}\mathbb{n}_{r\beta} + \mathbb{h}_{r\alpha\beta})$ with $\mathbb{n}_{r\alpha} = (\mathbb{n}_{1\alpha} + \mathbb{n}_{1\alpha})/2$ and $\mathbb{h}_{r\alpha\beta} = (\mathbb{h}_{1\alpha\beta} + \mathbb{h}_{1\alpha\beta})/2$. The Lagrangian $\mathcal{L}$ is a gauge-invariant scalar on the worldvolume. This form of the action makes all the spacetime and worldvolume symmetries of the effective theory manifest. However, the action is also required to obey a set of Schwinger-Keldysh requirements on the account of describing generic thermal field theories [8, 11]. These are

$$S^*[\Phi_1, \Phi_2; \mathbb{B}] = -S[\Phi_2, \Phi_1; \mathbb{B}] \,, \tag{39a}$$

$$S[\Phi, \Phi; \mathbb{B}] = 0 \,, \tag{39b}$$

$$\operatorname{Im} S[\Phi_1, \Phi_2; \mathbb{B}] \geq 0 \,, \tag{39c}$$

$$S[\Phi_1, \Phi_2; \mathbb{B}] = S[\tilde{\Phi}_1, \tilde{\Phi}_2; \tilde{\mathbb{B}}] \,, \tag{39d}$$

where the "tilde" KMS-conjugation in eq. (39d) is defined as

$$\tilde{\mathbb{n}}_{1\alpha}(\sigma) = \Theta\mathbb{h}_{1\alpha}(\sigma) \,, \quad \tilde{\mathbb{n}}_{2\alpha}(\sigma) = \Theta\mathbb{n}_{2\alpha}(\sigma) - i\hbar\Theta\$_{\mathbb{\beta}}\mathbb{n}_{2\alpha}(\sigma) + \mathcal{O}(\hbar) \,,$$

$$\tilde{\mathbb{h}}_{1\alpha\beta}(\sigma) = \Theta\mathbb{h}_{1\alpha\beta}(\sigma) \,, \quad \tilde{\mathbb{h}}_{2\alpha\beta}(\sigma) = \Theta\mathbb{h}_{2\alpha\beta}(\sigma) - i\hbar\Theta\$_{\mathbb{\beta}}\mathbb{h}_{2\alpha\beta}(\sigma) + \mathcal{O}(\hbar) \,,$$

$$\tilde{\mathbb{A}}_{1\alpha}(\sigma) = \Theta\mathbb{A}_{1\alpha}(\sigma) \,, \quad \tilde{\mathbb{A}}_{2\alpha}(\sigma) = \Theta\mathbb{A}_{2\alpha}(\sigma) - i\hbar\Theta\$_{\mathbb{\beta}}\mathbb{A}_{2\alpha}(\sigma) - i\hbar\Theta\partial_\alpha\Lambda_{\mathbb{\beta}}(\sigma) + \mathcal{O}(\hbar),$$

$$\tilde{\mathbb{\beta}}^\alpha(\sigma) = \Theta\mathbb{\beta}^\alpha(\sigma) \,, \qquad \tilde{\Lambda}_{\mathbb{\beta}}(\sigma) = \Theta\Lambda_{\mathbb{\beta}}(\sigma) \,. \tag{40}$$

Here $\Theta$ represents a discrete symmetry transformation involving a time-flip, e.g. T, PT, or CPT. Its action on various quantities is given in table 1. The operator $\$_{\mathbb{\beta}}$ denotes a Lie derivative along $\mathbb{\beta}^\alpha$. We can compactly denote these transformations as

$$\tilde{\Phi}_1 = \Theta\Phi_1 \,, \qquad \tilde{\Phi}_2 = \Theta\Phi_2 - i\hbar\Theta\delta_{\mathbb{B}}\Phi_2 + \mathcal{O}(\hbar) \,, \qquad \tilde{\mathbb{B}} = \Theta\mathbb{B} \,. \tag{41}$$

The operator $\delta_{\mathbb{B}}$ combines the Lie derivative $\$_{\mathbb{\beta}}$ along $\mathbb{\beta}^\alpha$ and a gauge shift along $\Lambda_{\mathbb{\beta}}$. Here we have focused on the KMS transformations in the statistical limit ($\hbar \to 0$); the finite $\hbar$ quantum versions are the same as those in the Galilean case [11].

## 3.2 Physical spacetime formulation

The effective theory on the fluid worldvolume can be rewritten on the physical spacetime. The average coordinates $X_r^\mu(\sigma)$ are interpreted as an embedding map, such that the location of the worldvolume in the physical spacetime is given by $x^\mu = X_r^\mu(\sigma)$. Inverting this map implies that the worldvolume coordinates $\sigma^\alpha = \sigma^\alpha(x)$ are seen as dynamical fields from the physical spacetime point of view. Analogously, we can express all other dynamical fields living on the worldvolume as functions of the physical spacetime coordinates, in particular, the U(1) phase $\varphi_r(x) = \varphi_r(\sigma(x))$ and the stochastic noise fields $X_a^\mu(x) = X_a^\mu(\sigma(x))$ and $\varphi_a(x) = \varphi_a(\sigma(x))$. It is useful to split the worldvolume sources (36) into average and difference combinations $\mathbb{n}_{1,2\,\alpha} = \mathbb{n}_{r\alpha} \pm \hbar/2\,\mathbb{n}_{a\alpha}$, $\mathbb{h}_{1,2\,\alpha\beta} = \mathbb{h}_{r\alpha\beta} \pm \hbar/2\,\mathbb{h}_{a\alpha\beta}$, and $\mathbb{A}_{1,2\,\alpha} = \mathbb{A}_{r\alpha} \pm \hbar/2\,\mathbb{A}_{a\alpha}$. Using these, one can define worldvolume gauge-invariant pushforwards onto the physical spacetime using the inverse map $\sigma^\alpha(x)$. In particular the average physical sources are given by[7]

$$N_{r\mu}(x) = \mathbb{n}_{r\alpha}(\sigma(x))\partial_\mu\sigma^\alpha(x) = n_{r\mu}(x) + \mathcal{O}(\hbar) \,,$$

$$H_{r\mu\nu}(x) = \mathbb{h}_{r\alpha\beta}(\sigma(x))\partial_\mu\sigma^\alpha(x)\partial_\nu\sigma^\beta(x) = h_{r\mu\nu}(x) + \mathcal{O}(\hbar) \,,$$

$$B_{r\mu}(x) = \mathbb{A}_{r\alpha}(\sigma(x))\partial_\mu\sigma^\alpha(x) - \partial_\mu\varphi_r(x) = A_{r\mu}(x) + \mathcal{O}(\hbar) \,, \tag{42a}$$

while the stochastic sources are defined as

$$N_{a\mu}(x) = \mathbb{n}_{a\alpha}(\sigma(x))\partial_\mu\sigma^\alpha(x) = n_{a\mu}(x) + \$_{X_a}n_{r\mu}(x) + \mathcal{O}(\hbar) \,,$$

$$H_{a\mu\nu}(x) = \mathbb{h}_{a\alpha\beta}(\sigma(x))\partial_\mu\sigma^\alpha(x)\partial_\nu\sigma^\beta(x) = h_{a\mu\nu}(x) + \$_{X_a}h_{r\mu\nu} + \mathcal{O}(\hbar) \,,$$

$$B_{a\mu}(x) = \mathbb{A}_{a\alpha}(\sigma(x))\partial_\mu\sigma^\alpha(x) = A_{a\mu}(x) + \partial_\mu\varphi_a(x) + \$_{X_a}A_{r\mu}(x) + \mathcal{O}(\hbar) \,. \tag{42b}$$

---

[7]Here $N_{r,a\mu}$ should not be confused with the free energy current $N^\mu$ in eq. (32).

In eq. (42), we have defined the Lie derivative along $X_a^\mu(x)$ as $\mathcal{L}_{X_a}$ and decomposed the background fields as well into average and difference combinations such that $n_{1,2\mu} = n_{r\mu} \pm \hbar/2 n_{a\mu}$, $h_{1,2\mu\nu} = h_{r\mu\nu} \pm \hbar/2 h_{a\mu\nu}$, and $A_{1,2\mu} = A_{r\mu} \pm \hbar/2 A_{a\mu}$ up to leading order in $\hbar$. These average background fields can be identified with the classical Aristotelian background fields of section 2.2.1. We can also identify a frame velocity $v_r^\mu$ and inverse spatial metric $h_r^{\mu\nu}$ on the physical spacetime as the averages $v_r^\mu = (v_1^\mu + v_2^\mu)/2$ and $h_r^{\mu\nu} = (h_1^{\mu\nu} + h_2^{\mu\nu})/2$. These satisfy the conditions (13) at leading order in $\hbar$.

Similarly, the hydrodynamic fields $\beta^\mu(x)$ and $\Lambda_\beta(x)$ are obtained by pushforward of $\mathbb{\beta}^\alpha(\sigma)$ and $\Lambda_{\mathbb{\beta}}(\sigma)$ such that[8]

$$\beta^\mu(x) = \mathbb{\beta}^\alpha(\sigma(x))\partial_\alpha X_r^\mu(\sigma(x)) \,, \qquad \Lambda_\beta(x) = \Lambda_{\mathbb{\beta}}(\sigma(x)) + \mathbb{\beta}^\alpha(\sigma(x))\partial_\alpha \varphi_r(\sigma(x)) \,. \tag{43}$$

Additionally, the classical hydrodynamic fields introduced in section 2, namely, the normalised fluid velocity $u^\mu(x)$ obeying $u^\mu N_{r\mu} = 1$, the local temperature $T(x)$, and the chemical potential $\mu(x)$ are defined as[9]

$$k_B T(x) = \frac{1}{\beta^\mu(x)N_{r\mu}(x)} \,, \qquad u^\mu(x) = \frac{\beta^\mu(x)}{\beta^\lambda(x)N_{r\lambda}(x)} \,, \qquad \mu(x) = \frac{\beta^\mu(x)B_{r\mu}(x) + \Lambda_\beta(x)}{\beta^\lambda(x)N_{r\lambda}(x)} \,, \tag{44}$$

which are gauge-invariant and do not depend on the stochastic fields.

It is necessary to make sure that the Schwinger-Keldysh spacetime symmetries (35) are correctly implemented in the physical spacetime. This can be done by requiring the resulting EFT to be invariant under "average" coordinate and gauge transformations using $\sigma(x)$ and $\varphi_r(x)$, i.e.

$$x^\mu \to x'^\mu(x) \,, \qquad \varphi_r(x) \to \varphi_r(x) - \Lambda(x) \,. \tag{45a}$$

Under such transformations the "average" background structures and hydrodynamic fields transform according to

$$N_{r\mu}(x) \to N'_{r\mu}(x') = \frac{\partial x^\nu}{\partial x'^\mu} N_{r\nu}(x) \,, \qquad H_{r\mu\nu}(x) \to H'_{r\mu\nu}(x') = \frac{\partial x^\rho}{\partial x'^\mu}\frac{\partial x^\sigma}{\partial x'^\nu} H_{r\rho\sigma}(x) \,,$$

$$B_{r\mu}(x) \to B'_{r\mu}(x') = \frac{\partial x^\nu}{\partial x'^\mu}\left(B_{r\nu}(x) + \partial_\nu \Lambda(x)\right) \,,$$

$$\beta^\mu(x) \to \beta'^\mu(x') = \frac{\partial x'^\mu(\sigma)}{\partial x^\nu}\mathbb{\beta}^\nu(x), \qquad \Lambda_\beta(x) \to \Lambda'_\beta(x') = \Lambda_\beta(x) - \beta^\mu(x)\partial_\mu \Lambda(x) \,, \tag{45b}$$

while the "difference" stochastic parts transform as

$$N_{a\mu\nu}(x) \to N'_{a\mu\nu}(x') = \frac{\partial x^\nu}{\partial x'^\mu} N_{a\nu}(x) \,, \qquad H_{a\mu\nu}(x) \to H'_{a\mu\nu}(x') = \frac{\partial x^\rho}{\partial x'^\mu}\frac{\partial x^\sigma}{\partial x'^\nu} H_{a\rho\sigma}(x) \,,$$

$$B_{a\mu}(x) \to B'_{a\mu}(x') = \frac{\partial x^\nu}{\partial x'^\mu} B_{a\nu}(x) \,. \tag{45c}$$

Let us introduce the compact notation $\Phi_{r,a} = (N_{r,a\mu}, H_{r,a\mu\nu}, B_{r,a\mu})$ and $\mathscr{B} = (\beta^\mu, \Lambda_\beta)$, which are essentially the physical spacetime versions of $\Phi_{1,2}$ and $\mathbb{B}$. In terms of these, the hydrodynamic effective action (38) can be rewritten in physical spacetime language leading to

$$S[\Phi_r, \Phi_a; \mathscr{B}] = \int d^{d+1}x \sqrt{\gamma_r}\, \mathcal{L}[\Phi_r, \Phi_a; \mathscr{B}] \,, \tag{46}$$

---

[8]If we pick the frame $\mathbb{\beta}^\alpha(\sigma) = \beta_0 \delta_\tau^\alpha$ and $\Lambda_{\mathbb{\beta}} = \beta_0 \mu_0$, one obtains the physical spacetime counterparts $\beta^\mu = \beta_0 \partial_\tau X_r^\mu$ and $\Lambda_\beta = \beta_0(\mu_0 + \partial_\tau \varphi_r)$.

[9]These definitions of the hydrodynamic fields are not boost invariant, since we are dealing with boost-agnostic hydrodynamics. The relation to Galilean and relativistic hydrodynamic fields is presented in section 5.

with $\gamma_r = \det(n_{r\mu}n_{r\nu} + h_{r\mu\nu})$. The action is manifestly invariant under worldvolume and physical spacetime symmetries. However, it does need to satisfy the Schwinger-Keldysh constraints

$$S^*[\Phi_r, \Phi_a; \mathscr{B}] = -S[\Phi_r, -\Phi_a; \mathscr{B}] \,, \tag{47a}$$

$$S[\Phi_r, \Phi_a = 0; \mathscr{B}] = 0 \,, \tag{47b}$$

$$\text{Im}\, S[\Phi_r, \Phi_a; \mathscr{B}] \geq 0 \,, \tag{47c}$$

$$S[\Phi_r, \Phi_a; \mathscr{B}] = S[\tilde{\Phi}_r, \tilde{\Phi}_a; \tilde{\mathscr{B}}] \,, \tag{47d}$$

where the KMS conjugation follows from

$$\tilde{\Phi}_r = \Theta \Phi_r + \mathcal{O}(\hbar) \,, \qquad \tilde{\Phi}_a = \Theta \Phi_a + i\Theta \delta_{\mathscr{B}} \Phi_r + \mathcal{O}(\hbar) \,, \qquad \tilde{\mathscr{B}} = \Theta \mathscr{B} + \mathcal{O}(\hbar) \,. \tag{48}$$

The operator $\delta_{\mathscr{B}}$ denotes a Lie derivative along $\beta^\mu$ combined with a gauge shift along $\Lambda_\beta$, i.e. $\delta_{\mathscr{B}} N_{r\mu} = \pounds_\beta N_{r\mu}$, $\delta_{\mathscr{B}} H_{r\mu\nu} = \pounds_\beta H_{r\mu\nu}$, and $\delta_{\mathscr{B}} B_{r\mu} = \pounds_\beta B_{r\mu} + \partial_\mu \Lambda_\beta$. As a reminder, $\Theta$ is a discrete symmetry transformation that the theory might enjoy such as T, PT, or CPT (see table 1). We can define the "$r/a$" variants of the hydrodynamic operators by varying the action with respect to "$a/r$" background fields according to

$$\delta S = \int \mathrm{d}^{d+1}x \sqrt{\gamma_r} \left[ \rho_r^\mu \delta A_{a\mu} - \epsilon_r^\mu \delta n_{a\mu} + \left( v_r^\mu \pi_r^\nu + \frac{1}{2}\tau_r^{\mu\nu} \right) \delta h_{a\mu\nu} \right.$$
$$\left. + \rho_a^\mu \delta A_{r\mu} - \epsilon_a^\mu \delta n_{r\mu} + \left( v_r^\mu \pi_a^\nu + \frac{1}{2}\tau_a^{\mu\nu} \right) \delta h_{r\mu\nu} \right]. \tag{49}$$

The "$r$" operators are understood as the physical hydrodynamic observables, while the "$a$" ones as the associated stochastic noise counterparts. Out-of-equilibrium thermal correlations functions of these operators can be computed by varying the Schwinger-Keldysh generating functional which takes the form

$$\exp W[\phi_r, \phi_a] = \int \mathcal{D}X_r \mathcal{D}X_a \mathcal{D}\varphi_r \mathcal{D}\varphi_a \exp\left( iS[\Phi_r, \Phi_a; \mathscr{B}] \right) \,, \tag{50}$$

where $\phi_{r,a} = (-n_{r,a\mu}, h_{r,a\mu\nu}, A_{r,a\mu})$.

### 3.3 Schwinger-Keldysh effective action

Based upon the considerations of the previous subsection, it is possible to find the explicit structure of the effective action entering in eq. (50). The procedure is directly analogous to that of Galilean fluids [11]. KMS conjugation acts on the building blocks of the effective action according to $\mathscr{B} \to \Theta \mathscr{B}$, $\Phi_r \to \Theta \Phi_r$, $\Phi_a \to \Theta \Phi_a + i\Theta \delta_{\mathscr{B}} \Phi_r$ for the hydrodynamic fields $\mathscr{B} = (\beta^\mu, \Lambda_\beta)$ and the invariants $\Phi_{r,a} = (N_{r,a\mu}, 1/2 H_{r,a\mu\nu}, B_{r,a\mu})$. Thus the most general effective action for hydrodynamics without boosts is given by a set of totally-symmetric multi-linear operators $\mathcal{D}_m(\circ, \dots)$ made out of $\Phi_r$ and $\mathscr{B}$, allowing $m$ number of arguments from the vector space spanned by $i\delta_{\mathscr{B}} \Phi_r$ and $\Phi_a$. In particular, the minimal Lagrangian for classical hydrodynamics is given by

$$\mathcal{L} = \mathcal{D}_1(\Phi_a) + i\mathcal{D}_2(\Phi_a, \Phi_a + i\delta_{\mathscr{B}}\Phi_r) + \mathcal{D}_3(\Phi_a + \tfrac{i}{2}\delta_{\mathscr{B}}\Phi_r, \Phi_a, \Phi_a + i\delta_{\mathscr{B}}\Phi_r) + \mathcal{O}(\hbar) \,, \tag{51}$$

where the $\mathcal{D}_m(\circ, \dots)$ operators satisfy the following constraints (see [11] for more details)

$$\mathcal{D}_1(\delta_{\mathscr{B}}\Phi_r) = \frac{1}{\sqrt{\gamma_r}} \partial_\mu \left( \sqrt{\gamma_r} \mathcal{N}_0^\mu \right) \quad \text{for some vector } \mathcal{N}_0^\mu \,, \tag{52a}$$

$$\mathcal{D}_1(\Phi_a), \quad \mathcal{D}_2(\Phi_a, \Phi_a), \quad \mathcal{D}_3(\Phi_a, \Phi_a, \Phi_a) \quad \text{are } \Theta\text{-even} \,, \tag{52b}$$

$$\mathcal{D}_2(\Phi, \Phi)\big|_{\text{leading order}} \geq 0 \quad \text{for arbitrary } \Phi = (N_\mu, 1/2 H_{\mu\nu}, B_\mu) \,. \tag{52c}$$

These constraints are consistent with the second law of thermodynamics (28); see [11] for a derivation. In the next section, we will provide the explicit form of the operators $\mathcal{D}_{1,2}$. As we are focusing in first order corrections, we do not provide the form of $\mathcal{D}_3$ as this operator contributes with second order and higher correction terms in the gradient expansion. Since in this work we are not interested in the most generic stochastic contributions, we have skipped the operators $\mathcal{D}_{n>3}$ in our discussion, as these do not contribute to the classical equations of hydrodynamics without boosts; see [23] for a detailed discussion.

## 4 One-derivative boost-agnostic hydrodynamics

In this section, we discuss charged boost-agnostic hydrodynamics up to first order in the derivative expansion. We start by writing down the most generic classical constitutive relations for the system allowed by the adiabaticity equation (second law of thermodynamics) discussed in section 2.3. We then proceed to write down the explicit effective action for one-derivative hydrodynamics utilising the machinery from section 3. We will briefly discuss how these are related to the more familiar constitutive relations of Galilean and relativistic hydrodynamics in section 5. As a simple application of these results, in section 6 we study fluctuations around an equilibrium state in boost agnostic hydrodynamics.

### 4.1 Classical constitutive relations

We want to work out the most generic constitutive relations that satisfy the adiabaticity equation (33), truncated at first order in the derivative expansion. Focusing on the parity-even sector, the solutions can be classified into three classes: (1) hydrostatic (hs; Class $H_S$) that survive in a hydrostatic configuration, i.e. when we set $\delta_{\mathscr{B}} n_\mu = \delta_{\mathscr{B}} h_{\mu\nu} = \delta_{\mathscr{B}} A_\mu = 0$; (2) non-hydrostatic non-dissipative (nhsnd; Class $\overline{\text{D}}$) that vanish in an hydrostatic configuration, but do not contribute to entropy production quadratic form $\Delta$; and finally (3) dissipative (diss; Class D) that also vanish in a hydrostatic configuration, but contribute non-trivially to entropy production.[10]

#### 4.1.1 Hydrostatic transport

The hydrostatic sector is completely characterised by the free-energy density

$$\mathcal{N} = p - F_0 v^\mu \partial_\mu \mu - F_1 v^\mu \partial_\mu T - F_2 v^\mu \partial_\mu \vec{u}^2 \ . \tag{53}$$

This is the most generic hydrostatic scalar that can be made out of the constituent fields at one-derivative order. Here $p(T, \mu, \vec{u}^2)$ and $F_{0,1,2}(T, \mu, \vec{u}^2)$ are arbitrary functions of zeroth order scalars. It is easy to check that

$$\frac{1}{\sqrt{\gamma}} \partial_\mu (\sqrt{\gamma} \mathcal{N} \beta^\mu) = \frac{1}{\sqrt{\gamma}} \delta_{\mathscr{B}} (\sqrt{\gamma} \mathcal{N})$$
$$= \mathcal{N} \left( v^\mu \delta_{\mathscr{B}} n_\mu + \frac{1}{2} h^{\mu\nu} \delta_{\mathscr{B}} h_{\mu\nu} \right) + \frac{\delta \mathcal{N}}{\delta A_\mu} \delta_{\mathscr{B}} A_\mu + \frac{\delta \mathcal{N}}{\delta n_\mu} \delta_{\mathscr{B}} n_\mu + \frac{\delta \mathcal{N}}{\delta h_{\mu\nu}} \delta_{\mathscr{B}} h_{\mu\nu}$$
$$- \frac{1}{\sqrt{\gamma}} \partial_\mu (\sqrt{\gamma} \Theta_{\mathcal{N}}^\mu) \ . \tag{54}$$

---

[10]In the parity-odd sector, there can be additional contributions coming from global anomalies (Class A) and transcendental anomalies (Class $H_V$). These contributions are entirely fixed up to a few constants. See e.g. [38] for a discussion.

The variational derivatives have been performed at constant $\beta^\mu = u^\mu/T$ and $\Lambda_\beta = (\mu - u^\mu A_\mu)/T$. Here $\Theta^\mu_\mathcal{N}$ denotes a total derivative term generated during the Euler-Lagrange procedure. Comparing this with eq. (33), we can read out

$$N^\mu_{\mathrm{hs}} = \mathcal{N}\beta^\mu + \Theta^\mu_\mathcal{N}\,, \quad \Delta_{\mathrm{hs}} = 0\,,$$

$$j^\mu_{\mathrm{hs}} = \frac{\delta \mathcal{N}}{\delta A_\mu}, \quad \epsilon^\mu_{\mathrm{hs}} = -\frac{\delta \mathcal{N}}{\delta n_\mu} - \mathcal{N}v^\mu\,, \quad \pi^\mu_{\mathrm{hs}} = h^\mu{}_\rho n_\sigma \frac{\delta \mathcal{N}}{\delta h_{\rho\sigma}}\,, \quad \tau^{\mu\nu}_{\mathrm{hs}} = 2h^\mu{}_\rho h^\nu{}_\sigma \frac{\delta \mathcal{N}}{\delta h_{\rho\sigma}} + \mathcal{N}h^{\mu\nu}\,.$$

$$(55)$$

Explicitly, we find

$$j^\mu_{\mathrm{hs}} = n\,u^\mu - \frac{\partial F_0}{\partial \mu}u^\mu v^\lambda \partial_\lambda \mu + \frac{1}{\sqrt{\gamma}}\partial_\lambda\left(\sqrt{\gamma}F_0 v^\lambda\right)u^\mu - \frac{\partial F_1}{\partial \mu}u^\mu v^\lambda \partial_\lambda T - \frac{\partial F_2}{\partial \mu}u^\mu v^\lambda \partial_\lambda \vec{u}^2 + \mathcal{O}(\partial^2)\,,$$

$$\epsilon^\mu_{\mathrm{hs}} = \epsilon\,u^\mu + p\,\vec{u}^\mu - w_{F_0}u^\mu v^\lambda \partial_\lambda \mu + \frac{\mu}{\sqrt{\gamma}}\partial_\lambda\left(\sqrt{\gamma}F_0 v^\lambda\right)u^\mu$$

$$- w_{F_1}u^\mu v^\lambda \partial_\lambda T + \frac{T}{\sqrt{\gamma}}\partial_\lambda\left(\sqrt{\gamma}F_1 v^\lambda\right)u^\mu - w_{F_2}u^\mu v^\lambda \partial_\lambda \vec{u}^2 + \frac{2\vec{u}^2}{\sqrt{\gamma}}\partial_\lambda\left(\sqrt{\gamma}F_2 v^\lambda\right)u^\mu + \mathcal{O}(\partial^2)\,,$$

$$\pi^\mu_{\mathrm{hs}} = \rho\,\vec{u}^\mu - \left(2\frac{\partial F_0}{\partial \vec{u}^2}\vec{u}^\mu v^\lambda - F_0 h^{\mu\lambda}\right)\partial_\lambda \mu - \left(2\frac{\partial F_1}{\partial \vec{u}^2}\vec{u}^\mu v^\lambda - F_1 h^{\mu\lambda}\right)\partial_\lambda T$$

$$- \left(2\frac{\partial F_2}{\partial \vec{u}^2}\vec{u}^\mu v^\lambda - F_2 h^{\mu\lambda}\right)\partial_\lambda \vec{u}^2 + \frac{2}{\sqrt{\gamma}}\partial_\lambda\left(\sqrt{\gamma}F_2 v^\lambda\right)\vec{u}^\mu + \mathcal{O}(\partial^2)\,,$$

$$\tau^{\mu\nu}_{\mathrm{hs}} = \rho\,\vec{u}^\mu \vec{u}^\nu + p\,h^{\mu\nu} - \left(2\frac{\partial F_0}{\partial \vec{u}^2}\vec{u}^\mu \vec{u}^\nu + F_0 h^{\mu\nu}\right)v^\lambda \partial_\lambda \mu - \left(2\frac{\partial F_1}{\partial \vec{u}^2}\vec{u}^\mu \vec{u}^\nu + F_1 h^{\mu\nu}\right)v^\lambda \partial_\lambda T$$

$$- \left(2\frac{\partial F_2}{\partial \vec{u}^2}\vec{u}^\mu \vec{u}^\nu + F_2 h^{\mu\nu}\right)v^\lambda \partial_\lambda \vec{u}^2 + \frac{2}{\sqrt{\gamma}}\partial_\lambda\left(\sqrt{\gamma}F_2 v^\lambda\right)\vec{u}^\mu \vec{u}^\nu + \mathcal{O}(\partial^2)\,, \quad (56)$$

and

$$\Theta^\mu_\mathcal{N} = \frac{1}{k_{\mathrm{B}}T}v^\mu\left(F_0 u^\lambda \partial_\lambda \mu + F_1 u^\lambda \partial_\lambda T + F_2 u^\lambda \partial_\lambda \vec{u}^2\right), \quad (57)$$

where we have used the identity $\delta_{\mathscr{B}}v^\mu = -v^\mu v^\nu \delta_{\mathscr{B}}n_\mu - h^{\mu\lambda}v^\nu \delta_{\mathscr{B}}h_{\lambda\nu}$ and defined

$$w_{F_i} = T\frac{\partial F_i}{\partial T} + \mu\frac{\partial F_i}{\partial \mu} + 2\vec{u}^2\frac{\partial F_i}{\partial \vec{u}^2}\,. \quad (58)$$

The readers can convince themselves that these are the most generic parity-preserving hydrostatic constitutive relations allowed by the adiabaticity equation. In the uncharged limit, that is, when $F_0 = 0$ and $F_{1,2} \equiv F_{1,2}(T,\vec{u}^2)$, the hydrostatic contributions (58) agree with those in [27].[11]

If the underlying microscopic theory respects a discrete $\Theta$ symmetry, such as T, PT, or CPT, this will need to be imposed on the free-energy density $\mathcal{N}$. For instance, for $\Theta = $ T or $\Theta = $ PT, all three one-derivative coefficients must vanish, i.e. $F_{0,1,2} = 0$. On the other hand for $\Theta = $ CPT, we can only state that $F_{0,1,2}$ must be odd functions of the chemical potential $\mu$. Note that if $j^\mu$ corresponds to the particle number current, it does not make sense to discuss CPT.

### 4.1.2  Non-hydrostatic non-dissipative transport

Next we have the non-hydrostatic non-dissipative transport made out of linear combinations of $\delta_{\mathscr{B}}A_\mu$, $\delta_{\mathscr{B}}n_\mu$, and $\delta_{\mathscr{B}}h_{\mu\nu}$ that satisfy eq. (33) with $\Delta_{\mathrm{nhsnd}} = 0$. Recall that due to our thermodynamic frame-fixing condition, all the dependence on these can only appear via their spatial

---

[11]The comparison requires using the identification (134) and flipping $v^\mu \to -v^\mu$ to match the conventions of [27].

components in eq. (31). Inspecting eq. (33), it immediately follows that $n_\mu j^\mu_{\mathrm{nhsnd}} = n_\mu \epsilon^\mu_{\mathrm{nhsnd}} = \pi^\mu_{\mathrm{nhsnd}} = 0$. For the remaining contributions, we find

$$
\begin{pmatrix} j^\mu_{\mathrm{nhsnd}} \\ \epsilon^\mu_{\mathrm{nhsnd}} \\ \tau^{\mu\nu}_{\mathrm{nhsnd}} \end{pmatrix} = -k_{\mathrm{B}} T \begin{pmatrix} 0 & \bar{D}^{\mu\rho}_{j\epsilon} & \bar{D}^{\mu(\rho\sigma)}_{j\tau} \\ -\bar{D}^{\rho\mu}_{j\epsilon} & 0 & \bar{D}^{\mu(\rho\sigma)}_{\epsilon\tau} \\ -\bar{D}^{\rho(\mu\nu)}_{j\tau} & -\bar{D}^{\rho(\mu\nu)}_{\epsilon\tau} & \bar{D}^{(\mu\nu)(\rho\sigma)}_{\tau\tau} \end{pmatrix} \begin{pmatrix} \delta_{\mathscr{B}} A_\rho \\ -\delta_{\mathscr{B}} n_\rho \\ \frac{1}{2}\delta_{\mathscr{B}} h_{\rho\sigma} \end{pmatrix} , \tag{59}
$$

where

$$
\begin{aligned}
\bar{D}^{\mu\rho}_{j\epsilon} &= \bar{\mathfrak{v}}_{01} P^{\mu\rho} + \bar{\mathfrak{s}}_{01} \hat{u}^\mu \hat{u}^\rho , \\
\bar{D}^{\mu(\rho\sigma)}_{j\tau} &= 2\bar{\mathfrak{v}}_{02} P^{\mu(\rho} \hat{u}^{\sigma)} + \bar{\mathfrak{s}}_{02} \hat{u}^\mu \hat{u}^\rho \hat{u}^\sigma + \bar{\mathfrak{s}}_{03} \hat{u}^\mu P^{\rho\sigma} , \\
\bar{D}^{\mu(\rho\sigma)}_{\epsilon\tau} &= 2\bar{\mathfrak{v}}_{12} P^{\mu(\rho} \hat{u}^{\sigma)} + \bar{\mathfrak{s}}_{12} \hat{u}^\mu \hat{u}^\rho \hat{u}^\sigma + \bar{\mathfrak{s}}_{13} \hat{u}^\mu P^{\rho\sigma} , \\
\bar{D}^{(\mu\nu)(\rho\sigma)}_{\tau\tau} &= \bar{\mathfrak{s}}_{23} \left( \hat{u}^\mu \hat{u}^\nu P^{\rho\sigma} - P^{\mu\nu} \hat{u}^\rho \hat{u}^\sigma \right) .
\end{aligned} \tag{60}
$$

Here $\hat{u}^\mu = \vec{u}^\mu/|\vec{u}|$ and $P^{\mu\nu} = h^{\mu\nu} - \hat{u}^\mu \hat{u}^\nu$, with $|\vec{u}| = \sqrt{\vec{u}^2}$. The transport coefficients appearing here are arbitrary functions of $T$, $\mu$, and $\vec{u}^2$. The coefficient matrix in eq. (59) is antisymmetric, which ensures that there is no contribution to $\Delta_{\mathrm{nhsnd}}$.

As for the discrete symmetries $\Theta$, Onsager's relations require that the constitutive relations in eq. (59) are even under $\Theta$. With $\Theta = \mathrm{T}$ or $\Theta = \mathrm{PT}$, the entire non-hydrostatic non-dissipative sector is set to zero. On the other hand for $\Theta = \mathrm{CPT}$, all the transport coefficients appearing here must be odd functions of $\mu$.

### 4.1.3 Dissipative transport

The dissipative sector is quite similar to the non-hydrostatic non-dissipative sector, but with the coefficient matrix being symmetric, leading to entropy production. We again find that $n_\mu j^\mu_{\mathrm{diss}} = n_\mu \epsilon^\mu_{\mathrm{diss}} = \pi^\mu_{\mathrm{diss}} = 0$, along with

$$
\begin{pmatrix} j^\mu_{\mathrm{diss}} \\ \epsilon^\mu_{\mathrm{diss}} \\ \tau^{\mu\nu}_{\mathrm{diss}} \end{pmatrix} = -k_{\mathrm{B}} T \begin{pmatrix} D^{\mu\rho}_{jj} & D^{\mu\rho}_{j\epsilon} & D^{\mu(\rho\sigma)}_{j\tau} \\ D^{\rho\mu}_{j\epsilon} & D^{\mu\rho}_{\epsilon\epsilon} & D^{\mu(\rho\sigma)}_{\epsilon\tau} \\ D^{\rho(\mu\nu)}_{j\tau} & D^{\rho(\mu\nu)}_{\epsilon\tau} & D^{(\mu\nu)(\rho\sigma)}_{\tau\tau} \end{pmatrix} \begin{pmatrix} \delta_{\mathscr{B}} A_\rho \\ -\delta_{\mathscr{B}} n_\rho \\ \frac{1}{2}\delta_{\mathscr{B}} h_{\rho\sigma} \end{pmatrix} , \tag{61}
$$

with

$$
\begin{aligned}
D^{\mu\rho}_{jj} &= \mathfrak{v}_{00} P^{\mu\rho} + \mathfrak{s}_{00} \hat{u}^\mu \hat{u}^\rho , \\
D^{\mu\rho}_{j\epsilon} &= \mathfrak{v}_{01} P^{\mu\rho} + \mathfrak{s}_{01} \hat{u}^\mu \hat{u}^\rho , \\
D^{\mu\rho}_{\epsilon\epsilon} &= \mathfrak{v}_{11} P^{\mu\rho} + \mathfrak{s}_{11} \hat{u}^\mu \hat{u}^\rho , \\
D^{\mu(\rho\sigma)}_{j\tau} &= 2\mathfrak{v}_{02} P^{\mu(\rho} \hat{u}^{\sigma)} + \mathfrak{s}_{02} \hat{u}^\mu \hat{u}^\rho \hat{u}^\sigma + \mathfrak{s}_{03} \hat{u}^\mu P^{\rho\sigma} , \\
D^{\mu(\rho\sigma)}_{\epsilon\tau} &= 2\mathfrak{v}_{12} P^{\mu(\rho} \hat{u}^{\sigma)} + \mathfrak{s}_{12} \hat{u}^\mu \hat{u}^\rho \hat{u}^\sigma + \mathfrak{s}_{13} \hat{u}^\mu P^{\rho\sigma} , \\
D^{(\mu\nu)(\rho\sigma)}_{\tau\tau} &= 2\mathfrak{t} \left( P^{\rho(\mu} P^{\nu)\sigma} - \tfrac{1}{d-1} P^{\mu\nu} P^{\rho\sigma} \right) + 4\mathfrak{v}_{22} \hat{u}^{(\mu} P^{\nu)(\rho} \hat{u}^{\sigma)} \\
&\quad + \mathfrak{s}_{22} \hat{u}^\mu \hat{u}^\nu \hat{u}^\rho \hat{u}^\sigma + \mathfrak{s}_{23} \left( P^{\mu\nu} \hat{u}^\rho \hat{u}^\sigma + \hat{u}^\mu \hat{u}^\nu P^{\rho\sigma} \right) + \mathfrak{s}_{33} P^{\mu\nu} P^{\rho\sigma} .
\end{aligned} \tag{62}
$$

The associated entropy production quadratic form is given by

$$
\Delta_{\mathrm{diss}} = k_{\mathrm{B}} T \begin{pmatrix} \delta_{\mathscr{B}} A_\rho \\ -\delta_{\mathscr{B}} n_\rho \\ \frac{1}{2}\delta_{\mathscr{B}} h_{\rho\sigma} \end{pmatrix}^{\mathrm{T}} \begin{pmatrix} D^{\mu\rho}_{nn} & D^{\mu\rho}_{n\epsilon} & D^{\mu(\rho\sigma)}_{n\pi} \\ D^{\rho\mu}_{n\epsilon} & D^{\mu\rho}_{\epsilon\epsilon} & D^{\mu(\rho\sigma)}_{\epsilon\pi} \\ D^{\rho(\mu\nu)}_{n\pi} & D^{\rho(\mu\nu)}_{\epsilon\pi} & D^{(\mu\nu)(\rho\sigma)}_{\pi\pi} \end{pmatrix} \begin{pmatrix} \delta_{\mathscr{B}} A_\rho \\ -\delta_{\mathscr{B}} n_\rho \\ \frac{1}{2}\delta_{\mathscr{B}} h_{\rho\sigma} \end{pmatrix} \geq 0 . \tag{63}
$$

Imposing the positivity constraint merely requires that the dissipative coefficient matrix is positive semi-definite. Explicitly, this leads to

$$
\begin{pmatrix}
\mathfrak{s}_{00} & \mathfrak{s}_{01} & \mathfrak{s}_{02} & \mathfrak{s}_{03} \\
\mathfrak{s}_{01} & \mathfrak{s}_{11} & \mathfrak{s}_{12} & \mathfrak{s}_{13} \\
\mathfrak{s}_{02} & \mathfrak{s}_{12} & \mathfrak{s}_{22} & \mathfrak{s}_{23} \\
\mathfrak{s}_{03} & \mathfrak{s}_{13} & \mathfrak{s}_{23} & \mathfrak{s}_{33}
\end{pmatrix} \geq 0, \qquad
\begin{pmatrix}
\mathfrak{v}_{00} & \mathfrak{v}_{01} & \mathfrak{v}_{02} \\
\mathfrak{v}_{01} & \mathfrak{v}_{11} & \mathfrak{v}_{12} \\
\mathfrak{v}_{02} & \mathfrak{v}_{12} & \mathfrak{v}_{22}
\end{pmatrix} \geq 0, \qquad \mathfrak{t} \geq 0,
\tag{64}
$$

where the positive semi-definiteness of a matrix means that all its eigenvalues are positive semi-definite.

The discrete symmetry $\Theta$ requirements here work opposite to the non-hydrostatic non-dissipative sector. With $\Theta = T$ or $\Theta = PT$ symmetry, the dissipative sector transport coefficients are left invariant, while with $\Theta = CPT$ symmetry, all the dissipative transport coefficients must be even functions of $\mu$.

Therefore, we have a total of 30 coefficients at one-derivative order: 4 hydrostatic (including the ideal order pressure $p$), 9 non-hydrostatic non-dissipative, and 17 dissipative transport coefficients. We note that this counting differs from that of [29].[12] In the (uncharged) limit in which the U(1) current is removed, we have that

$$
F_0 = \bar{\mathfrak{v}}_{01} = \bar{\mathfrak{s}}_{01} = \bar{\mathfrak{v}}_{02} = \bar{\mathfrak{s}}_{02} = \bar{\mathfrak{s}}_{03} = \mathfrak{v}_{00} = \mathfrak{s}_{00} = \mathfrak{v}_{01} = \mathfrak{s}_{01} = \mathfrak{s}_{02} = \mathfrak{v}_{02} = \mathfrak{v}_{03} = 0 .
\tag{65}
$$

This amounts to a total of 17 transport coefficients: 3 hydrostatic (including ideal order pressure), 4 non-hydrostatic non-dissipative, and 10 dissipative transport coefficients, agreeing with the counting of [27] when focusing in special case in which the additional $U(1)$ is not present. A precise comparison is given in appendix A.2.

### 4.1.4 Entropy current

The free-energy current associated with the one-derivative order constitutive relations above is simply given as

$$
N^\mu = \mathcal{N}\beta^\mu + \Theta^\mu_{\mathcal{N}}.
\tag{66}
$$

Note that there is no contribution to $N^\mu$ due to dissipative and non-dissipative non-hydrostatic constitutive relations. Correspondingly, the entropy current is given as

$$
\begin{aligned}
s^\mu &= \mathcal{N}\frac{u^\mu}{T} + k_{\mathrm{B}}\Theta^\mu_{\mathcal{N}} + \frac{u^\nu}{T}n_\nu\epsilon^\mu - \nu^\mu\frac{u^\nu}{T}\pi_\nu - \frac{u^\lambda}{T}h_{\lambda\nu}\tau^{\mu\nu} - \frac{\mu}{T}j^\mu \\
&= s^\mu_{\mathrm{can}} + s^\mu_{\mathrm{non\text{-}can}} ,
\end{aligned}
\tag{67}
$$

where

$$
s^\mu_{\mathrm{can}} = \frac{p}{T}u^\mu + \frac{u^\nu}{T}n_\nu\epsilon^\mu - \nu^\mu\frac{u^\nu}{T}\pi_\nu - \frac{u^\lambda}{T}h_{\lambda\nu}\tau^{\mu\nu} - \frac{\mu}{T}j^\mu ,
\tag{68}
$$

is known as the canonical entropy current, while

$$
\begin{aligned}
s^\mu_{\mathrm{non\text{-}can}} &= (\mathcal{N}-p)\frac{u^\mu}{T} + k_{\mathrm{B}}\Theta^\mu_{\mathcal{N}} \\
&= \frac{\nu^\mu}{T}\left(F_0\vec{u}^\lambda\partial_\lambda\mu + F_1\vec{u}^\lambda\partial_\lambda T + F_2\vec{u}^\lambda\partial_\lambda\vec{u}^2\right) \\
&\quad - \frac{\vec{u}^\mu}{T}\left(F_0\nu^\mu\partial_\mu\mu + F_1\nu^\mu\partial_\mu T + F_2\nu^\mu\partial_\mu\vec{u}^2\right) ,
\end{aligned}
\tag{69}
$$

is known as the non-canonical part of the entropy current.

---

[12]The authors in [29] counted 9 non-dissipative transport coefficients and 20 dissipative transport coefficients. Further discussion on this can be found in appendix A.2.

## 4.2  Explicit effective action

We implement the following derivative counting in the EFT. The hydrodynamic fields $\beta^\mu$, $\Lambda_\beta$ and the noise fields $X_a^\mu$, $\varphi_a$ are taken to be $\mathcal{O}(\partial^0)$. On the other hand, the average background fields $n_{r\mu}$, $h_{r\mu\nu}$, $A_{r\mu}$ are treated at $\mathcal{O}(\partial^0)$, while their difference partners $n_{a\mu}$, $h_{a\mu\nu}$, $A_{a\mu}$ at $\mathcal{O}(\partial^1)$. In compact notation, this means that $\Phi_r$, $\mathcal{B}$ are $\mathcal{O}(\partial^0)$, while $\Phi_a$, $\delta_{\mathcal{B}}\Phi_r$ are $\mathcal{O}(\partial^1)$. Note that the classical constitutive relations are given by a variational derivative of the effective action with respect to the "$a$" type background fields. Therefore, the derivative order of the effective action must be one more than that of the constitutive relations. It follows that one-derivative hydrodynamics is entirely characterised by the operators $\mathcal{D}_1(\circ)$ truncated at one-derivative order and $\mathcal{D}_2(\circ,\circ)$ truncated at zeroth order; $\mathcal{D}_3(\circ,\circ,\circ)$ only contributes to two-derivative constitutive relations and higher.

Due to eq. (52) we know that $\mathcal{D}_1(\circ)$ is the most generic operator such that $\mathcal{D}_1(\delta_{\mathcal{B}}\Phi_r)$ is a total derivative. Recalling that in the statistical limit $\Phi_r = (-n_{r\mu}, 1/2\, h_{r\mu\nu}, A_{r\mu}) + \mathcal{O}(\hbar)$, we see that this requirement is precisely the adiabaticity equation. The most generic solution is therefore characterised by the adiabatic (hydrostatic + non-hydrostatic non-dissipative) constitutive relations. Explicitly

$$\mathcal{D}_1(\Phi_a) = j^\mu_{\mathrm{hs+nhsnd}} B_{a\mu} - \epsilon^\mu_{\mathrm{hs+nhsnd}} N_{a\mu} + \left( \nu^\mu \pi^\nu_{\mathrm{hs}} + \frac{1}{2} \tau^{\mu\nu}_{\mathrm{hs+nhsnd}} \right) H_{a\mu\nu} \, . \tag{70}$$

On the other hand, $\mathcal{D}_2(\circ,\circ)$ needs to be most generic symmetric positive semi-definite bilinear operator. The contribution from the same to the effective action is given as

$$i\mathcal{D}_2(\Phi_a, \Phi_a + i\delta_{\mathcal{B}}\Phi_r) = ik_{\mathrm{B}} T \begin{pmatrix} B_{a\mu} \\ -N_{a\mu} \\ \frac{1}{2} H_{a\mu\nu} \end{pmatrix}^{\mathrm{T}} \begin{pmatrix} D^{\mu\rho}_{jj} & D^{\mu\rho}_{j\epsilon} & D^{\mu(\rho\sigma)}_{j\tau} \\ D^{\rho\mu}_{j\epsilon} & D^{\mu\rho}_{\epsilon\epsilon} & D^{\mu(\rho\sigma)}_{\epsilon\tau} \\ D^{\rho(\mu\nu)}_{j\tau} & D^{\rho(\mu\nu)}_{\epsilon\tau} & D^{(\mu\nu)(\rho\sigma)}_{\tau\tau} \end{pmatrix} \begin{pmatrix} B_{a\rho} + i\delta_{\mathcal{B}}A_{r\rho} \\ -N_{a\rho} - i\delta_{\mathcal{B}}n_{r\rho} \\ \frac{1}{2}H_{a\rho\sigma} + \frac{i}{2}\delta_{\mathcal{B}}h_{r\rho\sigma} \end{pmatrix}. \tag{71}$$

The $\Theta$-requirements on various constitutive relations discussed in the previous subsection follow from here by demanding that both $\mathcal{D}_1$ and $\mathcal{D}_2$ operators are $\Theta$-even, in accordance with the Schwinger-Keldysh requirements in eq. (52).

In flat spacetime, with $\Theta = \mathrm{T}$ or $\Theta = \mathrm{PT}$ discrete symmetry (that leads to vanishing of hydrostatic and non-hydrostatic non-dissipative sectors), the effective Lagrangian for one-derivative order boost-agnostic hydrodynamics takes the form

$$\mathcal{L} = n\, \partial_t \varphi_a + n\, u^i \partial_i \varphi_a - \epsilon\, \partial_t X_a^t - (\epsilon + p) u^i \partial_i X_a^t + \rho\, u^i \partial_t X_{ai} + \left( \rho\, u^j u^i + p\, \delta^{ji} \right) \partial_j X_{ai}$$

$$+ ik_{\mathrm{B}} T \begin{pmatrix} \partial_i \varphi_a \\ -\partial_i X_a^t \\ \partial_i X_{aj} \end{pmatrix}^{\mathrm{T}} \begin{pmatrix} D^{ik}_{jj} & D^{ik}_{j\epsilon} & D^{i(kl)}_{j\tau} \\ D^{ki}_{j\epsilon} & D^{ik}_{\epsilon\epsilon} & D^{i(kl)}_{\epsilon\tau} \\ D^{k(ij)}_{j\tau} & D^{k(ij)}_{\epsilon\tau} & D^{(ij)(kl)}_{\tau\tau} \end{pmatrix} \begin{pmatrix} \partial_k \varphi_a + \frac{i}{k_{\mathrm{B}}} \partial_k \frac{\mu}{T} \\ -\partial_k X_a^t - \frac{i}{k_{\mathrm{B}}} \partial_k \frac{1}{T} \\ \partial_k X_{al} + \frac{i}{k_{\mathrm{B}}} \partial_k \frac{u_l}{T} \end{pmatrix}. \tag{72}$$

We will use these considerations in section 6 to study linearised fluctuations.

## 5  Special limits

The spectrum of transport coefficients for a boost-agnostic fluid is quite rich. We have a total of 30 coefficients at one-derivative order. For a clearer picture of these coefficients, it is helpful to make contact with the special cases of fluids respecting Galilean or Lorentz boost symmetries. In both these instances, the spectrum only contains the thermodynamic pressure $p$ in the hydrostatic sector and 3 dissipative transport coefficients: shear viscosity $\eta$, bulk viscosity $\zeta$, and thermal/electric conductivity $\kappa/\sigma$. In particular, there are no allowed one-derivative terms in

the hydrostatic or non-hydrostatic non-dissipative sectors. We will also discuss fluids respecting anisotropic Lifshitz scaling symmetry, in which case the transport coefficients reduce to: 3 hydrostatic (including pressure $p$), 6 non-hydrostatic non-dissipative, and 13 dissipative. The temperature dependence of all these 22 coefficients is fixed by the scaling symmetry.

## 5.1 Galilean fluids

Galilean (Milne) boost symmetry acts on the background fields according to (see, e.g. [11])

$$n_\mu \to n_\mu, \qquad h_{\mu\nu} \to h_{\mu\nu} - 2n_{(\mu}\psi_{\nu)} + n_\mu n_\nu \psi^2, \qquad A_\mu \to A_\mu + m\left(\psi_\mu - \frac{1}{2}n_\mu\psi^2\right),$$
$$v^\mu \to v^\mu + \psi^\mu, \qquad h^{\mu\nu} \to h^{\mu\nu}, \tag{73}$$

for arbitrary parameters $\psi^\mu$ satisfying $\psi^\mu n_\mu = 0$, with the definitions $\psi_\mu = h_{\mu\nu}\psi^\nu$, and $\psi^2 = h_{\mu\nu}\psi^\mu\psi^\nu$. Here $m$ is an arbitrary constant signifying mass per unit charge/particle. These are essentially the rules governing how the background sources must change when we move to a different frame of reference given by $v^\mu \to v^\mu + \psi^\mu$. Using the (equilibrium) effective action variation in eq. (15), this leads to the Milne boost Ward identity

$$\pi_\mu = m h_{\mu\nu} j^\nu. \tag{74}$$

The hydrodynamic fields in the representation $\beta^\mu$, $\Lambda_\beta$ correspond to the local thermodynamic frame and hence do not transform under boosts. However $u^\mu$, $T$, $\mu$ fields can potentially transform, which can be derived using their definitions in eq. (30). We find

$$u^\mu \to u^\mu, \qquad T \to T, \qquad \mu \to \mu + m\vec{u}^\mu\psi_\mu - \frac{m}{2}\psi^2. \tag{75}$$

It is convenient to define the boost-invariant Galilean chemical potential $\mu_{\text{gal}} = \mu + m/2\,\vec{u}^2$. All the transport coefficients in a Galilean fluid are functions of $T$ and $\mu_{\text{gal}}$. The equation of state is given as $p(T, \mu, \vec{u}^2) = p(T, \mu_{\text{gal}})$. Using the thermodynamic relations (10), one then derives

$$\mathrm{d}p = s\,\mathrm{d}T + n\,\mathrm{d}\mu_{\text{gal}}, \qquad \epsilon_{\text{gal}} = Ts + \mu_{\text{gal}}n - p, \qquad \rho = mn, \tag{76}$$

where we have identified the Galilean internal energy density $\epsilon_{\text{gal}} = \epsilon - 1/2\rho\,\vec{u}^2$. The relation $\rho = mn$ can be understood as the "Galilean equation of state".

Constitutive relations for Galilean hydrodynamics in curved space-time are already known; see for instance [11] and references therein. Their translation to the boost-agnostic representation discussed in this paper is quite straightforward because the language and hydrodynamic frame that we have employed for boost-agnostic fluids is quite similar to the one used for Galilean hydrodynamics. The complete set of one-derivative order constitutive relations are given as

$$j^\mu = \frac{\rho}{m}u^\mu,$$
$$\epsilon^\mu = \left(\epsilon_{\text{gal}} + \frac{1}{2}\rho\vec{u}^2\right)u^\mu + p\,\vec{u}^\mu + T^2\kappa\,h^{\mu\nu}\delta_{\mathcal{B}}n_\nu$$
$$\qquad - \left(2\eta h^{\rho(\mu}h^{\nu)\sigma} + \left(\zeta - \frac{2}{d}\eta\right)h^{\mu\nu}h^{\rho\sigma}\right)\vec{u}_\nu\frac{1}{2}\left(\delta_{\mathcal{B}}h_{\rho\sigma} - 2\vec{u}_{(\rho}\delta_{\mathcal{B}}n_{\sigma)}\right),$$
$$\pi^\mu = \rho\,\vec{u}^\mu,$$
$$\tau^{\mu\nu} = \rho\,\vec{u}^\mu\vec{u}^\nu - \left(2\eta h^{\rho(\mu}h^{\nu)\sigma} + \left(\zeta - \frac{2}{d}\eta\right)h^{\mu\nu}h^{\rho\sigma}\right)\vec{u}_\nu\frac{1}{2}\left(\delta_{\mathcal{B}}h_{\rho\sigma} - 2\vec{u}_{(\rho}\delta_{\mathcal{B}}n_{\sigma)}\right). \tag{77}$$

Note that the constitutive relations are already in the density frame (27). The mapping of ideal order thermodynamic coefficients is already given above. At one-derivative order, we find

$$F_0 = F_1 = F_2 = 0 \,,$$

$$\bar{\mathfrak{s}}_{01} = \bar{\mathfrak{s}}_{02} = \bar{\mathfrak{s}}_{03} = \bar{\mathfrak{s}}_{12} = \bar{\mathfrak{s}}_{13} = \bar{\mathfrak{s}}_{23} = \bar{\mathfrak{v}}_{01} = \bar{\mathfrak{v}}_{02} = \bar{\mathfrak{v}}_{12} = 0 \,,$$

$$\mathfrak{s}_{00} = \mathfrak{s}_{01} = \mathfrak{s}_{02} = \mathfrak{s}_{03} = \mathfrak{v}_{00} = \mathfrak{v}_{01} = \mathfrak{v}_{02} = 0 \,,$$

$$\mathfrak{s}_{22} = \frac{\mathfrak{s}_{12}}{|\vec{u}|} = \zeta + 2\frac{d-1}{d}\eta, \qquad \mathfrak{s}_{23} = \frac{\mathfrak{s}_{13}}{|\vec{u}|} = \zeta - \frac{2}{d}\eta \,, \qquad \mathfrak{s}_{33} = \zeta + \frac{2}{d(d-1)}\eta \,,$$

$$\mathfrak{s}_{11} = T\kappa + u^2\left(\zeta + 2\frac{d-1}{d}\eta\right) \,, \qquad \mathfrak{v}_{11} = T\kappa + u^2\eta \,, \qquad \mathfrak{v}_{22} = \frac{\mathfrak{v}_{12}}{|\vec{u}|} = \mathfrak{t} = \eta \,. \tag{78}$$

We see that the hydrostatic and non-hydrostatic non-dissipative sectors are entirely absent. The coefficients appearing in the charge/mass current in the dissipative sector are also absent due to the Milne boost Ward identity (74). The remaining 10 non-zero dissipative coefficients are determined in terms of $\eta$, $\zeta$, and $\kappa$.

## 5.2 Relativistic fluids

Our generic discussion of boost-agnostic fluids is also capable of handling relativistic hydrodynamics. However, the discussion is considerably more involved than the Galilean case owing to the inherent non-linearity of relativistic hydrodynamics. Lorentz boost symmetry can be imposed by requiring the theory to be invariant under the transformation of background sources[13]

$$n_\mu \to n_\mu - \frac{1}{c^2}\left(\psi_\mu - \frac{1}{2}n_\mu\psi^2\right), \qquad h_{\mu\nu} \to h_{\mu\nu} - 2n_{(\mu}\psi_{\nu)} + n_\mu n_\nu\psi^2, \qquad A_\mu \to A_\mu,$$

$$v^\mu \to v^\mu + \psi^\mu, \qquad h^{\mu\nu} \to h^{\mu\nu} + \frac{1}{c^2}\left(2v^{(\mu}\psi^{\nu)} + \psi^\mu\psi^\nu\right). \tag{79}$$

When implemented on the action variation (15) at the linear level, this imposes the center-of-energy conservation by setting

$$\pi_\mu = \frac{1}{c^2}h_{\mu\nu}\epsilon^\nu \,. \tag{80}$$

The relativistic metric sources can be defined as

$$g_{\mu\nu} = -c^2 n_\mu n_\nu + h_{\mu\nu} \,, \qquad g^{\mu\nu} = -v^\mu v^\nu/c^2 + h^{\mu\nu} \,, \tag{81}$$

which are invariant under the above transformations. In particular $\sqrt{-g} = c\sqrt{\gamma}$. In relativistic theories, one typically works with an energy-momentum tensor $T^\mu_\nu$ and charge current $J^\mu$. These are related to our Aristotelian quantities as

$$\epsilon^\mu = -T^\mu_\nu v^\nu \,, \qquad \pi_\nu = n_\mu T^\mu_\rho h^\rho_\nu \,, \qquad \tau^{\mu\nu} = h^\mu_\rho T^\rho_\sigma h^{\sigma\nu} \,, \qquad j^\mu = J^\mu \,. \tag{82}$$

It can be explicitly checked that the relativistic conservation equations $\nabla^{\rm rel}_\mu T^\mu_\nu = F_{\nu\rho}J^\rho$, $\nabla^{\rm rel}_\mu J^\mu = 0$, where $\nabla^{\rm rel}_\mu$ is the covariant derivative associated with $g_{\mu\nu}$, reduces to the respective boost-agnostic versions stated in eq. (18). Similar to the Galilean discussion, requiring $\beta^\mu$, $\Lambda_\beta$ to be invariant, we can derive the transformation of the hydrodynamic fields as

$$T \to \frac{T}{1 - \frac{1}{c^2}\left(\vec{u}^\mu\psi_\mu - \frac{1}{2}\psi^2\right)} \,, \qquad \mu \to \frac{\mu}{1 - \frac{1}{c^2}\left(\vec{u}^\mu\psi_\mu - \frac{1}{2}\psi^2\right)} \,, \qquad u^\mu \to \frac{u^\mu}{1 - \frac{1}{c^2}\left(\vec{u}^\mu\psi_\mu - \frac{1}{2}\psi^2\right)} \,. \tag{83}$$

---

[13]These can be derived using the reverse logic and noting that the relativistic metric $g_{\mu\nu}$ and inverse metric $g^{\mu\nu}$ are invariant and related to the Aristotelian sources as in eq. (81). Milne boosts (73) follow from the Lorentz boosts (79) by identifying $A^{\rm rel}_\mu = mc^2 n_\mu + A^{\rm gal}_\mu$ and taking $c \to \infty$.

We can define the relativistic versions of hydrodynamic fields according to

$$u_{\text{rel}}^\mu = \gamma_u u^\mu \,, \qquad T_{\text{rel}} = \gamma_u T \,, \qquad \mu_{\text{rel}} = \gamma_u \mu \,, \tag{84}$$

where $\gamma_u = 1/\sqrt{1 - \vec{u}^2/c^2}$ is the Lorentz factor. Note that $g_{\mu\nu} u_{\text{rel}}^\mu u_{\text{rel}}^\nu = -c^2$. The equation of state of a relativistic fluid is expressed as $p(T, \mu, \vec{u}^2) = p(T_{\text{rel}}, \mu_{\text{rel}})$. We find the thermodynamic relations

$$\mathrm{d}p = s_{\text{rel}}\mathrm{d}T_{\text{rel}} + n_{\text{rel}}\mathrm{d}\mu_{\text{rel}} \,, \qquad \epsilon_{\text{rel}} = T_{\text{rel}}s_{\text{rel}} + \mu_{\text{rel}}n_{\text{rel}} - p \,, \qquad \rho = \frac{\epsilon + p}{c^2} = \frac{\gamma_u^2}{c^2}(\epsilon_{\text{rel}} + p) \,, \tag{85}$$

where relativistic proper densities are defined as $n_{\text{rel}} = n/\gamma_u$, $s_{\text{rel}} = s/\gamma_u$, and $\epsilon_{\text{rel}} = \epsilon - \rho \vec{u}^2$. The "relativistic equation of state" is given as $\epsilon + p = \rho c^2$.

The constitutive relations for a relativistic fluid in the Landau frame $(T^\mu{}_\nu)_{\text{diss}} u_{\text{rel}}^\nu = J_{\text{diss}}^\mu u_{\text{rel}}^\nu g_{\mu\nu} = 0$, are given as (see e.g. [51])

$$T^\mu{}_\rho g^{\rho\nu} = \frac{1}{c^2}(\epsilon_{\text{rel}} + p_{\text{rel}})u_{\text{rel}}^\mu u_{\text{rel}}^\nu + p\, g^{\mu\nu} - T_{\text{rel}}\left(2\eta\Delta^{\rho(\mu}\Delta^{\nu)\sigma} + (\zeta - \tfrac{2}{d}\eta)\Delta^{\mu\nu}\Delta^{\rho\sigma}\right)\frac{1}{2}\delta_{\mathscr{B}} g_{\rho\sigma} \,,$$
$$J^\mu = n_{\text{rel}}u_{\text{rel}}^\mu - T_{\text{rel}}\sigma\,\Delta^{\mu\nu}\delta_{\mathscr{B}} A_\nu \,. \tag{86}$$

Expressing these according to the definitions (84), (85), and (82), we find

$$j^\mu = n u^\mu - T\gamma_u\sigma\left(\bar{\Delta}^{\mu\nu} + \gamma_u^2\frac{\vec{u}^2}{c^2}v^\mu v^\nu + 2\frac{\gamma_u^2}{c^2}v^{(\mu}\vec{u}^{\nu)}\right)\delta_{\mathscr{B}} A_\nu \,,$$
$$\epsilon^\mu = \left(\epsilon_{\text{rel}} + \rho\,\vec{u}^2\right)u^\mu + p\,\vec{u}^\mu - T\left(\frac{1}{c^2}v^\mu \vec{u}_\alpha + h_\alpha^\mu\right)\vec{u}_\nu\eta^{\alpha\nu\rho\sigma} \times$$
$$\frac{1}{2}\left[\delta_{\mathscr{B}} h_{\rho\sigma} - 2\vec{u}_{(\sigma}\left(\delta_{\mathscr{B}} n_{\rho)} - \frac{1}{c^2}v^\lambda\delta_{\mathscr{B}} h_{\rho)\lambda}\right) - \frac{2}{c^2}\vec{u}_\rho\vec{u}_\sigma v^\lambda\delta_{\mathscr{B}} n_\lambda\right] \,,$$
$$\pi^\mu = \rho\,\vec{u}^\mu - \frac{T}{c^2}\vec{u}_\nu\eta^{\mu\nu\rho\sigma}\frac{1}{2}\left[\delta_{\mathscr{B}} h_{\rho\sigma} - 2\vec{u}_{(\sigma}\left(\delta_{\mathscr{B}} n_{\rho)} - \frac{1}{c^2}v^\lambda\delta_{\mathscr{B}} h_{\rho)\lambda}\right) - \frac{2}{c^2}\vec{u}_\rho\vec{u}_\sigma v^\lambda\delta_{\mathscr{B}} n_\lambda\right] \,,$$
$$\tau^{\mu\nu} = \rho\,\vec{u}^\mu\vec{u}^\nu + p\,h^{\mu\nu} - T\eta^{\mu\nu\rho\sigma} \times$$
$$\frac{1}{2}\left[\delta_{\mathscr{B}} h_{\rho\sigma} - 2\vec{u}_{(\sigma}\left(\delta_{\mathscr{B}} n_{\rho)} - \frac{1}{c^2}v^\lambda\delta_{\mathscr{B}} h_{\rho)\lambda}\right) - \frac{2}{c^2}\vec{u}_\rho\vec{u}_\sigma v^\lambda\delta_{\mathscr{B}} n_\lambda\right] \,. \tag{87}$$

Here, we have defined

$$\eta^{\mu\nu\rho\sigma} = 2\gamma_u\eta\bar{\Delta}^{\rho(\mu}\bar{\Delta}^{\nu)\sigma} + \gamma_u(\zeta - \tfrac{2}{d}\eta)\bar{\Delta}^{\mu\nu}\bar{\Delta}^{\rho\sigma} \,, \qquad \bar{\Delta}^{\mu\nu} = h^{\mu\nu} + \frac{\gamma_u^2}{c^2}\vec{u}^\mu\vec{u}^\nu \,. \tag{88}$$

In section 4, we obtained the generic constitutive relations in the boost-agnostic representation. However, these results were presented in the thermodynamic density frame and not the Landau frame. To make contact between (87) and the explicit transport coefficients discussed in section 4, we need to perform a frame transformation to the density frame. Before doing this explicitly, one can already infer that

$$F_0 = F_1 = F_2 = 0 \,,$$
$$\bar{\mathfrak{s}}_{01} = \bar{\mathfrak{s}}_{02} = \bar{\mathfrak{s}}_{03} = \bar{\mathfrak{s}}_{12} = \bar{\mathfrak{s}}_{13} = \bar{\mathfrak{s}}_{23} = \bar{\mathfrak{v}}_{01} = \bar{\mathfrak{v}}_{02} = \bar{\mathfrak{v}}_{12} = 0 \,,$$
$$\mathfrak{s}_{01} = \mathfrak{s}_{11} = \mathfrak{s}_{12} = \mathfrak{s}_{13} = \mathfrak{v}_{01} = \mathfrak{v}_{11} = \mathfrak{v}_{12} = 0 \,. \tag{89a}$$

These follow from the fact that relativistic fluids, like their Galilean counterparts, do not have any transport coefficients in the hydrostatic and non-hydrostatic non-dissipative sectors at one-derivative order. Also, certain coefficients in the dissipative sector are zero due to the Lorentz

boost Ward identity (80). The remaining 10 coefficients are non-trivially related to $\eta$, $\zeta$, $\sigma$ according to

$$\mathfrak{s}_{00} = \frac{1}{\gamma_u}\alpha_3^2(1-\alpha_1)^2\left(\zeta + 2\tfrac{d-1}{d}\eta\right) + \frac{1}{\gamma_u}(1+\alpha_2)^2\sigma\,,$$

$$\mathfrak{s}_{02} = -\frac{(1-\alpha_1)^2\alpha_3}{\gamma_u^2}\left(\zeta + 2\tfrac{d-1}{d}\eta\right) - \frac{(1+\alpha_2)\alpha_2}{\gamma_u^2\alpha_3}\sigma\,,$$

$$\mathfrak{s}_{03} = -(1-\alpha_1)\alpha_3\left(\zeta - \tfrac{2}{d}\eta\right) + (1-\alpha_1)\alpha_1\alpha_3\left(\zeta + 2\tfrac{d-1}{d}\eta\right) - \frac{(1+\alpha_2)\alpha_2}{\alpha_3}\sigma\,,$$

$$\mathfrak{s}_{22} = \frac{(1-\alpha_1)^2}{\gamma_u^3}\left(\zeta + 2\tfrac{d-1}{d}\eta\right) + \frac{\alpha_2^2}{\gamma_u^3\alpha_3^2}\sigma\,,$$

$$\mathfrak{s}_{23} = \frac{(1-\alpha_1)}{\gamma_u}\left(\zeta - \tfrac{2}{d}\eta\right) - \frac{(1-\alpha_1)\alpha_1}{\gamma_u}\left(\zeta + 2\tfrac{d-1}{d}\eta\right) + \frac{\alpha_2^2}{\gamma_u\alpha_3^2}\sigma\,,$$

$$\mathfrak{s}_{33} = \gamma_u\left(\zeta + \tfrac{2}{d(d-1)}\eta\right) - 2\gamma_u\alpha_1\left(\zeta - \tfrac{2}{d}\eta\right) + \gamma_u\alpha_1^2\left(\zeta + 2\tfrac{d-1}{d}\eta\right) + \gamma_u\frac{\alpha_2^2}{\alpha_3^2}\sigma\,,$$

$$\mathfrak{v}_{00} = \gamma_u\sigma + \gamma_u\alpha_3^2\eta\,, \qquad \mathfrak{v}_{02} = -\alpha_3\eta\,, \qquad \mathfrak{v}_{22} = \frac{\eta}{\gamma_u}\,, \qquad \mathfrak{t} = \gamma_u\eta\,, \tag{90}$$

where we have defined the thermodynamic coefficients

$$\alpha_1 = \gamma_u^2\frac{\vec{u}^2}{c^2}\left(\frac{\partial p}{\partial\epsilon} + \frac{1}{|\vec{u}|}\frac{\partial p}{\partial|\pi|}\right) = \frac{-\frac{\vec{u}^2}{c^2}\left(\frac{\partial p}{\partial\epsilon_{\text{rel}}} + \frac{\gamma_u n}{\epsilon+p}\frac{\partial p}{\partial n_{\text{rel}}}\right)}{1 - \frac{\vec{u}^2}{c^2}\left(\frac{\partial p}{\partial\epsilon_{\text{rel}}} + \frac{\gamma_u n}{\epsilon+p}\frac{\partial p}{\partial n_{\text{rel}}}\right)}\,,$$

$$\alpha_2 = \gamma_u^2\frac{\vec{u}^2}{c^2}\frac{n}{\epsilon+p}\frac{\partial p}{\partial n} = \frac{\frac{\vec{u}^2}{c^2}\frac{\gamma_u n}{\epsilon+p}\frac{\partial p}{\partial n_{\text{rel}}}}{1 - \frac{\vec{u}^2}{c^2}\left(\frac{\partial p}{\partial\epsilon_{\text{rel}}} + \frac{\gamma_u n}{\epsilon+p}\frac{\partial p}{\partial n_{\text{rel}}}\right)}\,, \qquad \alpha_3 = \gamma_u\frac{n|\vec{u}|}{\epsilon+p}\,. \tag{91}$$

Further details about this derivation can be found in appendix A.2.

## 5.3 Lifshitz fluids

The final example we want to consider is that of a Lifshitz fluid, which is invariant under anisotropic scaling of spacetime coordinates $t \to \lambda^z t$, $x^i \to \lambda x^i$. Covariantly, we can define the Lifshitz symmetry as its action on the background sources

$$n_\mu \to \lambda^{-z}n_\mu\,, \qquad h_{\mu\nu} \to \lambda^{-2}h_{\mu\nu}\,, \qquad A_\mu \to A_\mu\,,$$
$$v^\mu \to \lambda^z v^\mu\,, \qquad h^{\mu\nu} \to \lambda^2 h^{\mu\nu}\,. \tag{92}$$

Plugging these into the variational expression eq. (15), we can derive the Lifshitz Ward identity

$$\tau^{\mu\nu}h_{\mu\nu} = z\,\epsilon^\mu n_\mu\,, \tag{93}$$

which is the covariant version of the respective identities introduced in [27, 29]. Demanding the partition function or effective action to be invariant under this scaling leads to the scaling behaviour of the conserved currents

$$\epsilon^\mu \to \lambda^{d+2z}\epsilon^\mu\,, \qquad \pi_\mu \to \lambda^d\pi_\mu\,, \qquad \tau^{\mu\nu} \to \lambda^{d+z+2}\tau^{\mu\nu}\,, \qquad j^\mu \to \lambda^{d+z}j^\mu\,. \tag{94}$$

Requiring that $\beta^\mu$, $\Lambda_\beta$ remain invariant, results in the scaling properties of the hydrodynamic fields

$$u^\mu \to \lambda^z u^\mu\,, \qquad T \to \lambda^z T\,, \qquad \mu \to \lambda^z\mu\,. \tag{95}$$

Note that $\vec{u}^2 \to \lambda^{2z-2}\vec{u}^2$ and $\hat{u}^\mu \to \lambda\hat{u}^\mu$. This implies that the scalar ratios $\mu/T$ and $\vec{u}^2/T^{2-2/z}$ are scale invariant. Implementing this for ideal fluids, one can infer that the equation of state of a Lifshitz fluid takes the form

$$p(T, \mu, \vec{u}^2) = T^{d/z+1} \hat{p}(\mu/T, \vec{u}^2/T^{2-2z}) . \tag{96}$$

This leads to the thermodynamic identity

$$\rho \, \vec{u}^2 + p \, d = z \, \epsilon , \tag{97}$$

which can also be derived directly using eq. (93).

As for the one-derivative order transport coefficients, the scaling behaviour goes as

$$v(T, \mu, \vec{u}^2) = T^{w_v/z} \hat{\mathfrak{c}}(\mu/T, \vec{u}^2/T^{2-2z}) , \tag{98}$$

where the weight factor $w_\lambda$ for the various coefficients is given by

$$w_v = \begin{cases} d+z & \text{for} \quad v = p , \\ d-z & \text{for} \quad v = F_0, F_1 , \\ d-2z+2 & \text{for} \quad v = F_2 , \\ d-2 & \text{for} \quad v = \mathfrak{v}_{00}, \mathfrak{s}_{00} , \\ d+z-2 & \text{for} \quad v = \mathfrak{v}_{01}, \mathfrak{s}_{01} , \bar{\mathfrak{v}}_{01}, \bar{\mathfrak{s}}_{01}, \\ d+2z-2 & \text{for} \quad v = \mathfrak{v}_{11}, \mathfrak{s}_{11} , \\ d-1 & \text{for} \quad v = \mathfrak{v}_{02}, \mathfrak{s}_{02}, \mathfrak{s}_{03} , \bar{\mathfrak{v}}_{02}, \bar{\mathfrak{s}}_{02}, \bar{\mathfrak{s}}_{03} , \\ d+z-1 & \text{for} \quad v = \mathfrak{v}_{12}, \mathfrak{s}_{12}, \mathfrak{s}_{13} , \bar{\mathfrak{v}}_{12}, \bar{\mathfrak{s}}_{12}, \bar{\mathfrak{s}}_{13}, \\ d & \text{for} \quad v = \mathfrak{t}, \mathfrak{v}_{22}, \mathfrak{s}_{22}, \mathfrak{s}_{23}, \mathfrak{s}_{33}, \bar{\mathfrak{s}}_{23} . \end{cases} \tag{99}$$

We have included the ideal order pressure for completeness. However, not all of these coefficients are independent. Firstly, the derivative corrections must satisfy the Lifshitz Ward identities (93). In particular, in the density frame, this implies that the non-hydrostatic corrections (including both dissipative and non-hydrostatic non-dissipative) must satisfy that $\tau^{\mu\nu}_{\text{nhs}} h_{\mu\nu} = 0$.[14] Furthermore, one-derivative scalars in the hydrostatic free energy density (53) can only come via the scale-covariant combinations

$$v^\mu \partial_\mu \left( \frac{\mu}{T} \right) , \qquad v^\mu \partial_\mu \left( \frac{\vec{u}^2}{T^{2-2/z}} \right) . \tag{100}$$

Similarly, all the non-hydrostatic data must appear in combinations

$$\delta_{\mathscr{B}} A_\mu , \qquad \delta_{\mathscr{B}} n_\mu - \frac{z}{2d} n_\mu h^{\rho\sigma} \delta_{\mathscr{B}} h_{\rho\sigma} , \qquad \delta_{\mathscr{B}} h_{\mu\nu} - \frac{1}{d} h_{\mu\nu} h^{\rho\sigma} \delta_{\mathscr{B}} h_{\rho\sigma} . \tag{101}$$

This leads to 8 constraints among the transport coefficients: 1 in the hydrostatic sector, 3 in the non-hydrostatic non-dissipative sector, and 4 in the dissipative sector, namely

$$F_1 = -2\frac{(z-1)\vec{u}^2}{z\,T} F_2 - \frac{\mu}{T} F_0 ,$$

$$\bar{\mathfrak{s}}_{02} = -(d-1)\bar{\mathfrak{s}}_{03} , \qquad \bar{\mathfrak{s}}_{12} = -(d-1)\bar{\mathfrak{s}}_{13} , \qquad \bar{\mathfrak{s}}_{23} = 0 ,$$

$$\mathfrak{s}_{02} = -(d-1)\mathfrak{s}_{03} , \qquad \mathfrak{s}_{12} = -(d-1)\mathfrak{s}_{13} , \qquad \mathfrak{s}_{22} = -(d-1)\mathfrak{s}_{23} = (d-1)^2 \mathfrak{s}_{33} . \tag{102}$$

---

[14]The condition is slightly more non-trivial in the Landau frame employed in [27, 29]: $\tau^{\mu\nu}_{\text{nhs}} h_{\mu\nu} = z \, \epsilon^\mu_{\text{nhs}} n_\mu$.

In the uncharged limit (65), the total number of transport coefficients after Lifshitz scaling agrees with that of [27]. Scale invariant Galilean fluids are compatible with $z = 2$ Lifshitz symmetry. An easy way to see this is that requiring $\mu_{\text{gal}} = \mu + m/2\,\vec{u}^2$ to scale homogeneously forces us to set $z = 2$. The thermodynamic equation of state, in this case, leads to $\epsilon_{\text{gal}} = d/2\,p$. Comparing the Galilean coefficients (78) to eq. (102) we read out that $\zeta = 0$. Similarly, scale invariant relativistic fluids are compatible with $z = 1$ isotropic scaling symmetry, since the relativistic hydrodynamic fields in (83) should scale homogeneously. The equation of state becomes $\epsilon_{\text{rel}} = d\,p$, while at the one-derivative order we again find $\zeta = 0$ by imposing the Lifshitz constraints (102) on eq. (90).

# 6  Linearised fluctuations

In section 4, we determined the explicit effective action and all transport coefficients appearing at first order in the derivative expansion for boost-agnostic fluids. In this section we study fluctuations around equilibrium anisotropic states and determine the mode structure. We ignore hydrostatic and non-hydrostatic non-dissipative contributions; these can also be systematically switched off by imposing T or PT symmetry. We also turn off the background fields and the resultant effective action is given in eq. (72). Additionally, for simplicity, we do not turn on all dissipative transport coefficients but instead consider small deviations away from Galilean constitutive relations. We begin by obtaining the linearised equations and later study the possible modes in a 3+1 dimensional fluid living in flat spacetime.

## 6.1  Linearised equations

We want to perturb the Lagrangian (72) around an equilibrium anisotropic state with non-zero fluid velocity $u^i$.[15] The equilibrium configuration is characterised by

$$\tau = t \;\;,\;\; \sigma^i = x^i \;\;,\;\; \varphi_r = 0 \;\;,\;\; X_a^t = X_a^i = \varphi_a = 0 \;\;,\tag{103}$$

with constant temperature $T = T_0$, constant chemical potential $\mu = \mu_0$, and constant non-zero spatial velocity $u^i = u_0^i$, as well as constant energy density, pressure, and charge density. In order to understand fluctuations around this state, we consider small perturbations of the particle/charge density $\delta n$, energy density $\delta\epsilon$, momentum density $\delta\pi_i$ and the stochastic variables $\delta X_a^t$, $\delta X_a^i$, and $\delta\varphi_a$. The variation of the Lagrangian (72) under these small perturbations and underlying assumptions becomes

$$\delta\mathcal{L} = \varphi_a^I K_I{}^J \mathcal{O}_J + \frac{i}{2}\varphi_a^I G_{IJ}\varphi_a^J \;\;,\tag{104}$$

---

[15]Ref. [27] considered the isotropic case with $u^i = 0$.

where we have introduced the operators

$$
\mathcal{O}_I = \begin{pmatrix} \delta n \\ \delta \epsilon \\ \hat{u}^i \delta \pi_i \\ P_i^j \delta \pi_j \end{pmatrix}, \qquad
\varphi_a^I = \begin{pmatrix} \delta \varphi_a \\ -\delta X_a^t \\ \hat{u}_i \delta X_a^i \\ P_j^i \delta X_a^j \end{pmatrix},
$$

$$
M_I{}^J = \begin{pmatrix}
|\vec{\pi}|\frac{\partial \hat{n}}{\partial n}\hat{\partial}_u & |\vec{\pi}|\frac{\partial \hat{n}}{\partial \epsilon}\hat{\partial}_u & \left(2\vec{\pi}^2\frac{\partial \hat{n}}{\partial \vec{\pi}^2}+\hat{n}\right)\hat{\partial}_u & \hat{n}\,\hat{\partial}_j \\
|\vec{\pi}|\frac{\partial \hat{w}}{\partial n}\hat{\partial}_u & |\vec{\pi}|\frac{\partial \hat{w}}{\partial \epsilon}\hat{\partial}_u & \left(2\vec{\pi}^2\frac{\partial \hat{w}}{\partial \vec{\pi}^2}+\hat{w}\right)\hat{\partial}_u & \hat{w}\,\hat{\partial}_j \\
\frac{\partial (\rho \vec{u}^2+p)}{\partial n}\hat{\partial}_u & \frac{\partial (\rho \vec{u}^2+p)}{\partial \epsilon}\hat{\partial}_u & 2|\vec{\pi}|\frac{\partial (\rho \vec{u}^2+p)}{\partial \vec{\pi}^2}\hat{\partial}_u & |\vec{u}|\hat{\partial}_j \\
\frac{\partial p}{\partial n}\hat{\partial}_i & \frac{\partial p}{\partial \epsilon}\hat{\partial}_i & 2|\vec{\pi}|\frac{\partial p}{\partial \vec{\pi}^2}\hat{\partial}_i & P_{ij}|\vec{u}|\hat{\partial}_u
\end{pmatrix},
$$

$$
\frac{G_{IJ}}{k_B T} = -2 \begin{pmatrix}
\mathfrak{v}_{00}\hat{\partial}^2+\mathfrak{s}_{00}\hat{\partial}_u^2 & \mathfrak{v}_{01}\hat{\partial}^2+\mathfrak{s}_{01}\hat{\partial}_u^2 & \mathfrak{v}_{02}\hat{\partial}^2+\mathfrak{s}_{02}\hat{\partial}_u^2 & (\mathfrak{v}_{02}+\mathfrak{s}_{03})\hat{\partial}_u\hat{\partial}_j \\
\mathfrak{v}_{01}\hat{\partial}^2+\mathfrak{s}_{01}\hat{\partial}_u^2 & \mathfrak{v}_{11}\hat{\partial}^2+\mathfrak{s}_{11}\hat{\partial}_u^2 & \mathfrak{v}_{12}\hat{\partial}^2+\mathfrak{s}_{12}\hat{\partial}_u^2 & (\mathfrak{v}_{12}+\mathfrak{s}_{13})\hat{\partial}_u\hat{\partial}_j \\
\mathfrak{v}_{02}\hat{\partial}^2+\mathfrak{s}_{02}\hat{\partial}_u^2 & \mathfrak{v}_{12}\hat{\partial}^2+\mathfrak{s}_{12}\hat{\partial}_u^2 & \mathfrak{v}_{22}\hat{\partial}^2+\mathfrak{s}_{22}\hat{\partial}_u^2 & (\mathfrak{v}_{22}+\mathfrak{s}_{23})\hat{\partial}_u\hat{\partial}_j \\
(\mathfrak{v}_{02}+\mathfrak{s}_{03})\hat{\partial}_u\hat{\partial}_i & (\mathfrak{v}_{12}+\mathfrak{s}_{13})\hat{\partial}_u\hat{\partial}_i & (\mathfrak{v}_{22}+\mathfrak{s}_{23})\hat{\partial}_u\hat{\partial}_i & \left(\mathfrak{s}_{33}+\frac{d-3}{d-1}\mathfrak{t}\right)\hat{\partial}_i\hat{\partial}_j + P_{ij}(\mathfrak{t}\hat{\partial}^2+\mathfrak{v}_{22}\hat{\partial}_u^2)
\end{pmatrix},
$$

$$
\chi_{IJ} = \begin{pmatrix}
T\partial(\mu/T)/\partial n & T\partial(\mu/T)/\partial \epsilon & 2T|\vec{\pi}|\partial(\mu/T)/\partial \vec{\pi}^2 & 0 \\
1/T\,\partial T/\partial n & 1/T\,\partial T/\partial \epsilon & 2/T\,|\vec{\pi}|\partial T/\partial \vec{\pi}^2 & 0 \\
T|\vec{\pi}|\partial(T^{-1}\rho^{-1})/\partial n & T|\vec{\pi}|\partial(T^{-1}\rho^{-1})/\partial \epsilon & 2T\vec{\pi}^2\partial(T^{-1}\rho^{-1})/\partial \vec{\pi}^2+\frac{1}{\rho} & 0 \\
0 & 0 & 0 & P^{ij}/\rho
\end{pmatrix}^{-1},
$$

$$
K_I{}^J = -\delta_I^J \partial_t - M_I{}^J - \frac{1}{2T} G_{IK}(\chi^{-1})^{KJ} \ . \tag{105}
$$

Here we have defined $\hat{n} = n/\rho$ and $\hat{w} = (\epsilon + p)/\rho$, and introduced the differential operators $\hat{\partial}_u = \hat{u}^k \partial_k$, $\hat{\partial}^i = P^{ij}\partial_j$, and $\hat{\partial}^2 = P^{ij}\partial_i\partial_j$. It can be explicitly checked that the susceptibility matrix $\chi_{IJ}$ is symmetric. All quantities appearing in $M_I{}^J$, $G_{IJ}$, and $\chi_{IJ}$ should be understood as being evaluated in the equilibrium configuration; we have dropped the subscript "0" for clarity. Varying the perturbed Lagrangian with respect to the stochastic variables $\varphi_a^I$, one obtains the linearised equations of motion. We will now use these to find the mode structure.

## 6.2 Mode structure

Since the linearised equations are given by $K_I{}^J \mathcal{O}_J = 0$, it is possible to find the dispersion relations by looking at the zeros of $\det(K_I{}^J)$. In 3+1 dimensions, we find a pair of sound modes (with a different velocity along or opposite the fluid flow), one number-density diffusion mode, one shear mode along the fluid velocity, and another shear mode transverse to the fluid velocity. Assuming plane-wave perturbations, in general, the modes have the following structure

$$
\omega - u^i k_i = v_s^{\pm}(\theta)k - \frac{i}{2}\Gamma_s(\theta)k^2,
$$

$$
\omega - u^i k_i = -iD_0(\theta)k^2, \qquad \omega - u^i k_i = -iD_1(\theta)k^2, \qquad \omega - u^i k_i = -iD_2(\theta)k^2. \tag{106}
$$

Here $\theta$ is the angle between $u^i$ and $k^i$, $\Gamma_s(\theta)$ is the attenuation constant, and $D_{0,1,2}(\theta)$ are diffusion constants. Explicitly, we find that the velocity of sound $v_s^{\pm}(\theta)$ are the solutions of the quadratic equation

$$
v_s^{\pm}(\theta)^2 + X|\vec{u}|\cos\theta\, v_s^{\pm}(\theta) + Y\vec{u}^2\cos^2\theta + Z = 0 \ , \tag{107}
$$

for which the functions X, Y, Z are not particularly illuminating. The general solution is given by

$$
v_s^{\pm}(\theta) = \pm \sqrt{ v_{s,0}^2 \left( 1 - |\vec{u}|^2 \cos^2 \theta \left( \frac{\partial \rho}{\partial \epsilon} + 2\rho \frac{\partial \rho}{\partial \vec{\pi}^2} \right) \right) + |\vec{u}|^2 \cos^2 \theta \left( \frac{\partial p}{\partial \epsilon} + 2 \frac{\partial p}{\partial \vec{\pi}^2} + \frac{v_{s,1}}{2} \right)^2 }
$$
$$
- \frac{|\vec{u}|}{2} \cos \theta \ v_{s,1} \ ,
\tag{108}
$$

where we have defined

$$
v_{s,0} = \sqrt{ \frac{\partial p}{\partial n} \frac{n}{\rho} + \frac{\partial p}{\partial \epsilon} \frac{w}{\rho} + 2\rho |\vec{u}|^2 \frac{\partial p}{\partial \vec{\pi}^2} } \ , \quad v_{s,1} = \frac{\partial \rho}{\partial n} \frac{n}{\rho} + \frac{\partial \rho}{\partial \epsilon} \frac{w}{\rho} - \left( 1 + \frac{\partial p}{\partial \epsilon} + 2\rho \frac{\partial p}{\partial \vec{\pi}^2} - 2\rho |\vec{u}|^2 \frac{\partial \rho}{\partial \vec{\pi}^2} \right).
\tag{109}
$$

The pair of sound modes in eq. (108) propagate with different speeds due to the presence of a non-zero equilibrium fluid speed $|\vec{u}|$. The two speeds are equal if we impose the Galilean or relativistic equations of state discussed in section 5. Hence, the distinction between the parallel and anti-parallel sound speeds, $v_s^+ \neq v_s^-$ is an imprint of broken boost symmetry. In the isotropic case $|\vec{u}| \to 0$, the two sound speeds are again equal to each other and reduce to the results of [28].

In turn, the diffusion constants $D_{0,1}(\theta)$ are solutions of the quadratic equation

$$
D_{0,1}(\theta)^2 + A(\theta) D_{0,1}(\theta) + B(\theta) = 0 \ ,
\tag{110}
$$

for which the functions $A(\theta)$ and $B(\theta)$ are some cumbersome functions of the thermodynamic variables, while the transverse shear diffusion constant $D_2(\theta)$ is simply given by

$$
D_2(\theta) = \frac{\mathfrak{t}}{\rho} + \frac{\mathfrak{v}_{22} - \mathfrak{t}}{\rho} \cos^2 \theta \ .
\tag{111}
$$

In order to provide an analytically tractable example of $D_{0,1}(\theta)$ and the attenuation constant $\Gamma_s(\theta)$, we consider slight departures away from the transport properties of a Galilean fluid, characterised only by three transport coefficients at first order, namely, $\kappa, \eta, \zeta$ (see section 5). This is still a non-trivial example because we are taking into account the modified thermodynamics due to the absence of a boost symmetry. In this special case, we can split the attenuation and diffusion constants into Galilean contributions and corrections due to the absence of boosts in a small velocity expansion such that

$$
\Gamma_s(\theta) = \Gamma_{\text{gal}}(\theta) + |\vec{u}|^2 \Gamma_{\text{u}}(\theta) \ , \quad D_{0,1}(\theta) = D_{0,1\text{gal}}(\theta) + |\vec{u}|^2 D_{0,1\text{u}}(\theta) \ ,
\tag{112}
$$

where, focusing on the case of $\theta = \pi/2$, the Galilean contributions are given by

$$
\Gamma_{\text{gal}} \left( \frac{\pi}{2} \right) = \frac{\kappa}{\rho v_{s,\text{gal}}^2} \frac{\partial p}{\partial \epsilon} \left( \frac{\partial T}{\partial n} n + \frac{\partial T}{\partial \epsilon} w \right) + \frac{3\zeta + 4\eta}{3\rho} \ , \quad v_{s,\text{gal}}^2 = \left( \frac{\partial p}{\partial n} \frac{n}{\rho} + \frac{\partial p}{\partial \epsilon} \frac{w}{\rho} \right) \ ,
$$
$$
D_{0,\text{gal}} \left( \frac{\pi}{2} \right) = \frac{n\kappa}{v_{s,\text{gal}}^2 \rho} \left( \frac{\partial p}{\partial n} \frac{\partial T}{\partial \epsilon} - \frac{\partial p}{\partial \epsilon} \frac{\partial T}{\partial n} \right) \ , \quad D_{1,\text{gal}} \left( \frac{\pi}{2} \right) = \frac{\eta}{\rho} \ .
\tag{113}
$$

If one imposes Galilean thermodynamics as in section 5, one obtains the diffusion and attenuation constants presented in [11]. On the other hand, the corrections due to the absence of

a boost symmetry are given by

$$\Gamma_u\left(\frac{\pi}{2}\right) = -\frac{1}{\rho^3 v_{s,gal}^4}\left[\eta\frac{\partial p}{\partial\epsilon}\rho v_{s,gal}^2\left(\frac{\partial\rho}{\partial n}n + \frac{\partial\rho}{\partial\epsilon}w\right)\right.$$
$$+\rho^2 v_{s,gal}^2\left(2\eta\frac{\partial p}{\partial\vec{\pi}^2}\left(\frac{\partial\rho}{\partial n}n + \frac{\partial\rho}{\partial\epsilon}w\right) + \frac{\partial p}{\partial\epsilon}\left(T\kappa\frac{\partial\rho}{\partial\epsilon} - \eta\right)\right)$$
$$\left.+\rho^3 v_{s,gal}^2\left(\kappa\frac{\partial p}{\partial\epsilon}\frac{\partial T}{\partial\epsilon} - 2\eta\frac{\partial p}{\partial\vec{\pi}^2}\right) + 2\rho^3\kappa\frac{\partial p}{\partial\epsilon}\frac{\partial p}{\partial\vec{\pi}^2}\left(\frac{\partial T}{\partial n}n + \frac{\partial T}{\partial\epsilon}w\right)\right], \qquad (114)$$

$$D_{0,u}\left(\frac{\pi}{2}\right) = -n\kappa\left[\eta\rho^2 v_{s,gal}^2\left(\frac{\partial p}{\partial n}\frac{\partial\rho}{\partial\epsilon} - \frac{\partial p}{\partial\epsilon}\frac{\partial\rho}{\partial n}\right)\left(Tv_{s,gal}^2\frac{\partial\rho}{\partial\epsilon} + \frac{\partial\rho}{\partial\epsilon}\left(\frac{\partial T}{\partial n}n + \frac{\partial T}{\partial\epsilon}w\right)\right)\right.$$
$$+\rho^3 v_{s,gal}^2\left(2\eta\frac{\partial p}{\partial\vec{\pi}^2}\left(\frac{\partial T}{\partial n}n + \frac{\partial T}{\partial\epsilon}w\right)\left(\frac{\partial p}{\partial n}\frac{\partial\rho}{\partial\epsilon} - \frac{\partial p}{\partial\epsilon}\frac{\partial\rho}{\partial n}\right)\right.$$
$$+\frac{\partial p}{\partial\epsilon}\left(T\kappa\frac{\partial\rho}{\partial\epsilon} - \eta\right)\left(\frac{\partial p}{\partial\epsilon}\frac{\partial T}{\partial n} + \frac{\partial p}{\partial n}\frac{\partial T}{\partial\epsilon}\right)\right)$$
$$\left.+\kappa\rho^4\frac{\partial p}{\partial\epsilon}\left(\frac{\partial p}{\partial n}\frac{\partial T}{\partial\epsilon} - \frac{\partial p}{\partial\epsilon}\frac{\partial T}{\partial n}\right)\left(\frac{\partial T}{\partial\epsilon}v_{s,gal}^2 + 2\frac{\partial p}{\partial\vec{\pi}^2}\left(\frac{\partial T}{\partial n}n + \frac{\partial T}{\partial\epsilon}w\right)\right)\right]$$
$$\times v_{s,gal}^4\rho^3\left(\frac{\partial p}{\partial n}n\left(\eta - \kappa\rho\frac{\partial T}{\partial\epsilon}\right) + \frac{\partial p}{\partial\epsilon}\left(\eta w + \kappa\rho\frac{\partial T}{\partial n}n\right)\right)^{-1}, \qquad (115)$$

$$D_{1,u}\left(\frac{\pi}{2}\right) = \eta\left[2\frac{\partial p}{\partial\vec{\pi}^2}T\rho\left(\eta\frac{\partial\rho}{\partial\epsilon}w - \eta\rho + n\kappa\rho\frac{\partial T}{\partial n}\frac{\partial\rho}{\partial\epsilon} + \kappa\rho^2\frac{\partial T}{\partial\epsilon} + \frac{\partial\rho}{\partial n}n\left(\eta - \frac{\partial T}{\partial\epsilon}\kappa\rho\right)\right)\right.$$
$$+\frac{\partial p}{\partial n}n\left(\frac{\partial\rho}{\partial\epsilon}\left(\kappa T^2\frac{\partial\rho}{\partial\epsilon} - 2T\left(\eta - \frac{\partial T}{\partial\epsilon}\kappa\rho\right)\right)\right.$$
$$\left.-\rho\left(\frac{\partial T}{\partial\epsilon} + 2T\frac{\partial\rho}{\partial\vec{\pi}^2} + 2\rho\frac{\partial T}{\partial\vec{\pi}^2}\right)\left(\eta - \frac{\partial T}{\partial\epsilon}\kappa\rho\right)\right)$$
$$-\frac{\partial p}{\partial\epsilon}\left(\frac{\partial\rho}{\partial n}nT\left(T\kappa\frac{\partial\rho}{\partial\epsilon} + \kappa\rho\frac{\partial T}{\partial\epsilon} - \eta\right) + T\frac{\partial\rho}{\partial\epsilon}\left(w\eta + \kappa\rho\left(\frac{\partial T}{\partial n}n - T\right)\right)\right)$$
$$+\rho\left(\left(\frac{\partial T}{\partial\epsilon} + 2\rho\frac{\partial T}{\partial\vec{\pi}^2}\right)\left(w\eta + \kappa\rho\frac{\partial T}{\partial n}n\right)\right.$$
$$\left.\left.+T\left(\eta + 2\eta w\frac{\partial\rho}{\partial\vec{\pi}^2} - \kappa\rho\frac{\partial T}{\partial\epsilon} + 2n\kappa\rho\frac{\partial\rho}{\partial\vec{\pi}^2}\frac{\partial T}{\partial n}\right)\right)\right]$$
$$\times \rho T\left(n\frac{\partial p}{\partial n}\left(\eta - \kappa\rho\frac{\partial T}{\partial\epsilon}\right) + \frac{\partial p}{\partial\epsilon}\left(w\eta + n\kappa\rho\frac{\partial T}{\partial n}\right)\right)^{-1}. \qquad (116)$$

As one can explicitly observe, the longitudinal shear diffusion constant $D_1(\theta)$ receives corrections due to a non-vanishing fluid velocity, while the transverse shear diffusion constant $D_2(\theta)$ in eq. (111) does not. This is again an imprint of the broken boost symmetry.

We would like to point out that the linear mode analysis presented above has been done at finite equilibrium fluid velocity $u^i = u_0^i$, and yet we do not encounter any additional gapped unphysical poles in the upper-half complex $\omega$ plane. This is in contrast to the Landau and Eckart frames typically employed in relativistic hydrodynamics, that are unstable in a finite fluid-velocity state; see [30, 31]. This affirms that the density frame introduced in this paper is a stable hydrodynamic frame, applicable to hydrodynamic theories with arbitrary boost symmetry structure – Galilean, Lorentzian, or absence thereof.

# 7 Outlook

The main goal of this paper was to formulate a Schwinger-Keldysh effective field theory for hydrodynamics without boosts. The formal construction, presented in section 3, was based on the recently developed EFT for Galilean hydrodynamics [11]. In the process of building this EFT, we provided a spacetime covariant framework for hydrodynamics without boosts and a rigorous offshell analysis of the independent transport coefficients together with the constraints that need to be satisfied for the second law of thermodynamics to hold (see eq. (64)). An accurate counting of transport coefficients reveals that there are 4 hydrostatic coefficients (including the ideal order pressure), 9 non-hydrostatic non-dissipative coefficients, and 17 independent dissipative transport coefficients. Thus, boost-agnostic hydrodynamics is characterised by a total of 30 independent transport coefficients up to first order in a gradient expansion.

Part of this work, specifically section 4, can also be seen as an extension of the covariant formulation of [26, 27] to include an additional particle/charge current. In the uncharged limit, our results agree with those of [27]. As such, we provided a general covariant framework for treating simultaneously Lorentzian, Galilean, and Lifshitz fluids; the respective boost and scaling limits were performed in section 5. In addition, we studied the general spectrum of linear modes around an anisotropic finite-velocity state as an application of this theory in section 6. This analysis revealed specific imprints of the absence of boost invariance that were previously unknown. In particular, in 3+1 spacetime dimensions, we found a pair of sounds with different velocities depending on whether the sound wave propagates along the equilibrium fluid velocity $u_0^i$ or opposite to it. Furthermore, the shear modes, which usually have multiplicity 2 in Galilean or relativistic fluids, now split into a shear mode along the fluid velocity and another transverse to it. Such imprints are clear smoking guns for potential experimental realisations of hydrodynamic systems without boost invariance, and were not visible when fluctuating around isotropic equilibrium states as in [28]. In relation to this, we note that we have provided our results in a new hydrodynamic frame that is linearly stable, irrespective of the boost symmetry in place, making the system of equations of motion and constitutive relations ideal for performing numerical simulations without running into unphysical artefacts.

Besides a unified framework that can treat different physical systems on the same footing, one of main goals of this work was to set the stage for future non-trivial extensions. As revealed in the introduction, systems of interest with broken boost symmetry exhibit intertwined patterns of symmetry breaking that can include spontaneous/explicit translation symmetry breaking, superfluidity, or liquid crystal phases. One of our main motivators are the hydrodynamic theories of flocking, such as the Toner-Tu model [21]. In such settings, not only is the boost symmetry explicitly broken, but there are also additional non-conserved driving forces responsible for the activity that breaks the spacetime translation symmetry. As far as we are aware, though widely used, such models lack a rigorous derivation in terms of an effective field theory or even a complete classical understanding and characterisation of the allowed transport. Another system of interest is that of quantum matter exhibiting charge density wave phases [19], in which, besides the absence of a boost symmetry, spatial translations are also typically broken explicitly and spontaneously. Schwinger-Keldysh EFT provides a route for understanding these systems, as it offers a controlled framework for symmetry breaking, moving away from classical hydrodynamics, and exploring its consequences (see e.g. [52] for an ideal order non-dissipative discussion). It will also be interesting to explore the purely non-equilibrium non-classical stochastic effects arising from the EFT analysis, such as those reported in [23], in the context of broken boost scenarios. We leave these explorations for future work.

A natural extension of this work is to consider the case of spontaneous breaking of Lorentz boost symmetry, and more generally, of Poincaré symmetry. This is directly relevant for condensed matter systems where a classification of phases of matter has been partially provided in [13]. Such extension would result in a finite temperature version of the same and, as the world is Lorentz invariant, would be highly relevant to pursue for real-world applications.

In the absence of any controlled experiment that could probe all 30 transport coefficients, it would be interesting to understand better the phenomenology and mode structure of fluids without a boost symmetry. To this aim, it would be relevant to understand whether holographic models, in the spirit of [1], exhibiting explicit Lorentz boost symmetry breaking could be constructed. An analysis of the black hole spectrum of quasinormal modes within such models would provide reasonable theoretical input for the equation of state and transport coefficients, allowing to better probe the physics of these systems.

# Acknowledgements

We would like to thank Jan de Boer and Niels A. Obers for useful discussions. JA is partly supported by the Netherlands Organization for Scientific Research (NWO) and by the Dutch Institute for Emergent Phenomena (DIEP) cluster at the University of Amsterdam. AJ is supported in part by the NSERC Discovery Grant program of Canada.

# A  Frame transformations and comparison with previous works

In this appendix we discuss frame transformations in boost-agnostic hydrodynamics in detail and discuss the translation of our results to those of [27, 29]. We give a general procedure to convert constitutive relations in arbitrary frame to our density frame. The analysis can equivalently be adapted to arrive at other hydrodynamic frames.

## A.1  Generalities of hydrodynamic frame transformation

We know that the hydrostatic part of the constitutive relations, in particular the leading derivative order ideal fluid, can be generated from a hydrostatic generating functional. To wit, we can start from a free energy density $\mathcal{N}$ and use the variational formulae

$$\frac{1}{\sqrt{\gamma}} \delta \left( \sqrt{\gamma} \, p \right) = j_0^\mu \delta A_\mu - \epsilon_0^\mu \delta n_\mu + \frac{1}{2} \left( 2 v^\mu \pi_0^\nu + \tau_0^{\mu\nu} \right) \delta h_{\mu\nu} + \mathfrak{h}_\mu \delta \beta^\mu + \mathfrak{n} \left( \delta \Lambda_\beta + A_\mu \delta \beta^\mu \right), \quad (117)$$

where "0" denotes the ideal part of the constitutive relations. Here $\mathfrak{h}_\mu$ and $\mathfrak{n}$ denote variations with respect to hydrodynamic fields, that do not contribute if we replace $\delta \to \delta_\mathcal{B}$, leading to the adiabaticity equation. This form is particularly useful because it allows us to directly read out the equations of motion associated with the hydrostatic part of the constitutive relations. Employing gauge and diffeomorphism invariance of $\mathcal{N}$, the equations of motion (18) can be re-expressed as

$$\frac{1}{\sqrt{\gamma}} \delta_\mathcal{B} \left( \sqrt{\gamma} \, \mathfrak{n} \right) = \mathcal{O}(\partial^2) , \qquad \frac{1}{\sqrt{\gamma}} \delta_\mathcal{B} \left( \sqrt{\gamma} \, \mathfrak{h}_\mu \right) + \mathfrak{n} \, \delta_\mathcal{B} A_\mu = \mathcal{O}(\partial^2) . \quad (118)$$

This form of the equations of motion was already derived for ideal fluids in eq. (23). Note that this is only the hydrostatic part of the equations of motion and will admit derivative corrections. But it will be useful for us in our discussion of frame transformations. Using eq. (117), the

equations of motion can also be expressed as

$$
\frac{1}{\sqrt{\gamma}}\delta_{\mathscr{B}}\left(\sqrt{\gamma}\,\mathfrak{n}\right)=\frac{\delta j_0^\rho}{\delta\Lambda_\beta}\,\delta_{\mathscr{B}}A_\rho-\frac{\delta\epsilon_0^\rho}{\delta\Lambda_\beta}\,\delta_{\mathscr{B}}n_\rho+\frac{1}{2}\frac{\delta(2v^\rho\pi_0^\sigma+\tau_0^{\rho\sigma})}{\delta\Lambda_\beta}\,\delta_{\mathscr{B}}h_{\rho\sigma}=\mathcal{O}(\partial^2)\,,
$$

$$
\frac{1}{\sqrt{\gamma}}\delta_{\mathscr{B}}\left(\sqrt{\gamma}\,(\mathfrak{h}_\mu+\mathfrak{n}A_\mu)\right)=\frac{\delta j_0^\rho}{\delta\beta^\mu}\,\delta_{\mathscr{B}}A_\rho-\frac{\delta\epsilon_0^\rho}{\delta\beta^\mu}\,\delta_{\mathscr{B}}n_\rho+\frac{1}{2}\frac{\delta(2v^\rho\pi_0^\sigma+\tau_0^{\rho\sigma})}{\delta\beta^\mu}\,\delta_{\mathscr{B}}h_{\rho\sigma}=\mathcal{O}(\partial^2)\,.
$$

(119)

For explicit computations, it is convenient to instead work with $u^\mu$, $T$, $\mu$, in terms of which we can recast these as

$$
\begin{pmatrix}
\frac{1}{T}\frac{\delta j_0^\rho}{\delta(\mu/T)} & \frac{1}{T}\frac{\delta\epsilon_0^\rho}{\delta(\mu/T)} & \frac{1}{T}\frac{\delta(2v^\rho\pi_0^\sigma+\tau_0^{\rho\sigma})}{\delta(\mu/T)} \\
T\frac{\delta j_0^\rho}{\delta T} & T\frac{\delta\epsilon_0^\rho}{\delta T} & T\frac{\delta(2v^\rho\pi_0^\sigma+\tau_0^{\rho\sigma})}{\delta T} \\
\frac{1}{T}h_\mu^\tau\frac{\delta j_0^\rho}{\delta(u^\tau/T)} & \frac{1}{T}h_\mu^\tau\frac{\delta\epsilon_0^\rho}{\delta(u^\tau/T)} & \frac{1}{T}h_\mu^\tau\frac{\delta(2v^\rho\pi_0^\sigma+\tau_0^{\rho\sigma})}{\delta(u^\tau/T)}
\end{pmatrix}
\begin{pmatrix}
\delta_{\mathscr{B}}A_\rho \\
-\delta_{\mathscr{B}}n_\rho \\
\frac{1}{2}\delta_{\mathscr{B}}h_{\rho\sigma}
\end{pmatrix}=\mathcal{O}(\partial^2)\,.
$$

(120)

We define the matrices

$$
\chi=\begin{pmatrix}
\frac{1}{T}\frac{n_\lambda\delta j_0^\lambda}{\delta(\mu/T)} & \frac{1}{T}\frac{n_\lambda\delta\epsilon_0^\lambda}{\delta(\mu/T)} & \frac{1}{T}\frac{\delta\pi_0^\lambda}{\delta(\mu/T)} \\
T\frac{n_\lambda\delta j_0^\lambda}{\delta T} & T\frac{n_\lambda\delta\epsilon_0^\lambda}{\delta T} & T\frac{\delta\pi_0^\lambda}{\delta T} \\
\frac{1}{T}h_\mu^\tau\frac{n_\lambda\delta j_0^\lambda}{\delta(u^\tau/T)} & \frac{1}{T}h_\mu^\tau\frac{n_\lambda\delta\epsilon_0^\lambda}{\delta(u^\tau/T)} & \frac{1}{T}h_\mu^\tau\frac{\delta\pi_0^\lambda}{\delta(u^\tau/T)}
\end{pmatrix},\quad
\chi_S=\begin{pmatrix}
\frac{1}{T}\frac{h_\lambda^\rho\delta j_0^\lambda}{\delta(\mu/T)} & \frac{1}{T}\frac{h_\lambda^\rho\delta\epsilon_0^\lambda}{\delta(\mu/T)} & \frac{1}{T}\frac{\delta\tau_0^{\rho\sigma}}{\delta(\mu/T)} \\
T\frac{h_\lambda^\rho\delta j_0^\lambda}{\delta T} & T\frac{h_\lambda^\rho\delta\epsilon_0^\lambda}{\delta T} & T\frac{\delta\tau_0^{\rho\sigma}}{\delta T} \\
\frac{1}{T}h_\mu^\tau\frac{h_\lambda^\rho\delta j_0^\lambda}{\delta(u^\tau/T)} & \frac{1}{T}h_\mu^\tau\frac{h_\lambda^\rho\delta\epsilon_0^\lambda}{\delta(u^\tau/T)} & \frac{1}{T}h_\mu^\tau\frac{\delta\tau_0^{\rho\sigma}}{\delta(u^\tau/T)}
\end{pmatrix}.
$$

(121)

Here $\chi$ is the same susceptibility matrix defined in section 6. Let us also define

$$
M=\chi^{-1}\chi_S=\begin{pmatrix}
\rho\frac{\partial\hat{n}}{\partial n}\vec{u}^\rho & \rho\frac{\partial\hat{w}}{\partial n}\vec{u}^\rho & \frac{\partial p}{\partial n}h^{\rho\sigma}-\frac{\partial\rho}{\partial n}\vec{u}^\rho\vec{u}^\sigma \\
\rho\frac{\partial\hat{n}}{\partial\epsilon}\vec{u}^\rho & \rho\frac{\partial\hat{w}}{\partial\epsilon}\vec{u}^\rho & \frac{\partial p}{\partial\epsilon}h^{\rho\sigma}-\frac{\partial\rho}{\partial\epsilon}\vec{u}^\rho\vec{u}^\sigma \\
\frac{\rho}{|\vec{u}|}\frac{\partial\hat{n}}{\partial|\pi|}\vec{u}_\mu\vec{u}^\rho+\hat{n}h_\mu^\rho & \frac{\rho}{|\vec{u}|}\frac{\partial\hat{w}}{\partial|\pi|}\vec{u}_\mu\vec{u}^\rho+\hat{w}h_\mu^\rho & \frac{\vec{u}_\mu}{|\vec{u}|}\left(\frac{\partial p}{\partial|\pi|}h^{\rho\sigma}-\frac{\partial\rho}{\partial|\pi|}\vec{u}^\rho\vec{u}^\rho\right)+2\vec{u}^{(\rho}h_\mu^{\sigma)}
\end{pmatrix}.
$$

(122)

Here we have used $w=\epsilon+p$, $\hat{n}=n/\rho$, and $\hat{w}=(\epsilon+p)/\rho$. This allows us to re-express the equations of motion (119) as

$$
\begin{pmatrix}
v^\rho\delta_{\mathscr{B}}A_\rho \\
-v^\rho\delta_{\mathscr{B}}n_\rho \\
v^\rho\delta_{\mathscr{B}}h_{\rho\sigma}
\end{pmatrix}=-M\begin{pmatrix}
\delta_{\mathscr{B}}A_\rho \\
-\delta_{\mathscr{B}}n_\rho \\
\frac{1}{2}\delta_{\mathscr{B}}h_{\rho\sigma}
\end{pmatrix}+\mathcal{O}(\partial^2)\,.
$$

(123)

On the other hand, one-derivative order frame transformations of the hydrodynamic fields $u^\mu$, $T$, and $\mu$ act as

$$
\delta\begin{pmatrix}
j^\mu \\
\epsilon^\mu \\
2v^{(\mu}\pi^{\nu)}+\tau^{\mu\nu}
\end{pmatrix}=\begin{pmatrix}
\frac{1}{T}\frac{\delta j_0^\mu}{\delta(\mu/T)} & T\frac{\delta j_0^\mu}{\delta T} & \frac{1}{T}\frac{\delta j_0^\mu}{\delta(u^\lambda/T)} \\
\frac{1}{T}h_\rho^\lambda\frac{\delta\epsilon_0^\mu}{\delta(\mu/T)} & T\frac{\delta\epsilon_0^\mu}{\delta T} & \frac{1}{T}\frac{\delta\epsilon_0^\mu}{\delta(u^\lambda/T)} \\
\frac{1}{T}h_\rho^\lambda\frac{\delta(2v^\mu\pi_0^\nu+\tau_0^{\mu\nu})}{\delta(\mu/T)} & T\frac{\delta(2v^\mu\pi_0^\nu+\tau_0^{\mu\nu})}{\delta T} & \frac{1}{T}h_\rho^\lambda\frac{\delta(2v^\mu\pi_0^\nu+\tau_0^{\mu\nu})}{\delta(u^\lambda/T)}
\end{pmatrix}\begin{pmatrix}
T\delta(\mu/T) \\
\frac{1}{T}\delta T \\
T\delta(u^\rho/T)
\end{pmatrix}.
$$

(124)

Using the decomposition of the constitutive relations into hydrostatic "hs" and non-hydrostatic "nhs" pieces (where "nhs" contains both "diss" and "nhsnd"), the density frame is defined as

$$
n_\mu j_{\text{nhs}}^\mu=n_\mu\epsilon_{\text{nhs}}^\mu=\pi_{\text{nhs}}^\mu=0\,.
$$

(125)

Denoting the respective corrections in the generic frame with tilde, we get

$$
\begin{pmatrix}
T\delta(\mu/T) \\
\frac{1}{T}\delta T \\
T\delta(u^\rho/T)
\end{pmatrix}=-\chi^{-\text{T}}\begin{pmatrix}
n_\lambda\tilde{j}_{\text{nhs}}^\lambda \\
n_\lambda\tilde{\epsilon}_{\text{nhs}}^\lambda \\
\tilde{\pi}_{\text{nhs}}^\lambda
\end{pmatrix}.
$$

(126)

The superscript "T" denotes a transpose and "−T" an inverse transpose. The non-hydrostatic corrections in the thermodynamic density frame can be written out explicitly as

$$
\begin{pmatrix} j^\mu_{\text{nhs}} \\ \epsilon^\mu_{\text{nhs}} \\ \tau^{\mu\nu}_{\text{nhs}} \end{pmatrix} = \begin{pmatrix} h^\mu_{\;\lambda} \tilde{j}^\lambda_{\text{nhs}} \\ h^\mu_{\;\lambda} \tilde{\epsilon}^\lambda_{\text{nhs}} \\ \tilde{\tau}^{\mu\nu}_{\text{nhs}} \end{pmatrix} - M^{\text{T}} \begin{pmatrix} n_\lambda \tilde{j}^\lambda_{\text{nhs}} \\ n_\lambda \tilde{\epsilon}^\lambda_{\text{nhs}} \\ \tilde{\pi}^\lambda_{\text{nhs}} \end{pmatrix} .
\tag{127}
$$

To obtain a mapping between the respective transport coefficients, we need to do a final manipulation requiring the usage of the equations of motion. We start by decomposing the generic frame non-hydrostatic constitutive relations as

$$
\begin{pmatrix} \tilde{j}^\mu_{\text{nhs}} \\ \tilde{\epsilon}^\mu_{\text{nhs}} \\ 2\nu^{(\mu}\tilde{\pi}^{\nu)}_{\text{nhs}} + \tilde{\tau}^{\mu\nu}_{\text{nhs}} \end{pmatrix} = -k_{\text{B}}T \begin{pmatrix} \tilde{C}^{\mu\rho}_{nn} & \tilde{C}^{\mu\rho}_{n\epsilon} & \tilde{C}^{\mu(\rho\sigma)}_{n\pi} \\ \tilde{C}^{\mu\rho}_{\epsilon n} & \tilde{C}^{\mu\rho}_{\epsilon\epsilon} & \tilde{C}^{\mu(\rho\sigma)}_{\epsilon\pi} \\ \tilde{C}^{(\mu\nu)\rho}_{\pi n} & \tilde{C}^{(\mu\nu)\rho}_{\epsilon\pi} & \tilde{C}^{(\mu\nu)(\rho\sigma)}_{\pi\pi} \end{pmatrix} \begin{pmatrix} \delta_{\mathscr{B}}A_\rho \\ -\delta_{\mathscr{B}}n_\rho \\ \frac{1}{2}\delta_{\mathscr{B}}h_{\rho\sigma} \end{pmatrix} .
\tag{128}
$$

Note that the coefficient matrix is asymmetric because it contains both "diss" and "nhsnd" pieces. Let us decompose it further into space and time parts

$$
\tilde{\mathfrak{C}}_{TT} = \begin{pmatrix} n_\mu n_\rho \tilde{C}^{\mu\rho}_{nn} & n_\mu n_\rho \tilde{C}^{\mu\rho}_{n\epsilon} & n_\mu n_\rho h^\tau_\sigma \tilde{C}^{\mu(\rho\sigma)}_{n\pi} \\ n_\mu n_\rho h^\tau_\sigma \tilde{C}^{\mu\rho}_{\epsilon n} & n_\mu n_\rho \tilde{C}^{\mu\rho}_{\epsilon\epsilon} & n_\mu n_\rho \tilde{C}^{\mu(\rho\sigma)}_{\epsilon\pi} \\ n_\mu h^\beta_\nu n_\rho \tilde{C}^{(\mu\nu)\rho}_{\pi n} & n_\mu h^\beta_\nu n_\rho \tilde{C}^{(\mu\nu)\rho}_{\epsilon\pi} & n_\mu h^\beta_\nu n_\rho h^\tau_\sigma \tilde{C}^{(\mu\nu)(\rho\sigma)}_{\pi\pi} \end{pmatrix} ,
$$

$$
\tilde{\mathfrak{C}}_{TS} = \begin{pmatrix} n_\mu h^\lambda_\rho \tilde{C}^{\mu\rho}_{nn} & n_\mu h^\lambda_\rho \tilde{C}^{\mu\rho}_{n\epsilon} & n_\mu h^\lambda_\rho h^\tau_\sigma \tilde{C}^{\mu(\rho\sigma)}_{n\pi} \\ n_\mu h^\lambda_\rho h^\tau_\sigma \tilde{C}^{\mu\rho}_{\epsilon n} & n_\mu h^\lambda_\rho \tilde{C}^{\mu\rho}_{\epsilon\epsilon} & n_\mu h^\lambda_\rho \tilde{C}^{\mu(\rho\sigma)}_{\epsilon\pi} \\ n_\mu h^\beta_\nu h^\lambda_\rho \tilde{C}^{(\mu\nu)\rho}_{\pi n} & n_\mu h^\beta_\nu h^\lambda_\rho \tilde{C}^{(\mu\nu)\rho}_{\epsilon\pi} & n_\mu h^\beta_\nu h^\lambda_\rho h^\tau_\sigma \tilde{C}^{(\mu\nu)(\rho\sigma)}_{\pi\pi} \end{pmatrix} ,
$$

$$
\tilde{\mathfrak{C}}_{ST} = \begin{pmatrix} h^\alpha_\mu n_\rho \tilde{C}^{\mu\rho}_{nn} & h^\alpha_\mu n_\rho \tilde{C}^{\mu\rho}_{n\epsilon} & h^\alpha_\mu n_\rho h^\tau_\sigma \tilde{C}^{\mu(\rho\sigma)}_{n\pi} \\ h^\alpha_\mu n_\rho h^\tau_\sigma \tilde{C}^{\mu\rho}_{\epsilon n} & h^\alpha_\mu n_\rho \tilde{C}^{\mu\rho}_{\epsilon\epsilon} & h^\alpha_\mu n_\rho \tilde{C}^{\mu(\rho\sigma)}_{\epsilon\pi} \\ h^\alpha_\mu h^\beta_\nu n_\rho \tilde{C}^{(\mu\nu)\rho}_{\pi n} & h^\alpha_\mu h^\beta_\nu n_\rho \tilde{C}^{(\mu\nu)\rho}_{\epsilon\pi} & h^\alpha_\mu h^\beta_\nu n_\rho h^\tau_\sigma \tilde{C}^{(\mu\nu)(\rho\sigma)}_{\pi\pi} \end{pmatrix} ,
$$

$$
\tilde{\mathfrak{C}}_{SS} = \begin{pmatrix} h^\alpha_\mu h^\lambda_\rho \tilde{C}^{\mu\rho}_{nn} & h^\alpha_\mu h^\lambda_\rho \tilde{C}^{\mu\rho}_{n\epsilon} & h^\alpha_\mu h^\lambda_\rho h^\tau_\sigma \tilde{C}^{\mu(\rho\sigma)}_{n\pi} \\ h^\alpha_\mu h^\lambda_\rho h^\tau_\sigma \tilde{C}^{\mu\rho}_{\epsilon n} & h^\alpha_\mu h^\lambda_\rho \tilde{C}^{\mu\rho}_{\epsilon\epsilon} & h^\alpha_\mu h^\lambda_\rho \tilde{C}^{\mu(\rho\sigma)}_{\epsilon\pi} \\ h^\alpha_\mu h^\beta_\nu h^\lambda_\rho \tilde{C}^{(\mu\nu)\rho}_{\pi n} & h^\alpha_\mu h^\beta_\nu h^\lambda_\rho \tilde{C}^{(\mu\nu)\rho}_{\epsilon\pi} & h^\alpha_\mu h^\beta_\nu h^\lambda_\rho h^\tau_\sigma \tilde{C}^{(\mu\nu)(\rho\sigma)}_{\pi\pi} \end{pmatrix} .
\tag{129}
$$

This explicitly results in the compact expressions

$$
\begin{pmatrix} n_\mu \tilde{j}^\mu_{\text{nhs}} \\ n_\mu \tilde{\epsilon}^\mu_{\text{nhs}} \\ \tilde{\pi}^\mu_{\text{nhs}} \end{pmatrix} = -k_{\text{B}}T\,\tilde{\mathfrak{C}}_{TT} \begin{pmatrix} \nu^\rho \delta_{\mathscr{B}}A_\rho \\ -\nu^\rho \delta_{\mathscr{B}}n_\rho \\ \nu^\rho \delta_{\mathscr{B}}h_{\rho\sigma} \end{pmatrix} - k_{\text{B}}T\,\tilde{\mathfrak{C}}_{TS} \begin{pmatrix} \delta_{\mathscr{B}}A_\rho \\ -\delta_{\mathscr{B}}n_\rho \\ \frac{1}{2}\delta_{\mathscr{B}}h_{\rho\sigma} \end{pmatrix} ,
$$

$$
\begin{pmatrix} h^\mu_{\;\lambda} \tilde{j}^\lambda_{\text{nhs}} \\ h^\mu_{\;\lambda} \tilde{\epsilon}^\lambda_{\text{nhs}} \\ \tilde{\tau}^{\mu\nu}_{\text{nhs}} \end{pmatrix} = -k_{\text{B}}T\,\tilde{\mathfrak{C}}_{ST} \begin{pmatrix} \nu^\rho \delta_{\mathscr{B}}A_\rho \\ -\nu^\rho \delta_{\mathscr{B}}n_\rho \\ \nu^\rho \delta_{\mathscr{B}}h_{\rho\sigma} \end{pmatrix} - k_{\text{B}}T\,\tilde{\mathfrak{C}}_{SS} \begin{pmatrix} \delta_{\mathscr{B}}A_\rho \\ -\delta_{\mathscr{B}}n_\rho \\ \frac{1}{2}\delta_{\mathscr{B}}h_{\rho\sigma} \end{pmatrix} .
\tag{130}
$$

Plugging these into eq. (127) and using the equations of motion, we get

$$
\begin{pmatrix} j^\mu_{\text{hs}} \\ \epsilon^\mu_{\text{hs}} \\ \tau^{\mu\nu}_{\text{hs}} \end{pmatrix} = -k_{\text{B}}T \begin{pmatrix} -M \\ 1 \end{pmatrix}^{\text{T}} \begin{pmatrix} \tilde{\mathfrak{C}}_{TT} & \tilde{\mathfrak{C}}_{TS} \\ \tilde{\mathfrak{C}}_{ST} & \tilde{\mathfrak{C}}_{SS} \end{pmatrix} \begin{pmatrix} -M \\ 1 \end{pmatrix} \begin{pmatrix} \delta_{\mathscr{B}}A_\rho \\ -\delta_{\mathscr{B}}n_\rho \\ \frac{1}{2}\delta_{\mathscr{B}}h_{\rho\sigma} \end{pmatrix} .
\tag{131}
$$

All in all, the transformation from a general frame to the density frame is given by the transformation of non-hydrostatic transport coefficients

$$\mathfrak{C} = \begin{pmatrix} -M \\ 1 \end{pmatrix}^{\mathrm{T}} \begin{pmatrix} \tilde{\mathfrak{C}}_{TT} & \tilde{\mathfrak{C}}_{TS} \\ \tilde{\mathfrak{C}}_{ST} & \tilde{\mathfrak{C}}_{SS} \end{pmatrix} \begin{pmatrix} -M \\ 1 \end{pmatrix}, \tag{132}$$

where $\mathfrak{C}$ is related to combinations of dissipative and non-hydrostatic non-dissipative coefficient matrices from section 4, in particular

$$\mathfrak{C} = \begin{pmatrix} D_{nn}^{\mu\rho} & D_{n\epsilon}^{\mu\rho} + \bar{D}_{n\epsilon}^{\mu\rho} & D_{n\pi}^{\mu(\rho\sigma)} + \bar{D}_{n\pi}^{\mu(\rho\sigma)} \\ D_{n\epsilon}^{\mu\rho} - \bar{D}_{n\epsilon}^{\rho\mu} & D_{\epsilon\epsilon}^{\mu\rho} & D_{\epsilon\pi}^{\mu(\rho\sigma)} + \bar{D}_{\epsilon\pi}^{\mu(\rho\sigma)} \\ D_{n\pi}^{\mu(\rho\sigma)} - \bar{D}_{n\pi}^{\rho(\mu\nu)} & D_{\epsilon\pi}^{\mu(\rho\sigma)} - \bar{D}_{\epsilon\pi}^{\rho(\mu\nu)} & D_{\pi\pi}^{(\mu\nu)(\rho\sigma)} + \bar{D}_{\pi\pi}^{(\mu\nu)(\rho\sigma)} \end{pmatrix}. \tag{133}$$

## A.2   Landau frame

To discuss the constitutive relations in Landau frame, it is convenient to define a covariant energy-momentum tensor [27, 29] for boost-agnostic fluids

$$T^{\mu}_{\ \nu} \equiv -\epsilon^{\mu} n_{\nu} + \nu^{\mu} \pi_{\nu} + \tau^{\mu\lambda} h_{\lambda\nu}, \tag{134}$$

where $\pi_{\mu} = h_{\mu\nu} \pi^{\nu}$. This energy-momentum tensor satisfies the conservation equations

$$\left( \nabla_{\mu} + F^{n}_{\mu\lambda} \nu^{\lambda} \right) T^{\mu}_{\ \nu} = F_{\nu\mu} j^{\mu} - F^{n}_{\nu\mu} \epsilon^{\mu} - \frac{1}{2} \pi^{\lambda} \$_{\nu} h_{\nu\lambda},$$

$$\implies \frac{1}{\sqrt{\gamma}} \partial_{\mu} \left( \sqrt{\gamma}\, T^{\mu}_{\ \nu} \right) + \epsilon^{\mu} \partial_{\nu} n_{\mu} - \frac{1}{2} \left( \nu^{\mu} \pi^{\rho} + \pi^{\rho} \nu^{\mu} + \tau^{\mu\rho} \right) \partial_{\nu} h_{\mu\rho} = F_{\nu\mu} j^{\mu}. \tag{135}$$

In addition, we have the charge current $j^{\mu}$ that still satisfies the same conservation equation as given in eq. (18). The thermodynamic density frame that we have employed in this work can be expressed as $n_{\mu}(T^{\mu}_{\ \nu})_{\mathrm{nhs}} = 0$, $n_{\mu} j^{\mu}_{\mathrm{nhs}} = 0$, where "nhs" collectively denotes dissipative and non-dissipative non-hydrostatic contributions. By contrast, we can define the thermodynamic Landau frame as $(T^{\mu}_{\ \nu})_{\mathrm{nhs}} u^{\nu} = 0$, $n_{\mu} j^{\mu}_{\mathrm{nhs}} = j^{\mu} \vec{u}_{\mu}/c^2$, leading to[16]

$$n_{\mu} \epsilon^{\mu}_{\mathrm{nhs}} = \pi^{\mu}_{\mathrm{nhs}} \vec{u}_{\mu}, \qquad h^{\mu}_{\ \nu} \epsilon^{\nu}_{\mathrm{nhs}} = \tau^{\mu\nu}_{\mathrm{nhs}} \vec{u}_{\nu}, \qquad n_{\mu} j^{\mu}_{\mathrm{nhs}} = \frac{1}{c^2} j^{\mu}_{\mathrm{nhs}} \vec{u}_{\mu}. \tag{136}$$

Equivalently, this amounts to working with

$$h^{\mu\nu} \delta_{\mathscr{B}} A_{\nu} + \frac{1}{c^2} \vec{u}^{\mu} \nu^{\nu} \delta_{\mathscr{B}} A_{\nu}, \qquad \delta_{\mathscr{B}} h_{\mu\nu} - 2\vec{u}_{(\mu} \delta_{\mathscr{B}} n_{\nu)}, \tag{137}$$

as the set of independent non-hydrostatic data.

The hydrostatic constitutive relations are still the same as section 4.1.1, but the non-hydrostatic non-dissipative and dissipative constitutive relations in the thermodynamic Landau frame are given as follows. Firstly, we have the non-hydrostatic non-dissipative densities

$$\begin{pmatrix} n_{\nu} \tilde{j}^{\nu}_{\mathrm{nhsnd}} \\ n_{\nu} \tilde{\epsilon}^{\nu}_{\mathrm{nhsnd}} \\ \tilde{\pi}^{\mu}_{\mathrm{nhsnd}} \end{pmatrix} = -k_{\mathrm{B}} T \begin{pmatrix} 0 & \frac{1}{c^2} \tilde{\bar{D}}^{\mu\rho}_{j\pi} \vec{u}_{\mu} \vec{u}_{\rho} & \frac{1}{c^2} \tilde{\bar{D}}^{\mu\rho}_{j\pi} \vec{u}_{\mu} \\ -\frac{1}{c^2} \tilde{\bar{D}}^{\rho\mu}_{j\pi} \vec{u}_{\rho} \vec{u}_{\mu} & 0 & 0 \\ -\frac{1}{c^2} \tilde{\bar{D}}^{\rho\mu}_{j\pi} \vec{u}_{\rho} & 0 & 0 \end{pmatrix} \begin{pmatrix} \nu^{\sigma} \delta_{\mathscr{B}} A_{\sigma} \\ -\nu^{\sigma} \delta_{\mathscr{B}} n_{\sigma} \\ \nu^{\sigma} \delta_{\mathscr{B}} h_{\rho\sigma} \end{pmatrix}$$
$$- k_{\mathrm{B}} T \begin{pmatrix} 0 & \frac{1}{c^2} \tilde{\bar{D}}^{\mu\rho\sigma}_{j\tau} \vec{u}_{\mu} \vec{u}_{\sigma} & \frac{1}{c^2} \tilde{\bar{D}}^{\mu\rho\sigma}_{j\tau} \vec{u}_{\mu} \\ -\tilde{\bar{D}}^{\rho\mu}_{j\pi} \vec{u}_{\mu} & \tilde{\bar{D}}^{\mu\rho\sigma}_{\pi\tau} \vec{u}_{\mu} \vec{u}_{\sigma} & \tilde{\bar{D}}^{\mu\rho\sigma}_{\pi\tau} \vec{u}_{\mu} \\ -\tilde{\bar{D}}^{\rho\mu}_{j\pi} & \tilde{\bar{D}}^{\mu\rho\sigma}_{\pi\tau} \vec{u}_{\sigma} & \tilde{\bar{D}}^{\mu\rho\sigma}_{\pi\tau} \end{pmatrix} \begin{pmatrix} \delta_{\mathscr{B}} A_{\rho} \\ -\delta_{\mathscr{B}} n_{\rho} \\ \frac{1}{2} \delta_{\mathscr{B}} h_{\rho\sigma} \end{pmatrix}, \tag{138}$$

---

[16]The thermodynamic Landau frame agrees with the true Landau frame used in [27, 29], defined as $T^{\mu}_{\ \nu} u^{\nu} = (\epsilon - \rho \vec{u}^2) u^{\mu}$, $n_{\mu} j^{\mu} = n + \frac{1}{c^2} j^{\mu} \vec{u}_{\mu}$, only in the non-hydrostatic sector. The Landau frame definition of [29] has an arbitrary function $B$ in the U(1) sector, which we have taken to be $(1 - \vec{u}^2/c^2)^{-1}$. Since $c$ is an arbitrary parameter at this stage, not having specialised to relativistic fluids, we can recover the generality of [29] by choosing $c$ appropriately.

and non-hydrostatic non-dissipative fluxes

$$
\begin{pmatrix}
\tilde{j}^{\mu}_{\text{nhsnd}} \\
\tilde{\epsilon}^{\mu}_{\text{nhsnd}} \\
\tilde{\tau}^{\mu\nu}_{\text{nhsnd}}
\end{pmatrix}
= -k_{\text{B}}T
\begin{pmatrix}
0 & \tilde{\tilde{D}}^{\mu\rho}_{j\pi}\vec{u}_{\rho} & \tilde{\tilde{D}}^{\mu\rho}_{j\pi} \\
-\frac{1}{c^2}\tilde{\tilde{D}}^{\rho\mu\nu}_{j\tau}\vec{u}_{\nu}\vec{u}_{\rho} & -\tilde{\tilde{D}}^{\rho\mu\nu}_{\pi\tau}\vec{u}_{\nu}\vec{u}_{\rho} & -\tilde{\tilde{D}}^{\rho\mu\nu}_{\pi\tau}\vec{u}_{\nu} \\
-\frac{1}{c^2}\tilde{\tilde{D}}^{\rho\mu\nu}_{j\tau}\vec{u}_{\rho} & -\tilde{\tilde{D}}^{\rho\mu\nu}_{\pi\tau}\vec{u}_{\rho} & -\tilde{\tilde{D}}^{\rho\mu\nu}_{\pi\tau}
\end{pmatrix}
\begin{pmatrix}
v^{\sigma}\delta_{\mathscr{B}}A_{\sigma} \\
-v^{\sigma}\delta_{\mathscr{B}}n_{\sigma} \\
v^{\sigma}\delta_{\mathscr{B}}h_{\rho\sigma}
\end{pmatrix}
$$
$$
- k_{\text{B}}T
\begin{pmatrix}
0 & \tilde{\tilde{D}}^{\mu\rho\sigma}_{j\tau}\vec{u}_{\sigma} & \tilde{\tilde{D}}^{\mu\rho\sigma}_{j\tau} \\
-\tilde{\tilde{D}}^{\rho\mu\nu}_{j\tau}\vec{u}_{\nu} & \tilde{\tilde{D}}^{\mu\nu\rho\sigma}_{\tau\tau}\vec{u}_{\nu}\vec{u}_{\sigma} & \tilde{\tilde{D}}^{\mu\nu\rho\sigma}_{\tau\tau}\vec{u}_{\nu} \\
-\tilde{\tilde{D}}^{\rho\mu\nu}_{j\tau} & \tilde{\tilde{D}}^{\mu\nu\rho\sigma}_{\tau\tau}\vec{u}_{\sigma} & \tilde{\tilde{D}}^{\mu\nu\rho\sigma}_{\tau\tau}
\end{pmatrix}
\begin{pmatrix}
\delta_{\mathscr{B}}A_{\rho} \\
-\delta_{\mathscr{B}}n_{\rho} \\
\frac{1}{2}\delta_{\mathscr{B}}h_{\rho\sigma}
\end{pmatrix},
\tag{139}
$$

where we have defined the transport coefficient matrices

$$
\tilde{\tilde{D}}^{\mu\rho}_{j\pi} = \tilde{\tilde{\mathfrak{v}}}_{01}P^{\mu\rho} + \tilde{\tilde{\mathfrak{s}}}_{01}\hat{u}^{\mu}\hat{u}^{\rho} \,,
$$
$$
\tilde{\tilde{D}}^{\mu\rho\sigma}_{j\tau} = 2\tilde{\tilde{\mathfrak{v}}}_{02}P^{\mu(\rho}\hat{u}^{\sigma)} + \tilde{\tilde{\mathfrak{s}}}_{02}\hat{u}^{\mu}\hat{u}^{\rho}\hat{u}^{\sigma} + \tilde{\tilde{\mathfrak{s}}}_{03}\hat{u}^{\mu}P^{\rho\sigma},
$$
$$
\tilde{\tilde{D}}^{\mu\rho\sigma}_{\pi\tau} = 2\tilde{\tilde{\mathfrak{v}}}_{12}P^{\mu(\rho}\hat{u}^{\sigma)} + \tilde{\tilde{\mathfrak{s}}}_{12}\hat{u}^{\mu}\hat{u}^{\rho}\hat{u}^{\sigma} + \tilde{\tilde{\mathfrak{s}}}_{13}\hat{u}^{\mu}P^{\rho\sigma},
$$
$$
\tilde{\tilde{D}}^{\mu\nu\rho\sigma}_{\tau\tau} = \tilde{\tilde{\mathfrak{s}}}_{23}\left(\hat{u}^{\mu}\hat{u}^{\nu}P^{\rho\sigma} - P^{\mu\nu}\hat{u}^{\rho}\hat{u}^{\sigma}\right).
\tag{140}
$$

On the other hand, in the dissipative sector we have the densities

$$
\begin{pmatrix}
n_{\nu}\tilde{j}^{\nu}_{\text{diss}} \\
n_{\nu}\tilde{\epsilon}^{\nu}_{\text{diss}} \\
\tilde{\pi}^{\mu}_{\text{nhsnd}}
\end{pmatrix}
= -k_{\text{B}}T
\begin{pmatrix}
\frac{1}{c^4}\tilde{D}^{\mu\rho}_{jj}\vec{u}_{\mu}\vec{u}_{\rho} & \frac{1}{c^2}\tilde{D}^{\mu\rho}_{j\pi}\vec{u}_{\mu}\vec{u}_{\rho} & \frac{1}{c^2}\tilde{D}^{\mu\rho}_{j\pi}\vec{u}_{\mu} \\
\frac{1}{c^2}\tilde{D}^{\rho\mu}_{j\pi}\vec{u}_{\mu}\vec{u}_{\rho} & \tilde{D}^{\mu\rho}_{\pi\pi}\vec{u}_{\mu}\vec{u}_{\rho} & \tilde{D}^{\mu\rho}_{\pi\pi}\vec{u}_{\mu} \\
\frac{1}{c^2}\tilde{D}^{\rho\mu}_{j\pi}\vec{u}_{\rho} & \tilde{D}^{\mu\rho}_{\pi\pi}\vec{u}_{\rho} & \tilde{D}^{\mu\rho}_{\pi\pi}
\end{pmatrix}
\begin{pmatrix}
v^{\sigma}\delta_{\mathscr{B}}A_{\sigma} \\
-v^{\sigma}\delta_{\mathscr{B}}n_{\sigma} \\
v^{\sigma}\delta_{\mathscr{B}}h_{\rho\sigma}
\end{pmatrix}
$$
$$
- k_{\text{B}}T
\begin{pmatrix}
\frac{1}{c^2}\tilde{D}^{\mu\rho}_{jj}\vec{u}_{\mu} & \frac{1}{c^2}\tilde{D}^{\mu\rho\sigma}_{j\tau}\vec{u}_{\mu}\vec{u}_{\sigma} & \frac{1}{c^2}\tilde{D}^{\mu\rho\sigma}_{j\tau}\vec{u}_{\mu} \\
\tilde{D}^{\rho\mu}_{j\pi}\vec{u}_{\mu} & \tilde{D}^{\mu\rho\sigma}_{\pi\tau}\vec{u}_{\mu}\vec{u}_{\sigma} & \tilde{D}^{\mu\rho\sigma}_{\pi\tau}\vec{u}_{\mu} \\
\tilde{D}^{\rho\mu}_{j\pi} & \tilde{D}^{\mu\rho\sigma}_{\pi\tau}\vec{u}_{\sigma} & \tilde{D}^{\mu\rho\sigma}_{\pi\tau}
\end{pmatrix}
\begin{pmatrix}
\delta_{\mathscr{B}}A_{\rho} \\
-\delta_{\mathscr{B}}n_{\rho} \\
\frac{1}{2}\delta_{\mathscr{B}}h_{\rho\sigma}
\end{pmatrix},
\tag{141}
$$

and fluxes

$$
\begin{pmatrix}
\tilde{j}^{\mu}_{\text{diss}} \\
\tilde{\epsilon}^{\mu}_{\text{diss}} \\
\tilde{\tau}^{\mu\nu}_{\text{diss}}
\end{pmatrix}
= -k_{\text{B}}T
\begin{pmatrix}
\frac{1}{c^2}\tilde{D}^{\mu\rho}_{jj}\vec{u}_{\rho} & \tilde{D}^{\mu\rho}_{j\pi}\vec{u}_{\rho} & \tilde{D}^{\mu\rho}_{j\pi} \\
\frac{1}{c^2}\tilde{D}^{\rho\mu\nu}_{j\tau}\vec{u}_{\nu}\vec{u}_{\rho} & \tilde{D}^{\rho\mu\nu}_{\pi\tau}\vec{u}_{\nu}\vec{u}_{\rho} & \tilde{D}^{\rho\mu\nu}_{\pi\tau}\vec{u}_{\nu} \\
\frac{1}{c^2}\tilde{D}^{\rho\mu\nu}_{j\tau}\vec{u}_{\rho} & \tilde{D}^{\rho\mu\nu}_{\pi\tau}\vec{u}_{\rho} & \tilde{D}^{\rho\mu\nu}_{\pi\tau}
\end{pmatrix}
\begin{pmatrix}
v^{\sigma}\delta_{\mathscr{B}}A_{\sigma} \\
-v^{\sigma}\delta_{\mathscr{B}}n_{\sigma} \\
v^{\sigma}\delta_{\mathscr{B}}h_{\rho\sigma}
\end{pmatrix}
$$
$$
- k_{\text{B}}T
\begin{pmatrix}
\tilde{D}^{\mu\rho}_{jj} & \tilde{D}^{\mu\rho\sigma}_{j\tau}\vec{u}_{\sigma} & \tilde{D}^{\mu\rho\sigma}_{j\tau} \\
\tilde{D}^{\rho\mu\nu}_{j\tau}\vec{u}_{\nu} & \tilde{D}^{\mu\nu\rho\sigma}_{\tau\tau}\vec{u}_{\nu}\vec{u}_{\sigma} & \tilde{D}^{\mu\nu\rho\sigma}_{\tau\tau}\vec{u}_{\nu} \\
\tilde{D}^{\rho\mu\nu}_{j\tau} & \tilde{D}^{\mu\nu\rho\sigma}_{\tau\tau}\vec{u}_{\sigma} & \tilde{D}^{\mu\nu\rho\sigma}_{\tau\tau}
\end{pmatrix}
\begin{pmatrix}
\delta_{\mathscr{B}}A_{\rho} \\
-\delta_{\mathscr{B}}n_{\rho} \\
\frac{1}{2}\delta_{\mathscr{B}}h_{\rho\sigma}
\end{pmatrix}.
\tag{142a}
$$

The coefficient matrices in the dissipative sector are given as

$$
\tilde{D}^{\mu\rho}_{jj} = \tilde{\mathfrak{v}}_{00}P^{\mu\rho} + \tilde{\mathfrak{s}}_{00}\hat{u}^{\mu}\hat{u}^{\rho} \,,
$$
$$
\tilde{D}^{\mu\rho}_{j\pi} = \tilde{\mathfrak{v}}_{01}P^{\mu\rho} + \tilde{\mathfrak{s}}_{01}\hat{u}^{\mu}\hat{u}^{\rho} \,,
$$
$$
\tilde{D}^{\mu\rho}_{\pi\pi} = \tilde{\mathfrak{v}}_{11}P^{\mu\rho} + \tilde{\mathfrak{s}}_{11}\hat{u}^{\mu}\hat{u}^{\rho} \,,
$$
$$
\tilde{D}^{\mu\rho\sigma}_{j\tau} = 2\tilde{\mathfrak{v}}_{02}P^{\mu(\rho}\hat{u}^{\sigma)} + \tilde{\mathfrak{s}}_{02}\hat{u}^{\mu}\hat{u}^{\rho}\hat{u}^{\sigma} + \tilde{\mathfrak{s}}_{03}\hat{u}^{\mu}P^{\rho\sigma},
$$
$$
\tilde{D}^{\mu\rho\sigma}_{\pi\tau} = 2\tilde{\mathfrak{v}}_{12}P^{\mu(\rho}\hat{u}^{\sigma)} + \tilde{\mathfrak{s}}_{12}\hat{u}^{\mu}\hat{u}^{\rho}\hat{u}^{\sigma} + \tilde{\mathfrak{s}}_{13}\hat{u}^{\mu}P^{\rho\sigma},
$$
$$
\tilde{C}^{\mu\nu\rho\sigma}_{\tau\tau} = 2\tilde{\mathfrak{t}}\left(P^{\rho(\mu}P^{\nu)\sigma} - \frac{1}{d-1}P^{\mu\nu}P^{\rho\sigma}\right) + 4\tilde{\mathfrak{v}}_{22}\hat{u}^{(\mu}P^{\nu)(\rho}\hat{u}^{\sigma)}
$$
$$
+ \tilde{\mathfrak{s}}_{22}\hat{u}^{\mu}\hat{u}^{\nu}\hat{u}^{\rho}\hat{u}^{\sigma} + \tilde{\mathfrak{s}}_{23}\left(P^{\mu\nu}\hat{u}^{\rho}\hat{u}^{\sigma} + \hat{u}^{\mu}\hat{u}^{\nu}P^{\rho\sigma}\right) + \tilde{\mathfrak{s}}_{33}P^{\mu\nu}P^{\rho\sigma} \,.
\tag{143}
$$

We can use the generic procedure chalked out in the previous subsection to map the thermodynamic Landau frame coefficients to the thermodynamic density frame used in the bulk

of the paper. The coefficient coupling to the traceless tensor remains unchanged during the map

$$\mathfrak{t} = \tilde{\mathfrak{t}} \, . \tag{144}$$

However, in the vector sector, we find

$$
\begin{aligned}
\mathfrak{v}_{00} &= \tilde{\mathfrak{v}}_{00} + \hat{n}^2 \tilde{\mathfrak{v}}_{11} - 2\hat{n}\tilde{\mathfrak{v}}_{01}, \\
\mathfrak{v}_{01} &= |\vec{u}|\tilde{\mathfrak{v}}_{02} + \hat{n}\hat{w}\tilde{\mathfrak{v}}_{11} - \hat{n}\tilde{\mathfrak{v}}_{12}|\vec{u}| - \hat{w}\tilde{\mathfrak{v}}_{01}, \\
\mathfrak{v}_{02} &= \tilde{\mathfrak{v}}_{02} + \hat{n}|\vec{u}|\tilde{\mathfrak{v}}_{11} - \hat{n}\tilde{\mathfrak{v}}_{12} - |\vec{u}|\tilde{\mathfrak{v}}_{01} \, , \\
\mathfrak{v}_{11} &= \vec{u}^2 \tilde{\mathfrak{v}}_{22} + \hat{w}^2 \tilde{\mathfrak{v}}_{11} - 2\hat{w}\tilde{\mathfrak{v}}_{12}|\vec{u}| \, , \\
\mathfrak{v}_{12} &= |\vec{u}|\tilde{\mathfrak{v}}_{22} + \hat{w}|\vec{u}|\tilde{\mathfrak{v}}_{11} - \left(\hat{w} + \vec{u}^2\right)\tilde{\mathfrak{v}}_{12} \, , \\
\mathfrak{v}_{22} &= \tilde{\mathfrak{v}}_{22} + \vec{u}^2 \tilde{\mathfrak{v}}_{11} - 2|\vec{u}|\tilde{\mathfrak{v}}_{12} \, , \\
\bar{\mathfrak{v}}_{01} &= |\vec{u}|\bar{\tilde{\mathfrak{v}}}_{02} - \hat{n}\bar{\tilde{\mathfrak{v}}}_{12}|\vec{u}| - \hat{w}\bar{\tilde{\mathfrak{v}}}_{01} \, , \\
\bar{\mathfrak{v}}_{02} &= \bar{\tilde{\mathfrak{v}}}_{02} - \hat{n}\bar{\tilde{\mathfrak{v}}}_{12} - |\vec{u}|\bar{\tilde{\mathfrak{v}}}_{01} \, , \\
\bar{\mathfrak{v}}_{12} &= |\vec{u}|\bar{\tilde{\mathfrak{v}}}_{22} - \left(\hat{w} + \vec{u}^2\right)\bar{\tilde{\mathfrak{v}}}_{12} \, .
\end{aligned}
\tag{145}
$$

The mapping in the scalar sector is much messier to write down explicitly. We instead use the matrix representation for clarity; we first isolate the scalar part of the $M$ matrix as

$$
M^{\mathfrak{s}} = \begin{pmatrix}
|\vec{\pi}|\frac{\partial \hat{n}}{\partial n} & |\vec{\pi}|\frac{\partial \hat{w}}{\partial n} & \frac{\partial p}{\partial n} - \frac{\partial \rho}{\partial n}\vec{u}^2 & \frac{\partial p}{\partial n} \\
|\vec{\pi}|\frac{\partial \hat{n}}{\partial \epsilon} & |\vec{\pi}|\frac{\partial \hat{w}}{\partial \epsilon} & \frac{\partial p}{\partial \epsilon} - \frac{\partial \rho}{\partial \epsilon}\vec{u}^2 & \frac{\partial p}{\partial \epsilon} \\
|\vec{\pi}|\frac{\partial \hat{n}}{\partial |\pi|} + \hat{n} & |\vec{\pi}|\frac{\partial \hat{w}}{\partial |\pi|} + \hat{w} & \frac{\partial p}{\partial |\pi|} - \frac{\partial \rho}{\partial |\pi|}\vec{u}^2 + 2|\vec{u}| & \frac{\partial p}{\partial |\pi|}
\end{pmatrix} .
\tag{146}
$$

We define the coefficient matrices in the Landau frame

$$
\tilde{\mathfrak{D}}^{\mathfrak{s}}_{TT} = \begin{pmatrix}
\frac{\vec{u}^2}{c^4}\tilde{\mathfrak{s}}_{00} & \frac{\vec{u}^2}{c^2}\tilde{\mathfrak{s}}_{01} & \frac{|\vec{u}|}{c^2}\tilde{\mathfrak{s}}_{01} \\
\frac{\vec{u}^2}{c^2}\tilde{\mathfrak{s}}_{01} & \vec{u}^2\tilde{\mathfrak{s}}_{11} & |\vec{u}|\tilde{\mathfrak{s}}_{11} \\
\frac{|\vec{u}|}{c^2}\tilde{\mathfrak{s}}_{01} & |\vec{u}|\tilde{\mathfrak{s}}_{11} & \tilde{\mathfrak{s}}_{11}
\end{pmatrix}, \qquad
\bar{\tilde{\mathfrak{D}}}^{\mathfrak{s}}_{TT} = \begin{pmatrix}
0 & \frac{\vec{u}^2}{c^2}\bar{\tilde{\mathfrak{s}}}_{01} & \frac{|\vec{u}|}{c^2}\bar{\tilde{\mathfrak{s}}}_{01} \\
-\frac{\vec{u}^2}{c^2}\bar{\tilde{\mathfrak{s}}}_{01} & 0 & 0 \\
-\frac{|\vec{u}|}{c^2}\bar{\tilde{\mathfrak{s}}}_{01} & 0 & 0
\end{pmatrix},
$$

$$
\tilde{\mathfrak{D}}^{\mathfrak{s}}_{TS} = \begin{pmatrix}
\frac{|\vec{u}|}{c^2}\tilde{\mathfrak{s}}_{00} & \frac{\vec{u}^2}{c^2}\tilde{\mathfrak{s}}_{02} & \frac{|\vec{u}|}{c^2}\tilde{\mathfrak{s}}_{02} & \frac{|\vec{u}|}{c^2}\tilde{\mathfrak{s}}_{03} \\
|\vec{u}|\tilde{\mathfrak{s}}_{01} & \vec{u}^2\tilde{\mathfrak{s}}_{12} & |\vec{u}|\tilde{\mathfrak{s}}_{12} & |\vec{u}|\tilde{\mathfrak{s}}_{13} \\
\tilde{\mathfrak{s}}_{01} & |\vec{u}|\tilde{\mathfrak{s}}_{12} & \tilde{\mathfrak{s}}_{12} & \tilde{\mathfrak{s}}_{13}
\end{pmatrix}, \qquad
\bar{\tilde{\mathfrak{D}}}^{\mathfrak{s}}_{TS} = \begin{pmatrix}
0 & \frac{\vec{u}^2}{c^2}\bar{\tilde{\mathfrak{s}}}_{02} & \frac{|\vec{u}|}{c^2}\bar{\tilde{\mathfrak{s}}}_{02} & \frac{|\vec{u}|}{c^2}\bar{\tilde{\mathfrak{s}}}_{03} \\
-|\vec{u}|\bar{\tilde{\mathfrak{s}}}_{01} & \vec{u}^2\bar{\tilde{\mathfrak{s}}}_{12} & |\vec{u}|\bar{\tilde{\mathfrak{s}}}_{12} & |\vec{u}|\bar{\tilde{\mathfrak{s}}}_{13} \\
-\bar{\tilde{\mathfrak{s}}}_{01} & |\vec{u}|\bar{\tilde{\mathfrak{s}}}_{12} & \bar{\tilde{\mathfrak{s}}}_{12} & \bar{\tilde{\mathfrak{s}}}_{13}
\end{pmatrix},
$$

$$
\tilde{\mathfrak{D}}^{\mathfrak{s}}_{SS} = \begin{pmatrix}
\tilde{\mathfrak{s}}_{00} & |\vec{u}|\tilde{\mathfrak{s}}_{02} & \tilde{\mathfrak{s}}_{02} & \tilde{\mathfrak{s}}_{03} \\
|\vec{u}|\tilde{\mathfrak{s}}_{02} & \vec{u}^2\tilde{\mathfrak{s}}_{22} & |\vec{u}|\tilde{\mathfrak{s}}_{22} & |\vec{u}|\tilde{\mathfrak{s}}_{23} \\
\tilde{\mathfrak{s}}_{02} & |\vec{u}|\tilde{\mathfrak{s}}_{22} & \tilde{\mathfrak{s}}_{22} & \tilde{\mathfrak{s}}_{23} \\
\tilde{\mathfrak{s}}_{03} & |\vec{u}|\tilde{\mathfrak{s}}_{23} & \tilde{\mathfrak{s}}_{23} & \tilde{\mathfrak{s}}_{33}
\end{pmatrix}, \qquad
\bar{\tilde{\mathfrak{D}}}^{\mathfrak{s}}_{SS} = \begin{pmatrix}
0 & |\vec{u}|\bar{\tilde{\mathfrak{s}}}_{02} & \bar{\tilde{\mathfrak{s}}}_{02} & \bar{\tilde{\mathfrak{s}}}_{03} \\
-|\vec{u}|\bar{\tilde{\mathfrak{s}}}_{02} & 0 & 0 & |\vec{u}|\bar{\tilde{\mathfrak{s}}}_{23} \\
-\bar{\tilde{\mathfrak{s}}}_{02} & 0 & 0 & \bar{\tilde{\mathfrak{s}}}_{23} \\
-\bar{\tilde{\mathfrak{s}}}_{03} & -|\vec{u}|\bar{\tilde{\mathfrak{s}}}_{23} & -\bar{\tilde{\mathfrak{s}}}_{23} & 0
\end{pmatrix},
\tag{147}
$$

and in the density frame

$$
\mathfrak{D}^{\mathfrak{s}}_{SS} = \begin{pmatrix}
\mathfrak{s}_{00} & \mathfrak{s}_{02} & \mathfrak{s}_{02} & \mathfrak{s}_{03} \\
\mathfrak{s}_{02} & \mathfrak{s}_{11} & \mathfrak{s}_{12} & \mathfrak{s}_{13} \\
\mathfrak{s}_{02} & \mathfrak{s}_{12} & \mathfrak{s}_{22} & \mathfrak{s}_{23} \\
\mathfrak{s}_{03} & \mathfrak{s}_{13} & \mathfrak{s}_{23} & \mathfrak{s}_{33}
\end{pmatrix}, \qquad
\bar{\mathfrak{D}}^{\mathfrak{s}}_{SS} = \begin{pmatrix}
0 & \bar{\mathfrak{s}}_{02} & \bar{\mathfrak{s}}_{02} & \bar{\mathfrak{s}}_{03} \\
-\bar{\mathfrak{s}}_{02} & 0 & \bar{\mathfrak{s}}_{12} & \bar{\mathfrak{s}}_{13} \\
-\bar{\mathfrak{s}}_{02} & -\bar{\mathfrak{s}}_{12} & 0 & \bar{\mathfrak{s}}_{23} \\
-\bar{\mathfrak{s}}_{03} & -\bar{\mathfrak{s}}_{13} & -\bar{\mathfrak{s}}_{23} & 0
\end{pmatrix}.
\tag{148}
$$

The mapping is given in terms of these as

$$
\begin{aligned}
\mathfrak{D}^{\mathfrak{s}}_{SS} &= \tilde{\mathfrak{D}}^{\mathfrak{s}}_{SS} - (M^{\mathfrak{s}})^{\mathrm{T}}\tilde{\mathfrak{D}}^{\mathfrak{s}}_{TS} - (\tilde{\mathfrak{D}}^{\mathfrak{s}}_{TS})^{\mathrm{T}}M^{\mathfrak{s}} + (M^{\mathfrak{s}})^{\mathrm{T}}\tilde{\mathfrak{D}}^{\mathfrak{s}}_{TT}M^{\mathfrak{s}} \, , \\
\bar{\mathfrak{D}}^{\mathfrak{s}}_{SS} &= \bar{\tilde{\mathfrak{D}}}^{\mathfrak{s}}_{SS} - (M^{\mathfrak{s}})^{\mathrm{T}}\bar{\tilde{\mathfrak{D}}}^{\mathfrak{s}}_{TS} + (\bar{\tilde{\mathfrak{D}}}^{\mathfrak{s}}_{TS})^{\mathrm{T}}M^{\mathfrak{s}} + (M^{\mathfrak{s}})^{\mathrm{T}}\bar{\tilde{\mathfrak{D}}}^{\mathfrak{s}}_{TT}M^{\mathfrak{s}} \, .
\end{aligned}
\tag{149}
$$

In section 5.2, we have used this procedure to obtain the mapping for a relativistic fluid in the Landau frame to the density frame. The transport coefficients for a relativistic fluid in the Landau frame are given as

$$F_0 = F_1 = F_2 = 0,$$

$$\tilde{\tilde{\mathfrak{s}}}_{01} = \tilde{\tilde{\mathfrak{s}}}_{02} = \tilde{\tilde{\mathfrak{s}}}_{03} = \tilde{\tilde{\mathfrak{s}}}_{12} = \tilde{\tilde{\mathfrak{s}}}_{13} = \tilde{\tilde{\mathfrak{s}}}_{23} = \tilde{\tilde{\mathfrak{v}}}_{01} = \tilde{\tilde{\mathfrak{v}}}_{02} = \tilde{\tilde{\mathfrak{v}}}_{12} = 0 \ ,$$

$$\tilde{\mathfrak{s}}_{00} = \gamma_u^3 \sigma, \qquad \tilde{\mathfrak{s}}_{01} = \tilde{\mathfrak{s}}_{02} = \tilde{\mathfrak{s}}_{03} = 0,$$

$$\tilde{\mathfrak{s}}_{22} = \frac{c^2}{|\vec{u}|}\tilde{\mathfrak{s}}_{12} = \frac{c^4}{\vec{u}^2}\tilde{\mathfrak{s}}_{11} = \gamma_u^5\left(\zeta + 2\frac{d-1}{d}\eta\right), \ \tilde{\mathfrak{s}}_{23} = \frac{c^2}{|\vec{u}|}\tilde{\mathfrak{s}}_{13} = \gamma_u^3\left(\zeta - \frac{2}{d}\eta\right), \ \tilde{\mathfrak{s}}_{33} = \gamma_u\left(\zeta + \frac{2}{d(d-1)}\eta\right),$$

$$\mathfrak{v}_{00} = \gamma_u \sigma, \qquad \tilde{\mathfrak{v}}_{01} = \tilde{\mathfrak{v}}_{02} = 0, \qquad \tilde{\mathfrak{v}}_{22} = \frac{c^2}{|\vec{u}|}\tilde{\mathfrak{v}}_{12} = \frac{c^4}{\vec{u}^2}\tilde{\mathfrak{v}}_{11} = \gamma_u^3\eta, \qquad \tilde{\mathfrak{t}} = \gamma_u\eta \ . \tag{150}$$

We can use the formulas mentioned above to recover the respective transport coefficients in the density frame reported in eq. (90). While performing the mapping, it is useful to note that the relativistic equation of state implies the identities

$$\frac{\partial p}{\partial \epsilon} = \frac{\frac{1}{\gamma_u^2}\frac{\partial p}{\partial \epsilon_{\text{rel}}} + 2\frac{\vec{u}^2}{c^2}\frac{\partial p}{\partial \epsilon_{\text{rel}}} + \frac{\vec{u}^2}{c^2}\frac{\gamma_u n}{\epsilon+p}\frac{\partial p}{\partial n_{\text{rel}}}}{1 - \frac{\vec{u}^2}{c^2}\frac{\partial p}{\partial \epsilon_{\text{rel}}} - \frac{\vec{u}^2}{c^2}\frac{\gamma_u n}{\epsilon+p}\frac{\partial p}{\partial n_{\text{rel}}}}, \qquad \frac{\partial p}{\partial n} = \frac{\frac{1}{\gamma_u}\frac{\partial p}{\partial n_{\text{rel}}}}{1 - \frac{\vec{u}^2}{c^2}\frac{\partial p}{\partial \epsilon_{\text{rel}}} - \frac{\vec{u}^2}{c^2}\frac{\gamma_u n}{\epsilon+p}\frac{\partial p}{\partial n_{\text{rel}}}} \ ,$$

$$\frac{\partial p}{\partial \pi^2} = \frac{-\frac{1}{\rho}\left(\frac{\partial p}{\partial \epsilon_{\text{rel}}} + \frac{\gamma_u n}{2(\epsilon+p)}\frac{\partial p}{\partial n_{\text{rel}}}\right)}{1 - \frac{\vec{u}^2}{c^2}\frac{\partial p}{\partial \epsilon_{\text{rel}}} - \frac{\vec{u}^2}{c^2}\frac{\gamma_u n}{\epsilon+p}\frac{\partial p}{\partial n_{\text{rel}}}} \ , \qquad \hat{w} = \frac{\epsilon+p}{\rho} = c^2 \ . \tag{151}$$

### A.3 Comparison to previous works

The Landau frame dissipative and non-dissipative non-hydrostatic transport coefficients appearing above can be related to the ones discussed in the uncharged case in eq. (5.6) of [27] as[17]

$$\tilde{\tilde{\mathfrak{v}}}_{12} = -\frac{f^{\text{NHS}}}{|\vec{u}|} \ , \qquad \tilde{\tilde{\mathfrak{s}}}_{12} = -\frac{s_2^{\text{NHS}}}{|\vec{u}|} \ , \qquad \tilde{\tilde{\mathfrak{s}}}_{13} = -\frac{s_1^{\text{NHS}}}{|\vec{u}|} \ , \qquad \tilde{\tilde{\mathfrak{s}}}_{23} = s_3^{\text{NHS}} \ ,$$

$$\tilde{\mathfrak{t}} = -t \ , \qquad \tilde{\mathfrak{v}}_{11} = -\frac{f_1}{\vec{u}^2} \ , \qquad \tilde{\mathfrak{v}}_{12} = -\frac{f_3}{|\vec{u}|} \ , \qquad \tilde{\mathfrak{v}}_{22} = -f_2 \ ,$$

$$\tilde{\mathfrak{s}}_{11} = -\frac{s_1}{\vec{u}^2} \ , \quad \tilde{\mathfrak{s}}_{12} = -\frac{s_4}{|\vec{u}|} \ , \quad \tilde{\mathfrak{s}}_{13} = -\frac{s_5}{|\vec{u}|} \ , \qquad \tilde{\mathfrak{s}}_{22} = -s_2 \ , \quad \tilde{\mathfrak{s}}_{23} = -s_6 \ , \quad \tilde{\mathfrak{s}}_{33} = -s_3 \ . \tag{152}$$

9 non-hydrostatic non-dissipative and 17 dissipative coefficients reduce to 4 and 10 respectively in the uncharged case, as reported by [27]. In addition, three hydrostatic coefficients $F_{0,1,2}$ reduce down to just two $F_{1,2}$, as commented upon in section 4.1.1.

The comparison of our work to the analysis of [29], on the other hand, is considerably more involved.[18] Firstly, comparison with the constitutive relations, given in eq. (2.24) of [29], can only be done in the limit $c \to \infty$ in the thermodynamic Landau frame definition in eq. (136), or equivalently $B \to 0$ in eq. (2.23) in [29]. For $B \neq 0$, the basis of independent non-hydrostatic data used in [29] is not compatible with the off-shell formalism because the resultant dissipation matrices in eq. (142) are asymmetric. Specialising to the $c \to \infty$ case, we can find the mapping of the transport coefficients $\bar{\eta}, \bar{\zeta}, \bar{\sigma}, \bar{\alpha}, \bar{\gamma}, \bar{\pi}, \gamma_{1,\dots,23}$ appearing in eq. (2.24) of [29]

---

[17]The dissipative coefficient $t$ has been called $\mathfrak{t}$ in [27] and is negative semi-definite. We use the notation $t$ to avoid sign confusion with our convention of positive semi-definite dissipative coefficients. The mapping of transport coefficients with [27] requires that we flip $v^\mu \to -v^\mu$.

[18]We thank the authors of [29] for aiding us in this comparison.

to the ones introduced by us; in the non-hydrostatic non-dissipative sector we find[19]

$$\tilde{\tilde{\mathfrak{s}}}_{01} = \bar{\gamma} - \left(\frac{\gamma_4}{2T} + \gamma_{18}\right)\vec{u}^2, \quad \tilde{\tilde{\mathfrak{s}}}_{02} = -|\vec{u}|\left(\frac{\gamma_{12}}{2T} + \frac{\gamma_{14}}{2T} + \gamma_{19} + \frac{\gamma_{20}}{2}\right), \quad \tilde{\tilde{\mathfrak{s}}}_{03} = -\frac{|\vec{u}|}{2}\left(\gamma_{20} - \frac{\gamma_{17}}{T}\right),$$

$$\tilde{\tilde{\mathfrak{s}}}_{12} = |\vec{u}|\left(\gamma_2 + \frac{\gamma_3}{2} - \gamma_{10} - \frac{\gamma_7}{2}\right), \qquad \tilde{\tilde{\mathfrak{s}}}_{13} = |\vec{u}|\left(\frac{\gamma_3}{2} + \gamma_{16}\right), \qquad \tilde{\tilde{\mathfrak{s}}}_{23} = \vec{u}^2\left(\frac{\gamma_{11}}{2} + \gamma_{15}\right),$$

$$\tilde{\tilde{\mathfrak{v}}}_{01} = \bar{\gamma}, \qquad \tilde{\tilde{\mathfrak{v}}}_{02} = -|\vec{u}|\left(\frac{\gamma_{14}}{2T} + \gamma_{22}\right), \qquad \tilde{\tilde{\mathfrak{v}}}_{12} = -|\vec{u}|\left(\frac{\gamma_7}{2} + \gamma_5\right), \tag{153}$$

and in the dissipative sector

$$\tilde{\mathfrak{s}}_{00} = \bar{\sigma} - \frac{\vec{u}^2}{T}\gamma_{21}, \qquad \tilde{\mathfrak{s}}_{01} = -\bar{\alpha} + \left(\frac{\gamma_4}{2T} - \gamma_{18}\right)\vec{u}^2, \qquad \tilde{\mathfrak{s}}_{11} = \bar{\pi} + 2\vec{u}^2\gamma_1,$$

$$\tilde{\mathfrak{s}}_{02} = |\vec{u}|\left(\frac{\gamma_{12}}{2T} + \frac{\gamma_{14}}{2T} - \gamma_{19} - \frac{\gamma_{20}}{2}\right), \qquad \tilde{\mathfrak{s}}_{03} = -\frac{|\vec{u}|}{2}\left(\gamma_{20} + \frac{\gamma_{17}}{T}\right),$$

$$\tilde{\mathfrak{s}}_{12} = |\vec{u}|\left(\gamma_2 + \frac{\gamma_3}{2} + \gamma_{10} + \frac{\gamma_7}{2}\right), \qquad \tilde{\mathfrak{s}}_{13} = |\vec{u}|\left(\frac{\gamma_3}{2} - \gamma_{16}\right)$$

$$\tilde{\mathfrak{s}}_{23} = \bar{\zeta} + 2\frac{d-1}{d}\bar{\eta} + \vec{u}^2\left(\frac{\gamma_{11}}{2} - \gamma_{15}\right), \qquad \tilde{\mathfrak{s}}_{33} = \bar{\zeta} + \frac{2}{d(d-1)}\bar{\eta},$$

$$\tilde{\mathfrak{v}}_{00} = \bar{\sigma}, \qquad \tilde{\mathfrak{v}}_{01} = -\bar{\alpha}, \qquad \tilde{\mathfrak{v}}_{11} = \bar{\pi}, \qquad \tilde{\mathfrak{v}}_{02} = |\vec{u}|\left(\frac{\gamma_{14}}{2T} - \gamma_{22}\right), \qquad \tilde{\mathfrak{v}}_{12} = |\vec{u}|\left(\frac{\gamma_7}{2} - \gamma_5\right),$$

$$\tilde{\mathfrak{v}}_{22} = 2\gamma_8, \qquad \tilde{\mathfrak{s}}_{22} = \bar{\zeta} + 2\frac{d-1}{d}\bar{\eta} + \vec{u}^2(4\gamma_8 + 2\gamma_9 + \gamma_{11}), \qquad \tilde{\mathfrak{t}} = \bar{\eta}. \tag{154}$$

The three remaining coefficients $\gamma_6 - 2\gamma_5$, $\gamma_{13} - 2\gamma_8$, $\gamma_{23} - 2\gamma_{22}$ from [29] do not appear in the maps above. They will, however, get non-trivial contributions in the hydrostatic sector from $F_{0,1,2}$ in section 4.1.1. We do not perform this detailed analysis here.

The authors in [29] introduced a different set of dissipative coefficients $b_{0,\ldots,21}$ for the entropy-production quadratic form $\Delta$ in eqs. (2.33)-(2.39), and hydrostatic coefficients $\tilde{c}_{1,2,4,8}$ for the non-canonical entropy current $s^{\mu}_{\text{non-can}}$ in eqs. (2.30)-(2.32). The relation to the afore-mentioned $\bar{\eta}$, $\bar{\zeta}$, $\bar{\sigma}$, $\bar{\alpha}$, $\bar{\gamma}$, $\bar{\pi}$, $\gamma_{1,\ldots,23}$ coefficients is presented by the authors in a companion notebook. They also find 2 constraints

$$b_{15} = b_{14}, \qquad b_{20} + b_{21} = 2b_{19}. \tag{155}$$

It should be noted that a complete analysis of the second law constraints is not provided in [29]. It should also be noted that 9 non-dissipative non-hydrostatic coefficients, given in eq. (2.48)-(2.56) of [29], do not show up in the non-canonical entropy current or entropy production. To map the $b_{0,\ldots,21}$, $\tilde{c}_{1,2,4,8}$ coefficients to our formalism, it is easier to work in the thermodynamic density frame. Mapping the $\Delta$'s in the two frameworks, we find that the 20 independent $b_{0,\ldots,14}$, $b_{16,\ldots,20}$ coefficients map to 17 dissipative coefficients

$$\frac{1}{k_{\mathrm{B}}T}\begin{pmatrix}\Lambda^{\mathrm{T}} & 0 \\ 0 & 1\end{pmatrix}\begin{pmatrix}\mathfrak{s}_{00} & \mathfrak{s}_{01} & \mathfrak{s}_{02} & \mathfrak{s}_{03} \\ \mathfrak{s}_{01} & \mathfrak{s}_{11} & \mathfrak{s}_{12} & \mathfrak{s}_{13} \\ \mathfrak{s}_{02} & \mathfrak{s}_{12} & \mathfrak{s}_{22} & \mathfrak{s}_{23} \\ \mathfrak{s}_{03} & \mathfrak{s}_{13} & \mathfrak{s}_{23} & \mathfrak{s}_{33}\end{pmatrix}\begin{pmatrix}\Lambda & 0 \\ 0 & 1\end{pmatrix} =$$

$$\begin{pmatrix} b_0 + b_1\vec{v}^2 & b_2 + b_3\vec{v}^2 & b_6|\vec{v}| + b_7|\vec{v}|^3 + 2b_8|\vec{v}| & b_6|\vec{v}| \\ b_2 + b_3\vec{v}^2 & b_4 + b_5\vec{v}^2 & b_{10}|\vec{v}| + b_{11}|\vec{v}|^3 + 2b_{12}|\vec{v}| & b_{10}|\vec{v}| \\ b_6|\vec{v}| + b_7|\vec{v}|^3 + 2b_8|\vec{v}| & b_{10}|\vec{v}| + b_{11}|\vec{v}|^3 + 2b_{12}|\vec{v}| & 2b_{14} + b_{16} + b_{17}\vec{v}^4 + 2b_{18}\vec{v}^2 + 4b_{19}\vec{v}^2 & b_{16} + b_{18}\vec{v}^2 \\ b_6|\vec{v}| & b_{10}|\vec{v}| & b_{16} + b_{18}\vec{v}^2 & b_{16} + \frac{2}{d-1}b_{14} \end{pmatrix},$$

---

[19]The signs of the coefficients $\gamma_1$, $\gamma_3$, $\gamma_4$ in $\Pi^0_j$ in eq. (2.24) of [29] are incorrect, as they violate the Landau frame conditions. We are unable to reproduce the non-hydrostatic combinations reported in eqs. (2.48)–(2.56) in [29].

$$\frac{1}{k_B T} \Lambda^T \begin{pmatrix} \mathfrak{v}_{00} & \mathfrak{v}_{01} & \mathfrak{v}_{02} \\ \mathfrak{v}_{01} & \mathfrak{v}_{11} & \mathfrak{v}_{12} \\ \mathfrak{v}_{02} & \mathfrak{v}_{12} & \mathfrak{v}_{22} \end{pmatrix} \Lambda = \begin{pmatrix} b_0 & b_2 & b_8 \\ b_2 & b_4 & b_{12} \\ b_8 & b_{12} & b_{14} + b_{19}\vec{v}^2 \end{pmatrix},$$

$$\frac{1}{k_B T} \mathfrak{t} = b_{14}, \tag{156}$$

where

$$\Lambda = \begin{pmatrix} -\mu/T & 1 & 0 \\ 1/T & 0 & 0 \\ -|\vec{u}|/T & 0 & 1 \end{pmatrix}, \tag{157}$$

is the transformation matrix arising from converting $\partial_i(\mu/T)$, $\partial_i(1/T)$, $\partial_i(u_j/T)$ basis to $\partial_i T$, $\partial_i \mu$, $\partial_i u_j$ basis in [29]. The comparison also leads to 3 equality constraints[20]

$$b_9 = b_8, \qquad b_{13} = b_{12}, \qquad b_{20} = b_{19}, \tag{158}$$

which can be thought of as arising from requiring entropy production to be non-negative. To map the hydrostatic $\tilde{c}_{1,2,4,8}$ coefficients, we note that the non-canonical entropy current from eqs. (2.30)-(2.32) of [29] is given by

$$
\begin{aligned}
s^0_{\text{non-can}} &= \tilde{c}_1 u^k \partial_k \vec{u}^2 + \tilde{c}_2 \left( u^k \partial_k \frac{\mu}{T} + m_1 \partial_t \vec{u}^2 \right) + \tilde{c}_4 \partial_k u^k, \\
s^i_{\text{non-can}} &= -u^i \tilde{c}_1 \partial_t \vec{u}^2 - \tilde{c}_4 \partial_t u^i + \tilde{c}_8 \left( u^k \partial_k u^i - u^i \partial_k u^k \right) \\
&\quad + m_2 \tilde{c}_2 \partial_t u^i + \tilde{c}_2 u^i \left( m_4 u^k \partial_k \vec{u}^2 + u^k \partial_k \frac{\mu}{T} + (m_1 + m_4) \partial_t \vec{u}^2 + m_3 \partial_k u^k \right),
\end{aligned}
\tag{159}
$$

where $m_{1,2,3,4}$ are some known thermodynamic parameters. This can be compared to our eq. (69), in the absence of background fields, using the equations of motion, and up to a total-derivative shift of the entropy current $s^0_{\text{non-can}} \to s^0_{\text{non-can}} + \partial_i X^i$, $s^i_{\text{non-can}} \to s^i_{\text{non-can}} - \partial_t X^i$ for some $X^i$ that leaves the divergence of the entropy current invariant. This relates 3 independent $\tilde{c}_{1,2,4}$ coefficients to 3 hydrostatic coefficients $F_{0,1,2}$ according to

$$
\begin{aligned}
F_0 &= \tilde{c}_2 - TnX - T\frac{\partial Y}{\partial \mu}, \\
F_1 &= -\frac{\mu}{T}\tilde{c}_2 - TsX - T\frac{\partial Y}{\partial T}, \\
F_2 &= T\tilde{c}_1 - Tm_1\tilde{c}_2 - \frac{1}{2}T\rho X - T\frac{\partial Y}{\partial \vec{u}^2},
\end{aligned}
\tag{160}
$$

where

$$
\begin{aligned}
X &= 2m_1\tilde{c}_2 \left( \frac{1}{\rho} - \frac{\vec{u}^2}{\rho}\frac{\partial \rho}{\partial \epsilon} - 2\vec{u}^2\frac{\partial \rho}{\partial \vec{\pi}^2} \right), \\
Y &= \tilde{c}_4 - 2\vec{u}^2 m_1 \tilde{c}_2 \left( 1 - \frac{\epsilon + p}{\rho}\frac{\partial \rho}{\partial \epsilon} - \frac{n}{\rho}\frac{\partial \rho}{\partial n} - 2\rho\vec{u}^2\frac{\partial \rho}{\partial \vec{\pi}^2} \right).
\end{aligned}
\tag{161}
$$

We also get a constraint[21]

$$\tilde{c}_8 = 0. \tag{162}$$

---

[20]In eq. (2.33) of [29], the authors write the dissipation matrix in terms of $\partial_i T$, $\partial_i \mu$, and $\partial_i u_j$, however, only the symmetric derivatives of $u^i$ are non-hydrostatic and can contribute to dissipation. This leads to the said 3 constraints.

[21]An easy way to understand this constraint is by noting that the contribution to entropy-divergence coupled to $\tilde{c}_8$ does not vanish in equilibrium, and hence must be set to zero.

To summarise, [29] reports a total of 29 coefficients in the constitutive relations $\bar{\sigma}$, $\bar{\alpha}$, $\bar{\gamma}$, $\bar{\pi}$, $\gamma_{1,\dots,23}$ (related to our formalism according to eqs. (153) and (154)). Ref. [29] also classifies 20 possibly dissipative coefficients $b_{0,\dots,14}$, $b_{16,\dots,20}$ in the entropy production quadratic form, 4 possibly hydrostatic coefficients $\tilde{c}_{1,2,4,8}$ in the non-canonical entropy current, along with 9 non-hydrostatic non-dissipative coefficients that do not contribute to the non-canonical entropy current or to entropy production. The ensuing second law analysis was not performed in [29]. Accounting for the second law, we find 3 constraints among the coefficients $b$'s, given in eq. (158), and one constraint among the coefficients $\tilde{c}$'s, given in eq. (162). Thus, the final number of independent transport coefficients consists of 17 dissipative, 9 non-dissipative non-hydrostatic, and 3 hydrostatic.

## B  Interaction vertices

In this appendix, we record the effective Lagrangian that accounts for interactions between hydrodynamic and stochastic degrees of freedom. Taking into account only the ideal order part of the Lagrangian from eq. (72), and expanding to cubic order in fluctuations, we obtain the three-point interaction Lagrangian given by

$$
\begin{aligned}
\mathcal{L}_3 = {} & \frac{1}{2}\gamma_{nn}^i \delta n^2 \partial_i \varphi_a + \frac{1}{2}\gamma_{\epsilon\epsilon}^i \delta\epsilon^2 \partial_i \varphi_a + \gamma_{n\epsilon}^i \delta n\, \delta\epsilon\, \partial_i \varphi_a \\
& + \gamma_{n\pi}^{ij} \delta n\, \delta\pi_i \partial_j \varphi_a + \gamma_{\epsilon\pi}^{ij} \delta\epsilon\, \delta\pi_i \partial_j \varphi_a + \gamma_{\pi\pi}^{ijk} \delta\pi_i \delta\pi_j \partial_k \varphi_a \\
& - \frac{1}{2}\alpha_{nn}^i \delta n^2 \partial_i X_a^t - \frac{1}{2}\alpha_{\epsilon\epsilon}^i \delta\epsilon^2 \partial_i X_a^t - \alpha_{n\epsilon}^i \delta n\, \delta\epsilon\, \partial_i X_a^t \\
& - \alpha_{n\pi}^{ij} \delta n\, \delta\pi_i \partial_j X_a^t - \alpha_{\epsilon\pi}^{ij} \delta\epsilon\, \delta\pi_i \partial_j X_a^t - \alpha_{\pi\pi}^{ijk} \delta\pi_i \delta\pi_j \partial_k X_a^t \\
& + \frac{1}{2}\beta_{nn}^{ij} \delta n^2 \partial_j X_{ai} + \frac{1}{2}\beta_{\epsilon\epsilon}^{ij} \delta\epsilon^2 \partial_j X_{ai} + \beta_{n\epsilon}^{ij} \delta n\, \delta\epsilon\, \partial_j X_{ai} \\
& + \beta_{n\pi}^{ijk} \delta n\, \delta\pi_i\, \partial_j X_{ak} + \beta_{\epsilon\pi}^{ijk} \delta\epsilon\, \delta\pi_i\, \partial_j X_{ak} + \beta_{\pi\pi}^{ijkl} \delta\pi_i \delta\pi_j\, \partial_l X_{ak} \;.
\end{aligned}
\tag{163}
$$

Here we have defined the following coupling structures

$$
\begin{aligned}
\gamma_{nn}^i &= \rho \frac{\partial^2 \hat{n}}{\partial n^2} u^i\,, \qquad \gamma_{\epsilon\epsilon}^i = \rho \frac{\partial^2 \hat{n}}{\partial \epsilon^2} u^i\,, \qquad \gamma_{n\epsilon}^i = \rho \frac{\partial^2 \hat{n}}{\partial n \partial \epsilon} u^i\,, \\
\gamma_{n\pi}^{ij} &= 2\rho^2 \frac{\partial^2 \hat{n}}{\partial n \partial \vec{\pi}^2} u^i u^j + \frac{\partial \hat{n}}{\partial n} \delta^{ij}\,, \qquad \gamma_{\epsilon\pi}^{ij} = 2\rho^2 \frac{\partial^2 \hat{n}}{\partial \epsilon \partial \vec{\pi}^2} u^i u^j + \frac{\partial \hat{n}}{\partial \epsilon} \delta^{ij}\,, \\
\gamma_{\pi\pi}^{ijk} &= 2\rho^3 \frac{\partial^2 \hat{n}}{\partial (\vec{\pi}^2)^2} u^i u^j u^k + \rho \frac{\partial \hat{n}}{\partial \vec{\pi}^2}(\delta^{ij} u^k + 2u^{(i}\delta^{j)k})\,, \\
\alpha_{nn}^i &= \rho \frac{\partial^2 \hat{w}}{\partial n^2} u^i\,, \qquad \alpha_{\epsilon\epsilon}^i = \rho \frac{\partial^2 \hat{w}}{\partial \epsilon^2} u^i\,, \qquad \alpha_{n\epsilon}^i = \rho \frac{\partial^2 \hat{w}}{\partial n \partial \epsilon} u^i\,, \\
\alpha_{n\pi}^{ij} &= 2\rho^2 \frac{\partial^2 \hat{w}}{\partial n \partial \vec{\pi}^2} u^i u^j + \frac{\partial \hat{w}}{\partial n} \delta^{ij}\,, \qquad \alpha_{\epsilon\pi}^{ij} = 2\rho^2 \frac{\partial^2 \hat{w}}{\partial \epsilon \partial \vec{\pi}^2} u^i u^j + \frac{\partial \hat{w}}{\partial \epsilon} \delta^{ij}\,, \\
\alpha_{\pi\pi}^{ijk} &= 2\rho^3 \frac{\partial^2 \hat{w}}{\partial (\vec{\pi}^2)^2} u^i u^j u^k + \rho \frac{\partial \hat{w}}{\partial \vec{\pi}^2}(\delta^{ij} u^k + 2u^{(i}\delta^{j)k})\,, \\
\beta_{nn}^{ij} &= \rho^2 \frac{\partial^2 \rho^{-1}}{\partial n^2} u^i u^j + \frac{\partial^2 p}{\partial n^2} \delta^{ij}\,, \qquad \beta_{\epsilon\epsilon}^{ij} = \rho^2 \frac{\partial^2 \rho^{-1}}{\partial \epsilon^2} u^i u^j + \frac{\partial^2 p}{\partial \epsilon^2} \delta^{ij}\,, \\
\beta_{n\epsilon}^{ij} &= \rho^2 \frac{\partial^2 \rho^{-1}}{\partial n \partial \epsilon} u^i u^j + \frac{\partial^2 p}{\partial n \partial \epsilon} \delta^{ij}\,,
\end{aligned}
\tag{164}
$$

$$\beta_{n\pi}^{ijk} = -\frac{2}{\rho}\frac{\partial \rho}{\partial n}\delta^{i(j}u^{k)} + 2\rho^3 \frac{\partial^2 \rho^{-1}}{\partial n \partial \vec{\pi}^2}u^i u^j u^k + 2\rho \frac{\partial^2 p}{\partial n \partial \vec{\pi}^2}u^i \delta^{jk} \,,$$

$$\beta_{\epsilon\pi}^{ijk} = -\frac{2}{\rho}\frac{\partial \rho}{\partial \epsilon}\delta^{i(j}u^{k)} + 2\rho^3 \frac{\partial^2 \rho^{-1}}{\partial \epsilon \partial \vec{\pi}^2}u^i u^j u^k + 2\rho \frac{\partial^2 p}{\partial \epsilon \partial \vec{\pi}^2}u^i \delta^{jk},$$

$$\beta_{\pi\pi}^{ijkl} = \frac{1}{\rho}\delta^{ik}\delta^{jl} + \frac{\partial p}{\partial \vec{\pi}^2}\delta^{ij}\delta^{kl} - \frac{\partial \rho}{\partial \vec{\pi}^2}(4u^{(i}\delta^{j)(k}u^{l)} + \delta^{ij}u^k u^l)$$

$$+ 2\rho^2 \frac{\partial^2 p}{\partial (\vec{\pi}^2)^2}u^i u^j \delta^{kl} + 2\rho^4 \frac{\partial^2 \rho^{-1}}{\partial (\vec{\pi}^2)^2}u^i u^j u^k u^l \,. \tag{165}$$

This procedure can analogously be iterated to obtain higher derivative and higher-point interactions (see [11] for the discussion in Galilean case). We leave the analysis of the effects of (163) on hydrodynamic equations of motion and correlation functions to future work.

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
