# Peer review of "Effective field theory for hydrodynamics without boosts"

_SciPost Physics, doi:SciPost Phys. 11, 054 (2021)_

## Round 2 · Referee Report · Anonymous (Referee 2) · 2021-5-25

Strengths

1) The manuscript formulated the theory of hydrodynamic applicable for a systems with or without boost symmetry in Schwinger-Keldysh effective action. The authors combined the framework of Glorioso et. al. such as Ref. [8] and by Haehl et. al. [9] in a seemless and understandable manner.

2) The author classify hydrodynamic transport coefficients of a theory with U(1) and translational symmetry which can be viewed as either
i) Extending Ref.[25] to first order in the derivative expansions
ii) Extedning Ref. [26] to the case that also involved U(1) global symmetry.
iii) Correct the wrong counting of transport coefficients in Ref.[28] and confirm that it agrees with zero density limit of [26]. These task requires very careful consideration and efforts which the authors demonstrated in their detailed analysis.
iv) Extend the study of hydrodynamic modes in Ref. [27] to include finite fluid velocity as well as those of [28] to include propagating modes.

3) The author proposed a new frame choice which produce only stable modes in the presence of finite fluid velocity (which is a problem in the Landau frame). This frame choice, in itself, is particularly useful as it can be applied to both Galilean, relativistic or the theory without any boost symmetry.

4) The author not only classify all the transport coefficients and call it a day, they also study the potential observable consequences when there is no boost symmetry as one can see in how the boost affect the property of the speed of sound.

Weaknesses

1) First of all, the title is somewhat misleading. Not only that the presented formulations works in a theory without boost, it is applicable irrespective of boost symmetry (although they are not the first one, see Ref.[26] and [28]). If I may suggest a more accurate title, it would be something along the line of "Effective field theory for hydrodynamics with or boosts without" or Effective field theory for boost-agnostic hydrodynamics" as in section 2 and 4 of the manuscript.

2) It is somewhat unclear where is the role of Schwinger-Keldysh effective theory in this work. For example, one could forget the effective action part of this manuscript and solving adiabatic equation in Section 2 to obtain the same classification. To the best of my knowledge, the power of SK formalism is to incorporate the statistical fluctuations (which ended up in the appendices and bookmarking it for future work) or case of stochastic correlations break the gradient expansoins in Ref.[22] (which was not mentioned again). In all fairness, I think that the analysis presented here is valuable in itself without the, seemingly decorative, part concerning SK effective action that only bloat the manuscript unnecessarily.

3) Despite a nontrivial amount of work and careful analysis in great details, it is an amalgum of of extensions of the previous works (see point 2) in strength section). This does not meant to diminish the quality of the work nor saying that the information presented here is not valuable but I cannot rank it very high in terms of creativity and originality.

4) There are many applications on various fields hep beyond hep-th mentioned in the introduction. However, no comparison between the new result of this manuscript and the aforementioned potential application were made (particularly Ref.[20] and [21] which the authors specifically emphasised). This not only makes the maniscript difficult for the readers in hep-th community who are more familiar with the technology to understand the potential applications but also the reader who, for example, are familiar with Ref[20] to understand all the technicality and terminology used in this very technical work. For this, and the issue raised in 2) of weakness section, I cannot rate the significance of this work higher than the given one.

Report

I think the manuscript contains valuable and highly nontrivial analysis on boost agnostic hydrodynamic descriptions.

There are a few issues pointed out in the requested changes that can be fixed easily.

The premise of the manuscript on potential application seems somewhat exaggerated without more detailed comparison of their presented constructions and what has been done on the applicable topics by different communities. However, I understand that it would requires much more work that could and should be on separated papers.

Despite these weaknesses, I think the quality of the manuscript easily meets the criteria of SciPost. I would recommend a minor corrections to improve the manuscript before publication.

Requested changes

There are a few minor correction to the text, which are not entirely accurante and it will make the article more valuable to the readers upon fixing.

- On page 10: " However these frames are known to exhibit unphysical pathologies such as superluminal propagation and unstable modes in the linear spectrum in a finite velocity state [29]."

The first paper that discussed the issues mentioned here is Ref.[48]in relativistic fluid back in 1985, that already appeared in modern textbooks, not [29]. In fact, Ref[49] is the first one, according to Ref.[29], that formulate a first-order hydrodynamic descriptions that does not have superluminal signal and stable.

This mis-citation does not occur when Ref.[29] was mentioned again on page 11. But it would be great if the same statements were cited consistently.

- On footnote 3: "This framework can also describe Carrollian fluids [23–25] but we have not considered this special limit here."

It is not entirely obvious that the framework here can capture the c-> 0 limit of Carrollian fluid in the sense of Ref [24] that was introduced by Lévy-Leblond (note also the subtle between difference two 0-> c limit in [24] and [25]).
In this limit, the Carrollian Ward identity implies that the spatial part of energy current and/or U(1) current vanish identically (see e.g. Section 3.3. of [24] and [30]), which is difficult to see in (2.9) and (2.21) of the manuscript.
Another indication of the Carrollian fluid is the absence of sound in stationary fluid which does seems unobtainable from Eq.(6.7). It would be nice if the author can clarify this issues or remove the statement about Carollian fluid.

- On page 3: "In the context of quantum matter, spatial translations are usually also
spontaneously broken, due to the presence of the ionic lattice, or explicitly broken due to the presence
of impurities [18]. Charge density wave phases are one such example [19]"

I don't think both references are mentioned appropriately here. The above statements are textbook material so perhaps a book by Chaikin & Lubensky is more appropriate for both cases, particularly a well-known example such as charge density wave. Ref [18], while relevant to CDW phase, is dealing with the case where both explicitly and sponteneously broken translational symmetry occur in the same system.

  • validity: top
  • significance: good
  • originality: ok
  • clarity: high
  • formatting: excellent
  • grammar: excellent

Author:  Akash Jain  on 2021-08-10  [id 1649]

(in reply to Report 2 on 2021-05-25)

We thank the referee for the comments on our manuscript and apologise for the delay in our response. We agree with the referee's recommendation about the mis-citations and we have fixed these in the new version. Regarding Carrollian fluids, we have not explored the relevant technicalities in any detail. In line with the referee's suggestion, we have weakened the statement of footnote 3. We also have referred to Chaikin and Lubensky for generic cases of spontaneous and explicit symmetry breaking, as suggested by the referee.

We hope that the referee finds our manuscript publishable in its present state.

---

## Round 2 · Referee Report · Anonymous (Referee 3) · 2021-5-25

Strengths

1. The paper provides the first systematic formulation of hydrodynamics without boost symmetries to first order in a derivative expansion using modern effective field theory techniques.
2. It sufficiently addresses comparisons with existing literature when available.
3. It is well-written, clear, and concise.

Report

The paper provides a novel link between the modern theoretical framework of hydrodynamics and the realm of soft matter applications with a clear potential for follow-up work, meeting criteria 4 and 5 of the expectations of SciPost submissions. Moreover, the paper is well written, concise, and structured, and as such it meets all the general acceptance criteria too. For these reasons I recommend the paper for publication provided one important issue is resolved and some more minor changes are addressed.

Requested changes

Important:

1. Even after reading the details provided in [11], it is not clear Eq. 3.18b is the correct requirement. If we understood correctly, ${\cal D}_2$ is a generic derivative operator which acts on the arguments in parenthesis. Say we allow this operator to act only on the second argument. We can write it as an operator as follows

${\cal D}_2(\Phi_a,\Phi_a+i\delta_{\cal B}\Phi_r) =\Phi_a {\cal H}_2(\Phi_r,\partial \Phi_r, \dots,{\cal B},\partial)(\Phi_a+i\delta_{\cal B}\Phi_r) $

where ${\cal H}_2$ is a generic operator which depends on $\Phi_r$, its derivatives, the background fields ${\cal B}$ and generic derivatives acting on the field that follows.

Now, let us apply KMS conjugation 3.14. We get

$KMS(\Phi_a {\cal H}_2(\Phi_r,\partial \Phi_r,{\cal B},\partial)(\Phi_a+i\delta_{\cal B}\Phi_r))=(\Theta\Phi_a+i\Theta\delta_{\cal B}\Phi_r){\cal H}_2(\Theta\Phi_r,\Theta(\partial \Phi_r)\Theta{\cal B},\partial)\Theta \Phi_a$

If $\Phi_a{\cal H}_2(\Phi_r,\partial \Phi_r,{\cal B},\partial)\Phi_a$ is $\Theta$-even, presumably we have

$\Theta\Phi_a{\cal H}_2(\Theta\Phi_r,\Theta(\partial \Phi_r)\Theta{\cal B},\partial)\Theta\Phi_a=\Phi_a(-x){\cal H}_2(\Phi_r(-x),\partial \Phi_r(-x){\cal B}(-x),\partial)\Phi_a(-x)=\\
=\Phi_a(x){\cal H}_2(\Phi_r(x),-\partial \Phi_r(x),{\cal B}(x),-\partial)\Phi_a(x)$

where in the last equality we have assumed the term appears inside the sign of integral, we can perform a change of coordinates and the Jacobian of the transformation is 1. While the combined $\Theta$ eigenvalue of the whole term is 1 ($\Theta$-even), the arguments of the various functions are changed. Therefore we have

$KMS(\Phi_a {\cal H}_2(\Phi_r,\partial \Phi_r,{\cal B},\partial)(\Phi_a+i\delta_{\cal B}\Phi_r))=\\
=\Phi_a{\cal H}_2(\Phi_r,-\partial \Phi_r,{\cal B},-\partial) \Phi_a+i\Theta\delta_{\cal B}\Phi_r{\cal H}_2(\Theta\Phi_r,\Theta(\partial \Phi_r),\Theta{\cal B},\partial)\Theta \Phi_a$

Considering only the first term we are lead to conclude that

$\Phi_a{\cal H}_2(\Phi_r,\partial \Phi_r,{\cal B},\partial) \Phi_a=\Phi_a{\cal H}_2(\Phi_r,-\partial \Phi_r,{\cal B},-\partial) \Phi_a$

which seems to imply that $\Phi_a{\cal H}_2\Phi_a$ has to be even in derivatives if also $\Theta$-even. It is less clear what happens to the second term.

We are kindly asking the authors to clarify their arguments leading to 3.18b and/or provide where the above chain of thoughts is erroneous.

Minor:

- Second equation in 2.9: $t^{ij}\rightarrow \tau^{ij}$?
- End of page 7: it would be useful if the authors specify $d=$ number of spatial dimensions?
- Below 2.28 "...if..." $\rightarrow$ "...is..."?
- Below Equations 2.31 the nomenclature hydrostatic and non hydrostatic is not explained. Therefore the comment is not clear. It would be desirable if the authors could elaborate.
-3.3c : should the second equation contain also a diffeomorphism transformation of the argument of the scalar function?
- First paragraph Section 4.2, $X^{\mu}_a$ and $\varphi_a$ are taken to be ${\cal O}(\partial^0)$. Is that correct given the subsequent sentence where $\phi_a$ are taken to be ${\cal O}(\partial^1)$?
- First paragraph Section 6 "non-hydrodynamic..." $\rightarrow$ "non-hydrostatic"?

  • validity: good
  • significance: good
  • originality: good
  • clarity: high
  • formatting: perfect
  • grammar: perfect

Author:  Akash Jain  on 2021-08-10  [id 1648]

(in reply to Report 1 on 2021-05-25)

We thank the referee for the comments on our manuscript and apologise for the delay in our response. Regarding the important point raised by the referee, first off, we would like to note that the ${\cal D}_2$ operator needs to be symmetric in its arguments, i.e. $\mathcal{D}_2(A,B) = \mathcal{D}_2(B,A)$. Therefore, the definition suggested by the referee should be modified to

$$ {\cal D}2(\Phi_a,\Phi_a+i\delta\Phi_r) = \Phi_a {\cal H}2(\Phi_r,\partial \Phi_r, \dots,{\cal B},\partial\mathcal{B},\ldots,\partial)(\Phi_a+i\delta\Phi_r) + \left(\Phi_a + i\delta_{\cal B}\Phi_r \right) {\cal H}_2(\Phi_r,\partial \Phi_r, \dots,{\cal B},\partial\mathcal{B},\ldots,\partial) \Phi_a. $$
However, this has no bearing on the argument presented by the referee. As for the confusion, we notice that at no point has the referee actually used the $\Theta$-even constraint in the argument. Let us elaborate. The $\Theta$-even constraint for any term $\mathcal{X}(\psi,\partial\psi,\ldots)$ in the Lagrangian, where $\psi$ collectively denotes all the fields, requires that $\mathcal{X}$ must transform as a scalar under the $\Theta$ transformation. In other words
$$ \mathcal{X}(\Theta\psi,\Theta\partial\psi,\ldots) = \mathcal{X}(\psi,\partial\psi,\ldots)\big|_{x\to\Theta x}. $$
For instance, for a scalar field $\psi$ and $\Theta=\text{PT}$, the term $\partial_\mu\psi\partial^\mu\psi$ is allowed in the Lagrangian, but $\psi\partial_t\psi$ is not. So, $\mathcal{H}_2$ being even in derivatives is not an independent constraint, but a consequence of the $\Theta$-even condition.

As for the minor points, we agree with most of the comments raised by the referee and have made the necessary edits in our draft. For the fifth point, however, we do not think there is an error in eq. (3.3c). Similarly, for the sixth point, we believe that the derivative ordering in section 4.2 is correct. Please look at, for instance, eq. (3.8b); the "$a$" type background fields naturally appear at one-derivative order higher than the "$a$" type dynamical fields.

We hope that the referee finds our manuscript publishable in its present state.

Anonymous on 2021-08-23  [id 1698]

(in reply to Akash Jain on 2021-08-10 [id 1648])

We would like to thank the authors for patiently answering our questions and clarifications requests.
We understand KMS symmetry restricts ${\cal D}_2$ to be symmetric in its arguments and the CPT requirements will follow. We consider ourselves satisfied with the discussion and recommend this paper for publication without further modifications.

Anonymous on 2021-08-20  [id 1697]

(in reply to Akash Jain on 2021-08-10 [id 1648])
Category:
answer to question

We thank the referee for their question. The functions $\cal D_m$ have to be symmetric in the multivariable sense. Let us take $\cal D_2(\Phi,\Phi')$ with arbitrary arguments $\Phi = (N,H,B)$ and $\Phi' = (N',H',B')$. The symmetric property requires that ${\cal D}_2(\Phi,\Phi') = {\cal D}_2(\Phi',\Phi)$. In contrast, the terms that are antisymmetric in the arguments would behave as ${\cal D}_2(\Phi,\Phi') = -{\cal D}_2(\Phi',\Phi)$. These will of course be zero when we set $\Phi=\Phi'=\Phi_a$.

The referee seems to be confused with the multiple variables involved in the definition of $\Phi_a$. However, this does not have any bearing on the arguent presented above. Let us expand a generic ${\cal D}_2(\Phi,\Phi')$ as

$$ {\cal D}_2(\Phi,\Phi') = {\cal D}^{11}_2(N,N') + {\cal D}^{12}_2(N,H') + {\cal D}^{13}_2(N,B') + {\cal D}^{21}_2(H,N') + {\cal D}^{22}_2(H,H') + {\cal D}^{23}_2(H,B') + {\cal D}^{31}_2(B,N') + {\cal D}^{32}_2(B,H') + {\cal D}^{33}_2(B,B') $$

To say that the $ {\cal D}_2(\Phi,\Phi') $ is symmetric in its arguments is to say that ${\cal D}_2(\Phi,\Phi') = {\cal D}_2(\Phi',\Phi)$. This would imply that the diagonal terms ${\cal D}^{11}_2(N,N') $, ${\cal D}^{22}_2(H,H')$, ${\cal D}^{33}_2(B,B') $ are symmetric in their arguments, while the off-diagonal terms equate to each other

$$ {\cal D}^{12}_2(N,H') = {\cal D}^{21}_2(H',N), \qquad {\cal D}^{13}_2(N,B') = {\cal D}^{31}_2(B',N), \qquad {\cal D}^{23}_2(H,B') = {\cal D}^{32}_2(B',H)$$

On the other hand, terms "antisymmetric" in the arguments behave like

$$ {\cal D}^{12}_2(N,H') = -{\cal D}^{21}_2(H',N), \qquad {\cal D}^{13}_2(N,B') = -{\cal D}^{31}_2(B',N), \qquad {\cal D}^{23}_2(H,B') = -{\cal D}^{32}_2(B',H)$$

and trivially drop out from the expression of $ {\cal D}_2(\Phi,\Phi') $ on setting $\Phi = \Phi' = \Phi_a$. This should also be clear in the context of the discussion in section 4.2. In particular, the matrix in eq. (4.19) is symmetric. This same construction also appears in section 5.2.3 of [https://arxiv.org/pdf/2008.03994.pdf] and is not specific to boost-agnostic fluids.

We hope that this answers the question of the referee.

Anonymous on 2021-08-16  [id 1675]

(in reply to Akash Jain on 2021-08-10 [id 1648])

We thank the authors for their clarifications. We understand how the ${\cal D}_m$ operators are required to be symmetric in their arguments by construction in the context of [https://arxiv.org/pdf/2008.03994.pdf]. However, we believe the same arguments do not follow for the present work given that $\Phi_a = \{ N_{a\mu},H_{a\mu\nu},B_{a\mu} \}$. For example, in a term like ${\cal L}\sim {\cal F}_2(N_{a\mu},B_{a\mu})$ the antisymmetric contributions do not seem to trivially drop out of the Lagrangian.

We urge the authors to provide more clarifications or to reconsider their claims in the central Section 3.3.

Anonymous on 2021-08-11  [id 1655]

(in reply to Akash Jain on 2021-08-10 [id 1648])
Category:
answer to question

We thank the referee for their immediate response. The $\cal D_m$ operators are required to be symmetric in their arguments by construction. We mention this in the first paragraph of section 3.3. A detailed derivation of this fact can be found in section 3.4 of [https://arxiv.org/pdf/2008.03994.pdf]. This amounts to the fact that, to begin with, the effective action can be organized as a Taylor series in $\Phi_a$, i.e.
$$ {\cal L} \sim {\cal F}_m(\Phi_a,\Phi_a,\ldots)$$
The operators ${\cal F}_m$ are by construction symmetric in their arguments because the antisymmetric contributions will trivially drop out of the Lagrangian. Now the $\cal D_m$ operators are just particular linear combinations of the $\cal F_m$ operators, given in eq. (3.28) of [https://arxiv.org/pdf/2008.03994.pdf]. So they are also symmetric by construction. We hope this answers the referee's question.

Anonymous on 2021-08-10  [id 1653]

(in reply to Akash Jain on 2021-08-10 [id 1648])
Category:
objection

We thank the authors for taking the time to answer our questions and clarification requests which have been addressed sufficiently in detail. Nevertheless, it is still unclear:

  • Why should $D_m$ be totally symmetric in its arguments? If it is not, are $\Theta$-odd terms allowed? If so, would this change the authors' conclusions?

We kindly ask the authors to clarify this urgent point before I can recommend this paper for publication on SciPost.

---

## Round 2 · Referee Report · Anonymous (Referee 1) · 2021-6-1

Report

The paper is well-written, interesting and makes valuable contributions towards the development of hydrodynamics without boost symmetry, building on existing works. They have made significant efforts to connect their work with existing results. It meets the criteria for publication in SciPost, but I would like to see the following points addressed:

1. The authors present a confused message about whether they are discussing hydrodynamics, or an exact one-derivative model which results from truncation of hydrodynamics. However, the distinction is important because it is useful to understand which of their results are universal hydrodynamic results and which are specific to their choice of truncated model. This issue is exacerbated by the authors discussion of frame choices; the authors have presented a choice of frame for which the one-derivative truncated theory is linearly stable and contains no gapped modes in the UHP. Whilst this is a nice observation and the resulting exact theory is good for initial-value problems as indicated by the authors, a statement of the spectrum of gapped modes in general sits outside the scope of the hydrodynamic expansion (at least at these low orders). The paper would benefit significantly from a clarification of which statements are universal hydrodynamic results and which are not.

2. I read "boost-agnostic" as "Lorentz invariant" since Lorentz invariant physics is agnostic to the choice of Lorentz frame. Perhaps "boost symmetry-agnostic" or similar would be more appropriate.

  • validity: good
  • significance: good
  • originality: good
  • clarity: high
  • formatting: perfect
  • grammar: perfect

Author:  Akash Jain  on 2021-08-10  [id 1650]

(in reply to Report 3 on 2021-06-01)

We thank the referee for the comments on our manuscript and apologise for the delay in our response. If we understand the first comment correctly, the referee is uncertain whether our construction pertains to only one-derivative truncated hydrodynamics or does it generalise to higher-derivative corrections as well. As such, much of our formal discussion in sections 2 and 3 applies to all-order hydrodynamics, but from section 4 onwards we are only discussing the specific example of hydrodynamics truncated at one-derivative order. As for the applicability of our results, the hydrodynamic framework, irrespective of the order at which the constitutive relations are truncated, is only trustworthy perturbatively at low-energy and low-wavelength. Hydrodynamics is simply not appropriate to predict the behaviour of gapped modes. In this sense, the absence of unstable gapped modes in the spectrum is not meant to be a physical result. Rather, it illustrates that we have a "better model" for hydrodynamics that agrees with other hydrodynamic models (i.e. other hydrodynamic frames) in the "hydrodynamic regime of applicability". Coming back to the referee's question, everything that we say (or any hydrodynamic model says) about low-energy modes is a "universal hydrodynamic result", while anything to do with gapped modes is "model-dependent". Our message is that if one was only interested in low-energy physics, our model provides a viable alternative as compared to other hydrodynamic models in the literature due to a well-defined initial value problem. If one was interested in high(er)-energy physics instead, hydrodynamics is not a good place to be looking at anyway. We have added a few comments in the introduction section, in particular in the second to last paragraph.

As for the nomenclature "boost-agnostic", we sympathise with the referee's point of view. We are certainly not the ones to coin the phrase; see e.g. [https://arxiv.org/pdf/2004.10759.pdf]. However, it is undoubtedly catchy and the best defence we can offer is that the phrase "boost-agnostic" should be understood as being "agnostic of the type of boost" rather than "agnostic of the amount of boost".

---

## Round 3 · Author Response

We have fixed some references and typos according to the referees' suggestions.

---

## Round 3 · List of Changes

1. Fixed typos in eq. (2.9), below eq. (2.28), and in the first paragraph of section 6.
2. Added clarifying comments on page 3, in footnote 3, above eq. (2.14), and below eq. (2.31).
3. Fixed citations on page 3 and in section 2.2.3.

---

## Editorial Decision

published